# Transfer Learning in Nonparametric Regression with Deep ReLU Networks

**Junpeng Ren** [1]  **Carlos Misael Madrid Padilla** [2]  **Yanzhen Chen** [3]  **Oscar Hernan Madrid Padilla** [1]

## Abstract

This paper develops a general transfer learning framework for nonparametric regression with data consisting of multiple groups. Under the assumption that groups share a common structure along with group-specific deviations in additive form, the proposed method employs a two-stage offset learning procedure: the first stage pools data from all groups to estimate an overall mean function, and the second stage estimates offsets for each group, yielding final group-level estimators through additive combination. Upper bounds on the $\mathcal{L}_2$ error are established for the proposed framework, covering a broad class of nonparametric estimators under mild complexity and noise conditions. When instantiated with deep ReLU networks, explicit convergence rates are derived under hierarchical composition models, demonstrating the ability to overcome the curse of dimensionality. Conditions that enable positive transfer with faster rates are considered, including learning with simpler functions and data augmentation through pooling samples across groups. Various simulations and real-data experiments further validate the effectiveness of the proposed method.

## 1. Introduction

Nonparametric regression is a flexible modeling paradigm that captures complex data relationships without assuming specific functional forms. With the increasing diversity of modern data, transfer learning provides an effective framework for leveraging information from related sources to improve performance on a target, achieving remarkable empirical success in fields such as natural language processing (Daumé III, 2007; Howard & Ruder, 2018), computer vision (Gong et al., 2012; Tzeng et al., 2017), bioinformatics (Schweikert et al., 2008), transportation (Lu et al., 2019), and epidemiology (Apostolopoulos & Mpesiana, 2020). Consequently, enhancing nonparametric regression through transfer learning becomes a meaningful problem. Meanwhile, the growing complexity of modern data is also reflected in higher dimensionality, as exemplified by text and image data. Deep neural networks (DNNs), as a cornerstone class of nonparametric estimators, have demonstrated outstanding empirical performance on such tasks (Krizhevsky et al., 2012; Vaswani et al., 2017; Radford et al., 2018) and have been theoretically shown to possess strong advantages in modeling such high-dimensional nonlinear structures (Schmidt-Hieber, 2020; Kohler & Langer, 2021; Chen et al., 2022). Therefore, studying how transfer learning can further improve neural network estimators provides a practically meaningful instantiation of this problem. Motivated by these developments, this work investigates nonparametric regression under the transfer learning framework, focusing on deep neural networks as the primary estimator.

We formalize nonparametric regression with data composed of distinct groups that share structural similarities. Given $n$ independent copies of $(X, Y, Z)$, denoted by $\{(x_i, y_i, z_i)\}_{i=1}^n$, where $x_i \in \mathcal{X} \subset \mathbb{R}^p$ represents covariates, $y_i \in \mathbb{R}$ is the response, and $z_i \in \{1, \ldots, L\}$ indexes the group membership, we assume the data are generated from the model

$$y_i = f_0(x_i) + f_{0,z_i}(x_i) + \epsilon_i, \tag{1}$$

where $\epsilon_i$ are independent errors with $\mathbb{E}(\epsilon_i \mid x_i, z_i) = 0$. Here, $f_0$ represents a shared function common to all groups, while $f_{0,\ell}$ denotes the group-specific deviation for group $\ell \in \{1, \ldots, L\}$. Our goal is to estimate the group-specific conditional mean function for each group:

$$g_{0,\ell}(x) := \mathbb{E}(Y \mid X = x, Z = \ell) = f_0(x) + f_{0,\ell}(x). \tag{2}$$

To tackle this problem, we propose a two-stage offset transfer learning framework inspired by pretraining. In the first stage, we pool data from all $L$ groups to estimate the overall conditional mean function, defined as the average of the conditional mean functions across all groups. In the second stage, for each target group $\ell$, we use its group-specific data

---

[1] Department of Statistics and Data Science, University of California, Los Angeles, California, USA [2] Department of Statistics and Data Science, Washington University in St. Louis, Missouri, USA [3] Department of Information Systems, Business Statistics and Operations Management, Hong Kong University of Science and Technology, Hong Kong, China. Correspondence to: Oscar Hernan Madrid Padilla <oscar.madrid@stat.ucla.edu>.

*Proceedings of the 43rd International Conference on Machine Learning*, Seoul, South Korea. PMLR 306, 2026. Copyright 2026 by the author(s).

to estimate the offset between the overall mean function and the group-specific conditional mean. The final estimator combines the overall mean function with the estimated group-specific offset in an additive form.

Our main contributions are summarized as follows:

**General Theoretical Framework.** We study a general transfer learning framework for nonparametric regression. In Theorem 3.2, we derive an $\mathcal{L}_2$ error upper bound for a broad class of nonparametric estimators under mild complexity and noise conditions that accommodate sub-exponential noise. We additionally study the variant where the two stages are estimated on independent data via sample splitting, and provide a complementary general bound (Theorem 3.7) that yields a tighter rate. Notably, our framework accommodates the regime in which the number of groups $L$ grows with the total sample size $n$, capturing the essence of modern machine learning on data of growing size and diversity, in contrast to existing transfer learning results that focus on a fixed number of groups. As a direct consequence of the generality of our framework, we yield the first convergence guarantees for trend filtering (Tibshirani, 2014) in a transfer learning setting (Appendix D.1), and recover the existing convergence rates established by (Wang et al., 2016) for orthogonal series regression using Sobolev sieves up to logarithmic factors (Appendix D.2).

**Transfer Learning with Dense ReLU Networks.** Building on the general theory, in Theorem 3.4 and Corollary 3.8 we derive explicit upper bounds for transfer learning with dense ReLU neural networks under hierarchical composition models (Kohler & Langer, 2021), showing that the ability to overcome the curse of dimensionality is preserved within the transfer learning framework. As emphasized in (Cagnetta et al., 2024; Danhofer et al., 2025), such property remains critical for explaining the empirical success of DNNs, even as dataset sizes have scaled dramatically in modern architectures such as ImageNet (Deng et al., 2009), ResNet (He et al., 2016), and GPT (Brown et al., 2020). We further identify regimes in which transfer learning with DNNs achieves strictly faster convergence rates than single-group estimation, providing theoretical justification for the empirical success of pretrained models. Through extensive simulations across diverse scenarios and two real-data applications, we demonstrate that the proposed estimators consistently outperform a range of competitors.

## 1.1. Related Literature

**Transfer Learning.** There are vast works of transfer learning having been discussed, including but not limited to transfer learning in models like linear regression (Chen et al., 2015), high dimensional linear regression (Gross & Tibshirani, 2016; Bastani, 2021; Li et al., 2022; Lai et al., 2024), high dimensional generalized linear model (Tian &

Feng, 2023; Li et al., 2024), functional linear regression (Lin & Reimherr, 2022), and graphical models (Li et al., 2023). Various regularization schemes have also been explored to facilitate transfer, including $\ell_1$ (Craig et al., 2026), $\ell_2$ (Duan & Wang, 2023), and graph-based penalties (Dinh et al., 2022).

In the nonparametric regime, (Cai & Wei, 2021) investigates transfer learning for classification, while a series of works (Wang & Schneider, 2015; Wang et al., 2016; Du et al., 2017; Lin & Reimherr, 2024; Cai & Pu, 2024) focus on nonparametric regression with kernel-based estimators. Specifically, (Wang & Schneider, 2015) introduces a two-stage kernel ridge regression (KRR) framework that models the target function as an additive combination of a source function and an offset, demonstrating that transfer learning can improve estimation when the offset is smoother than the target. (Wang et al., 2016) further formalizes this framework under Sobolev smoothness assumptions. Subsequent work (Du et al., 2017) generalizes it via a nonlinear transformation linking the source and target functions, and (Lin & Reimherr, 2024) develops adaptive KRR algorithms that automatically adjust to unknown smoothness levels. Beyond KRR, (Cai & Pu, 2024) considers transfer learning using local polynomial estimators. In contrast to these studies, our theory establishes convergence rates for general nonparametric estimators and achieves faster rates that overcome the curse of dimensionality for hierarchically composed function classes by leveraging deep neural networks.

Another important line of research studies transfer learning through shared representations, where source and target tasks are assumed to depend on a common low-dimensional structure along with task-specific prediction functions (Maurer et al., 2016; Du et al., 2020; Tripuraneni et al., 2020; 2021; Xu & Tewari, 2021; Tian et al., 2023). However, most previous studies in this line discuss neural networks under parametric formulations, including the analyses in (Tripuraneni et al., 2020; Xu & Tewari, 2021). A recent work (Jiao et al., 2024) studies nonparametric transfer with deep ReLU networks under a representation learning framework. Our work differs by adopting an additive formulation and offering relaxed conditions on noise and network constraints, establishing rates that overcome the curse of dimensionality.

Transfer learning with shared representations is also closely tied to multi-task learning, which jointly learns multiple related tasks in parallel to improve performance, often by assuming shared structures across tasks, with a rich body of early work (Caruana, 1997; Baxter, 2000; Evgeniou & Pontil, 2004; Argyriou et al., 2008). The multi-task learning setting typically treats tasks symmetrically and aims to estimate functions for all tasks jointly without distinguishing source and target, in contrast to the transfer learning formulation of leveraging a data-rich source for a data-scarce

target. However, the distinction between transfer learning and multi-task learning is not sharply drawn in the literature; for instance, the works cited above (Tripuraneni et al., 2020; Jiao et al., 2024) use transfer learning terminology while adopting symmetric multi-task learning setups, or analyze transfer to a data-scarce target within a multi-task framework. We retain the terminology of transfer learning throughout this work. Nevertheless, our framework accommodates both regimes within a unified theory: the target group proportion over the whole dataset is allowed to vanish as the total sample size $n$ grows (transfer learning with data-scarce target), while our algorithm simultaneously produces estimators for all groups on equal footing (multi-task learning).

**Deep Neural Networks as Nonparametric Estimators.** A fundamental challenge in nonparametric estimation is the curse of dimensionality (Donoho et al., 2000). Recent theoretical work shows that deep ReLU networks can mitigate this issue under structured function classes, such as hierarchical composition and low-dimensional manifold assumptions (Schmidt-Hieber, 2020; Kohler & Langer, 2021; Chen et al., 2022; Padilla et al., 2022; Jiao et al., 2023; Padilla et al., 2024b;a). In particular, hierarchical composition has been identified as a natural and practically relevant assumption for explaining the empirical success of deep networks (Cagnetta et al., 2024). Building on this line of research, we study transfer learning with deep ReLU networks under hierarchical composition models and establish explicit convergence rates within our two-stage framework.

### 1.2. Notation

The following notation is used throughout the article: $\mathbb{Z}^+$ denotes the set of positive integers, and $\mathbb{N}$ denotes the set of natural numbers. For two positive sequences $a_n$ and $b_n$, let $a_n = O(b_n)$ or $a_n \lesssim b_n$ when $a_n \leq C b_n$ for some constant $C > 0$ which is independent of $n$, and $a_n = \Theta(b_n)$ or $a_n \asymp b_n$ when $a_n = O(b_n)$ and $b_n = O(a_n)$. Besides, we write $a_n = O(\text{poly}(\log n))$ if there exists a polynomial $h$ such that $a_n = O(h(\log n))$. For a sequence of random variables $X_n$ and a positive sequence $a_n$, we write $X_n = O_{\mathbb{P}}(a_n)$ if for any $\varepsilon > 0$, there exist constants $M > 0$ and $N > 0$ such that $\mathbb{P}(|X_n| > M a_n) < \varepsilon$ for all $n > N$. The $\mathcal{L}_\infty$-norm of a function $f(\cdot) : \mathbb{R}^d \to \mathbb{R}$ is defined by $\|f\|_\infty = \sup_{\mathbf{x} \in \mathbb{R}^d} |f(\mathbf{x})|$. Given the probability distribution $\mathbb{P}_{\mathbf{X}}$ of the random vector $\mathbf{X}$ over $\mathcal{X}$, define the $\mathcal{L}_2(\mathbb{P}_{\mathbf{X}})$-norm as $\|f\|_{\mathcal{L}_2(\mathbb{P}_{\mathbf{X}})} := \left(\int_{\mathcal{X}} f^2(\mathbf{x}) \mathbb{P}_{\mathbf{X}}(d\mathbf{x})\right)^{1/2} = \mathbb{E}\left[f^2(\mathbf{X})\right]^{1/2}$. Besides, given $n$ random samples $\mathbf{x}_1^n := \{\mathbf{x}_i\}_{i=1}^n$ independently and identically distributed according to $\mathbb{P}_{\mathbf{X}}$, the corresponding empirical probability measure is $\mathbb{P}_n(\mathbf{x}_1^n) := \frac{1}{n} \sum_{i=1}^n \delta_{\mathbf{x}_i}(\mathbf{x})$. Define the empirical $\mathcal{L}_2$-norm as $\|f\|_{\mathcal{L}_2(\mathbb{P}_n)} := \left(\frac{1}{n} \sum_{i=1}^n f^2(\mathbf{x}_i)\right)^{1/2} = \left(\int_{\mathcal{X}} f^2(\mathbf{x}) \mathbb{P}_n(d\mathbf{x})\right)^{1/2}$. For a metric space $(\mathcal{X}, d)$, let

$\mathcal{K} \subseteq \mathcal{X}$ and $r > 0$. A subset $C \subset \mathcal{X}$ is an $r$-external cover of $\mathcal{K}$ if $\mathcal{K} \subseteq \bigcup_{x \in C} B_r(x, d)$, where $B_r(x, d)$ denotes the ball of radius $r$ centered at $x$. The external covering number $N(r, \mathcal{K}, d)$ is the minimal cardinality among all $r$-external covers of $\mathcal{K}$. For any index set $\mathcal{I} \subseteq \{1, \ldots, n\}$, we use $\|f\|_{\mathcal{I}}$ to denote the empirical $\mathcal{L}_2$-norm computed over the subset of data points $\{\mathbf{x}_i : i \in \mathcal{I}\}$. For a function $f : \mathbb{R}^d \to \mathbb{R}$ and $\mathcal{A}_n > 0$, let $f_{\mathcal{A}_n}$ denote the truncation of $f$ at level $\mathcal{A}_n$, defined as $f_{\mathcal{A}_n}(x) = \max\{-\mathcal{A}_n, \min\{f(x), \mathcal{A}_n\}\}$.

### 1.3. Outline

The remainder of this paper is organized as follows. Section 2 introduces the general methodology of the proposed transfer learning framework. Section 3.1 establishes general theoretical upper bounds for our estimator, covering a broad class of nonparametric estimators. Section 3.2 provides concrete upper bounds specialized to deep ReLU networks, with discussion on conditions for positive transfer. Section 3.3 further provides theoretical results for the independent two-stage transfer learning estimator based on data splitting. Section 4 presents simulation studies and real-data experiments, comparing the proposed transfer learning approach with alternative training strategies and estimators in both low- and high-dimensional settings. Finally, Section 5 concludes the paper and discusses potential extensions.

## 2. Methodology

We formalize our transfer learning framework based on a two-stage offset learning procedure, which shares similarities with the approaches in (Wang & Schneider, 2015; Wang et al., 2016; Du et al., 2017; Lin & Reimherr, 2024). Under model (1), we define the overall mean function $\bar{f}(x) := \mathbb{E}(Y \mid X = x)$, which can be written as

$$\bar{f}(x) = f_0(x) + \sum_{\ell=1}^L \mathbb{P}(Z = \ell \mid X = x) \, f_{0,\ell}(x).$$

In the first stage, we use a nonparametric estimator to estimate $\bar{f}$ based on data pooled from all groups. Let $\mathcal{F}$ denote the function class of the nonparametric estimator. We construct the estimator as

$$\hat{f} := \arg\min_{f \in \mathcal{F}} \left\{ \sum_{i=1}^n (y_i - f(x_i))^2 \right\}, \qquad (3)$$

and consider its clipped version $\hat{f}_{\mathcal{A}_n}$ with $\mathcal{A}_n > 0$. Next, in the second stage, for $\ell = 1, \ldots, L$, we construct group-specific estimators for the offset between the overall mean $\mathbb{E}(Y|X = x)$ and the group-specific mean function $g_{0,\ell}$ as

$$\hat{f}_\ell := \arg\min_{f \in \mathcal{F}_\ell} \left\{ \sum_{i \,:\, z_i = \ell} (y_i - \hat{f}_{\mathcal{A}_n}(x_i) - f(x_i))^2 \right\}, \quad (4)$$

where $\mathcal{F}_\ell$ denotes the function class for the group-specific nonparametric estimator. The group data used in this stage may be either identical or independent of the data used in the first stage. As in the first stage, we consider the truncated version of the second-stage estimator, denoted by $\hat{f}_{\ell,\mathcal{B}_n}$. The final estimator, which is our main object of theoretical interest, is then constructed as the sum of the two estimators:

$$\hat{g}_\ell(x) := \hat{f}_{\mathcal{A}_n}(x) + \hat{f}_{\ell,\mathcal{B}_n}(x). \quad (5)$$

### 2.1. Background of Deep ReLU Neural Networks as Nonparametric Estimators

We now introduce the basic notation for deep ReLU neural networks, which serve as the primary class of nonparametric estimators in this paper. Let $\tau(x) = \max\{0, x\}$ denote the ReLU activation function, applied componentwise to vectors. A fully connected feedforward neural network with $M$ hidden layers defines a function

$$f = \Phi_M \circ \tau \circ \Phi_{M-1} \circ \cdots \circ \tau \circ \Phi_1,$$

where each $\Phi_s : \mathbb{R}^{w_{s-1}} \to \mathbb{R}^{w_s}$ is an affine map of the form $\Phi_s(x) = W_s x + b_s$. Here $w_0 = d$, $w_M = 1$, $W_s \in \mathbb{R}^{w_s \times w_{s-1}}$, and $b_s \in \mathbb{R}^{w_s}$.

Following (Kohler & Langer, 2021; Padilla et al., 2024a), we restrict attention to dense architectures in which all hidden layers have the same width $\nu$. We denote by $\mathcal{F}(M, \nu)$ the corresponding class of deep ReLU networks. Under the proposed transfer learning framework, the first-stage estimator $\hat{f}$ is obtained by empirical risk minimization over $\mathcal{F}(M, \nu)$, while the second-stage group-specific offset estimator $\hat{f}_\ell$ is obtained from an analogous class $\mathcal{F}(M_\ell, \nu_\ell)$.

## 3. Theory

In this section, we provide theoretical guarantees for the two-stage offset learning framework introduced in Section 2. Suppose the covariate space is $\mathcal{X} = [0,1]^d$, which is standard in nonparametric theory (Györfi et al., 2002). We begin with a mild overlap assumption requiring the group membership probabilities to be bounded away from 0 and 1 given the covariates, similar to the overlap condition commonly assumed in causal inference (Rosenbaum & Rubin, 1983).

**Assumption 3.1.** For every $\ell \in \{1, \ldots, L\}$ there exists $\underline{\pi}_\ell, \overline{\pi}_\ell$ such that $\forall x \in \mathcal{X}$,

$$0 < \underline{\pi}_\ell < \mathbb{P}(Z = \ell | X = x) < \overline{\pi}_\ell < 1.$$

### 3.1. General Result

Our general theoretical analysis proceeds by first establishing a bound for the first-stage estimator $\hat{f}_{\mathcal{A}_n}$ and then leveraging it to provide an upper bound for the final estimator $\hat{g}_\ell$ in (5). Let $\mathcal{F}$ denote a generic function class from which $\hat{f}$

is obtained. Leveraging Theorem 1 of (Padilla et al., 2024a), under mild complexity conditions on $\mathcal{F}$, we obtain

$$\|\bar{f} - \hat{f}_{\mathcal{A}_n}\|_{\mathcal{L}_2}^2 = O_{\mathbb{P}}(r_n),$$

where the rate $r_n$ consists of an approximation error term $\phi_n$, defined via the existence of $\tilde{f} \in \mathcal{F}$ satisfying $\|\bar{f} - \tilde{f}\|_\infty \le \sqrt{\phi_n}$, and an estimation error term determined by the complexity of $\mathcal{F}$ (see Theorem B.1 in the Appendix for a formal statement).

In the second stage, for each group $\ell \in \{1, \ldots, L\}$, we use a separate group-specific nonparametric function class $\mathcal{F}_\ell$ to estimate the offset between overall mean and group mean, which is

$$G_\ell(x) := f_{0,\ell}(x) - \sum_{k=1}^{L} f_{0,k}(x)\, \mathbb{P}(Z = k | X = x). \quad (6)$$

This results in the estimator $\hat{f}_{\ell,\mathcal{B}_n}$ in (4). Building upon the first-stage bound, we now establish bounds for the final group-specific estimator $\hat{g}_\ell$.

**Theorem 3.2.** *Under the conditions of Theorem B.1, and where $\hat{f}$ has been constructed as in (3). Suppose for $\ell \in \{1, \ldots, L\}$ there exists $\widetilde{G}_\ell \in \mathcal{F}_\ell$ such that*

$$\|G_\ell - \widetilde{G}_\ell\|_\infty \le \sqrt{\phi_{\ell,n}},$$

*where $\phi_{\ell,n}$ denotes the approximation error. Let $\eta_{\ell,n} : \mathbb{R}_+ \to \mathbb{R}_+$ for $\ell = 1, \ldots, L$ be functions such that $\forall \delta \in (0,1)$,*

$$\max_{\frac{n\underline{\pi}_\ell}{2} \le n_\ell \le 2n\overline{\pi}_\ell} \sup_{\{x_i\}_{i \in \mathcal{I}_\ell}} \log N(\delta, \mathcal{F}_{\ell,\mathcal{B}_n}, \|\cdot\|_{\mathcal{I}_\ell}) \le \eta_{\ell,n}(\delta), \quad (7)$$

*where $n_\ell = |\mathcal{I}_\ell|$, where $\mathcal{F}_{\ell,\mathcal{B}_n} := \{f_{\mathcal{B}_n}/(2\mathcal{B}_n) : f \in \mathcal{F}_\ell\}$, and $\mathcal{B}_n$ is chosen to be sufficiently large.*

*Suppose the function classes $\mathcal{F}_{\ell,\mathcal{B}_n}$ and $\mathcal{F}_{\mathcal{A}_n}$ satisfy appropriate complexity conditions (see Theorem B.2 for a formal statement), and $\mathbb{P}(\|\epsilon\|_\infty > \mathcal{U}_n) \to 0$ as $n \to \infty$, for some $\mathcal{U}_n > 0$. Then, for $\delta \in (0,1)$,*

$$\|g_{0,\ell} - \hat{g}_\ell\|_{\mathcal{L}_2}^2 = O_{\mathbb{P}}\left( \phi_{\ell,n} + \mathcal{B}_n\delta^2 + \frac{\mathcal{U}_n^2 \eta_{\ell,n}(\delta)}{n\underline{\pi}_\ell} + r_n \right) \quad (8)$$

*provided that $n\underline{\pi}_\ell^3/\overline{\pi}_\ell^2 \to \infty$, $\delta^2 n\underline{\pi}_\ell \to \infty$.*

Theorem 3.2 establishes a general upper bound for two-stage transfer learning, where the convergence rate in (8) decomposes into the first-stage error $r_n$ and the other terms as errors from the second-stage estimation of $G_\ell$, with $\phi_{\ell,n}$ and $\mathcal{B}_n\delta^2 + \mathcal{U}_n^2 \eta_{\ell,n}(\delta)/(n\underline{\pi}_\ell)$ denoting the approximation and estimation errors, respectively. The result holds under $n\underline{\pi}_\ell^3/\overline{\pi}_\ell^2 \to \infty$, a mild condition satisfied as long as the number of groups does not grow too fast with $n$; for example,

the condition $n\underline{\pi}_\ell^3/\overline{\pi}_\ell^2 \to \infty$ reduces to $n\underline{\pi}_\ell \to \infty$ when $\underline{\pi}_\ell \asymp \overline{\pi}_\ell$, which simply means that the expected number of observations in group $\ell$ grows without bound. In addition to complexity conditions on $\mathcal{F}_\ell$ similar in form to those in Theorem B.1, the two-stage estimation procedure requires two additional localized complexity constraints, namely conditions (19) and (20) in the detailed version of Theorem 3.2, imposed on the function classes $\mathcal{F}$ and $\mathcal{F}_\ell$ with respect to group $\ell$. These complexity conditions are mild, and in Section 3.2 we demonstrate that they are satisfied by standard nonparametric function classes, such as deep ReLU neural networks. Moreover, in Section 3.3, we show that sample splitting can further relax these constraints and potentially accelerate the convergence rate.

## 3.2. Theory for Transfer Learning with Deep ReLU Networks

Our proposed general theorems serve as powerful tools for studying the convergence rates of nonparametric estimators within a two-stage transfer learning framework. In this section, we focus on deep ReLU networks as nonparametric estimators in both stages to further investigate the properties of the proposed framework. Throughout, we assume that the overall mean and offset functions satisfy a hierarchical composited structure (Kohler & Langer, 2021).

**Assumption 3.3.** Assume the overall mean $\bar{f}$ and offset $G_\ell$ for $\ell \in \{1, \ldots, L\}$ satisfy the hierarchical compositional structure (see Appendix A for a formal statement), with $\bar{f} \in \mathcal{H}(l_0, \mathcal{P}_0)$ and $G_\ell \in \mathcal{H}(R_\ell, \mathcal{P}_\ell)$. Furthermore, assume that each component function $m$ in the definition of $\bar{f}$ or $G_\ell$ can have different smoothness $p_m = q_m + s_m$, for $q_m \in \mathbb{N}$, $s_m \in (0,1]$, and of potentially different input dimension $K_m$, so that $(p_m, K_m) \in \mathcal{P}_0 \cup (\cup_\ell \mathcal{P}_\ell)$. Let $K_{\max}$ be the largest input dimension and $p_{\max}$ the largest smoothness of any of the functions $m$. Suppose that all the partial derivatives of order less than or equal to $q_m$ are uniformly bounded by constant $c_2$, and each function $m$ is Lipschitz continuous with Lipschitz constant $C_{\mathrm{Lip}} \geq 1$. Also, assume that $\max\{p_{\max}, K_{\max}\} = O(1)$.

With Assumption 3.3 in place, we are ready to present our results. Specifically, the convergence rate for the first-stage estimator $\hat{f}_{\mathcal{A}_n}$, which is estimated by deep ReLU networks class $\mathcal{F}(M, \nu)$ with appropriately chosen depth and width, is established by leveraging Theorem 2 of (Padilla et al., 2024a). Consequently, it holds that $\|\bar{f} - \hat{f}_{\mathcal{A}_n}\|_{\mathcal{L}_2}^2 = O_{\mathbb{P}}(r_n)$, where the rate $r_n$ depends only on

$$\phi_n = \max_{(p,K) \in \mathcal{P}_0} n^{-2p/(2p+K)}, \tag{9}$$

the approximation error of $\bar{f}$ by $\mathcal{F}(M, \nu)$, with $\mathcal{A}_n$ and $\mathcal{U}_n$ chosen appropriately. We refer readers to Theorem B.3 for a formal statement.

Building upon this first-stage result, we present the upper bound for the final transfer-learning estimator $\hat{g}_\ell$ of the group-specific conditional mean, also constructed using deep neural networks.

**Theorem 3.4.** *Suppose that the conditions of Theorem B.3 hold. Let*

$$\phi_{\ell,n} = \max_{(p,K) \in \mathcal{P}_\ell} (n\underline{\pi}_\ell)^{\frac{-2p}{(2p+K)}}. \tag{10}$$

*Then there exists sufficiently large positive constants $c_3$ and $c_4$ such that if $M_\ell$ and $\nu_\ell$ are appropriately chosen (see Theorem B.4 for a formal statement), then $\hat{g}_\ell$ as defined in (5) with $\mathcal{F}_\ell := \mathcal{F}(M_\ell, \nu_\ell)$, satisfies,*

$$\|g_{0,\ell} - \hat{g}_\ell\|_{\mathcal{L}_2}^2 = O_{\mathbb{P}}\Big(\phi_{\ell,n} \max\{\mathcal{U}_n^2, \mathcal{B}_n\} \log^4(\mathcal{B}_n^2 n\overline{\pi}_\ell)$$
$$+ \phi_n \underline{\pi}_\ell^{-1} \max\{\mathcal{A}_n, \mathcal{U}_n^2\} \log^3(n) \log(\mathcal{A}_n^2 n\overline{\pi}_\ell)\Big). \tag{11}$$

*provided that $n\underline{\pi}_\ell^3/\overline{\pi}_\ell^2 \to \infty$, $\mathcal{B}_n$ is chosen to satisfy (18).*

*Remark* 3.5. Suppose that the following quantity

$$\max\{\mathcal{U}_n, \mathcal{A}_n, \mathcal{B}_n, \|\bar{f}\|_\infty, \max_{\ell=1,\ldots,L} \|f_{0,\ell}\|_\infty, \|G_\ell\|_\infty\}, \tag{12}$$

grows at most in $O(\mathrm{poly}(\log n))$. This is a general setting that includes the case where the error distribution can be sub-Gaussian or even sub-exponential. Let us now compare the result in Theorem 3.4 (based on pretraining) with the alternative approach of estimating each $g_{0,\ell}$ in (2) separately:

$$\hat{h}_\ell := \arg\min_{g \in \mathcal{F}_\ell} \sum_{i:z_i=\ell} \big(y_i - g(x_i)\big)^2. \tag{13}$$

The convergence rate of $\hat{h}_\ell$ depends on the approximation error of the class $\mathcal{F}_\ell$ relative to $g_{0,\ell}(x) = f_0(x) + f_{0,\ell}(x)$ as well as the estimation error (e.g., for ReLU neural networks) based on approximately $n\underline{\pi}_\ell$ observations, assuming $\underline{\pi}_\ell \asymp \overline{\pi}_\ell$. In contrast, under Theorem 3.4—ignoring logarithmic factors and assuming $\underline{\pi}_\ell \asymp 1$—the rate for our estimator $\hat{g}_\ell$ becomes

$$\phi_n + \phi_{\ell,n},$$

where $\phi_n$, as defined in (9), is the rate for estimating $\bar{f}$ based on $n$ samples, and $\phi_{\ell,n}$ is the convergence rate for estimating $G_\ell(x)$ based on approximately $n\underline{\pi}_\ell \asymp n$ samples.

Thus, when the functions $\bar{f}$ and $G_\ell$ are less complex than $g_{0,\ell}$, pretraining can yield better performance than estimating each $g_{0,\ell}$ separately via $\hat{h}_\ell$. This scenario is reasonable: $\bar{f}$ involves a weighted average of the difference functions $f_{0,k}$, which can mitigate the influence of extreme values associated with a particular $f_{0,k}$, while $G_\ell$ may be small or close to zero in regions where the functions $f_{0,k}$ take similar values.

As another example, consider the case where

$$\frac{\phi_n}{\phi_{\ell,n}} \lesssim \underline{\pi}_\ell,$$

for some fixed $\ell \in \{1, \ldots, L\}$, which can allow for $\overline{\pi}_\ell \to 0$. Then the convergence rate of our estimator depends only on estimating the function $G_\ell(x)$ using a neural network class and a sample size of the same order as that required for estimating $g_{0,\ell}$ with a neural network class. In this setting, the comparison between our pretraining-based estimator $\hat{g}_\ell$ and the naive estimator $\hat{h}_\ell$ reduces to determining which function is easier to estimate with the same number of samples using neural networks. When $g_{0,\ell}$ is not substantially different from the functions $\{g_{0,k}\}$, the function $G_\ell$ is typically easier to estimate than $g_{0,\ell}$, as in problems with shared information across groups (see 6)—precisely the type of setting in which pretraining is most beneficial.

*Remark* 3.6. Our upper bounds demonstrate two key points: (1) the convergence rate of transfer learning with deep ReLU networks can overcome the curse of dimensionality, and (2) the benefits of transfer learning. However, we do not claim optimality. We emphasize that this is not a limitation of our work. In fact, lower bounds for the hierarchical function class are only known in very specific cases (see Remark 2 in (Kohler & Langer, 2021)).

### 3.3. Independent Two Stages Estimator

In this subsection, we assume that the estimator $\hat{f}$ in (3) is computed using data independent of that in (4), e.g., via sample splitting.

**Theorem 3.7.** *Consider the conditions and notations from Theorem 3.2. However, instead of assuming that (20) holds, suppose that $\hat{f}$ is computed with data independent of that used in (4). Then $\|g_{0,\ell} - \hat{g}_\ell\|^2_{\mathcal{L}_2}$ is of order*

$$O_{\mathbb{P}}\left( \phi_{\ell,n} + r_n + \mathcal{A}_n^2 \sqrt{\frac{\log n}{n\underline{\pi}_\ell}} + \frac{\mathcal{U}_n^2 \eta_{\ell,n}(\delta)}{n\underline{\pi}_\ell} + \mathcal{B}_n \delta^2 \right).$$

It is important to note that if $\hat{f}$ is computed using data independent of that used in (4), the conclusion of Theorem 3.2 continues to hold under its original assumptions. The main difference between the upper bound in Theorem 3.7 and that in Theorem 3.2 is the relaxation of the complexity constraints (specifically, the replacement of condition (20)) with an additional term $\mathcal{A}_n^2 \sqrt{\log n/(n\underline{\pi}_\ell)}$. The potential advantage of Theorem 3.7 is that the additional term may yield a faster convergence rate than that in Theorem 3.2. We illustrate this next.

**Corollary 3.8.** *Consider the conditions of Theorems B.3 and 3.4, with the only modification that $\hat{f}$ is independent of the data used in (4). Let $(p^\ell, K^\ell)$ be given as*

$$(p^\ell, K^\ell) = \underset{(p,K)\in\mathcal{P}_\ell}{\arg\max} \frac{-2p}{2p + K}.$$

*Then, if $K^\ell > 2p^\ell$, the ReLU neural network pretraining*

*estimator $\hat{g}_\ell$ satisfies*

$$\|g_{0,\ell} - \hat{g}_\ell\|^2_{\mathcal{L}_2} = O_{\mathbb{P}}\Big( \phi_{\ell,n} \max\{\mathcal{U}_n^2, \mathcal{B}_n, \mathcal{A}_n^2\} \log^4(\mathcal{B}_n^2 n\overline{\pi}_\ell)$$
$$+ \phi_n \log^3(n) \log(\mathcal{A}_n) \max\{\mathcal{A}_n, \mathcal{U}_n^2\} \Big).$$

*In contrast, when $K^\ell \leq 2p^\ell$, we obtain that $\hat{g}_\ell$ satisfies (11).*

*Remark* 3.9. Corollary 3.8 shows that in the case $K^\ell > 2p^\ell$—which corresponds to a setting where the dimensionality is sufficiently large relative to the smoothness driving the quantity $\phi_{\ell,n}$—and under the condition (12), the ReLU pretraining estimator (ignoring logarithmic factors) attains the rate

$$\phi_n + \phi_{\ell,n}.$$

This rate naturally decomposes into two components: the rate for estimating $\bar{f}$ based on $n$ samples, plus the rate for estimating $G_\ell$ based on $n\underline{\pi}_\ell$ samples. The key difference from Theorem 3.4 is that here we are able to remove the factor $\underline{\pi}_\ell$ from the denominator on the right-hand side of (11). This distinction matters primarily when $\underline{\pi}_\ell \to 0$. If the latter holds, the first stage estimation is done with significantly more samples than if the estimation was done only using the data from the $\ell$th group ($n$ vs $n\underline{\pi}_\ell$), while the second stage estimation is done with the same number of samples (roughly $n\underline{\pi}_\ell$) though potentially for estimating a simpler function.

We conjecture that the appearance of $\underline{\pi}_\ell$ in Theorem 3.4 is an artifact of the proof. Indeed, in practice we observe that the estimator performs better without sample splitting—that is, in the setting considered in Theorem 3.4—than with sample splitting, as in Corollary 3.8. That said, both results coincide and yield the same rate when $\underline{\pi}_\ell \asymp 1$.

*Remark* 3.10. We illustrate the benefit of transfer learning with DNNs in comparison with classical estimators whose rates depend on the ambient dimension and therefore suffer from the curse of dimensionality, such as transfer learning by kernel smoothing (Du et al., 2017), through a simple example applying Corollary 3.8. Let $X \in [0,1]^{10}$ and $\ell \in \{1,2,3\}$ index three groups. For simplicity, suppose that $\overline{\pi}_\ell = \underline{\pi}_\ell = \pi_\ell$ for all $\ell$. Consider $g_{0,\ell}(x) = f_0(x) + f_{0,\ell}(x)$ with $f_0(x) = |x_1 + \cdots + x_{10} - 1/2|$ and $f_{0,\ell}(x) = \ell \cdot |x_1 + x_2 + x_3 + x_4 - 1/2|^{3/2}$. Then $g_{0,\ell}$ and $\bar{f}$ has smoothness $p = 1$ on the full $K = 10$ ambient space, while the Stage-2 offset $G_1(x) = (1 - \pi_1 - 2\pi_2 - 3\pi_3)|x_1 + x_2 + x_3 + x_4 - 1/2|^{3/2}$ for group 1 is much simpler, with smoothness $p = 3/2$ and intrinsic dimension $K = 4$.[1] Table 1 provides the convergence rates of four estimators across regimes. As $\pi_1$ shrinks, classical TL estimator degrades to $(n\pi_1)^{-3/13}$, while NN with TL

---

[1]We note that the same function may admit different hierarchical compositions, and the values $(p, K)$ we chosen here are for illustrative purposes. In fact, our upper bounds apply at the function-class level rather than to a specific function instance.

remains at $n^{-1/6}$ until $\pi_1 \ll n^{-11/18}$ and then transitions to $(n\pi_1)^{-3/7}$, strictly faster than classical TL. This example clearly illustrates how a small $\pi_\ell$ can drastically worsen the rate of transfer learning using classical estimators while NN with TL remains robust. We refer readers to Appendix E for a detailed explanation of this example and other concrete examples illustrating the rate $\phi_n + \phi_{\ell,n}$ using DNNs.

*Table 1.* Convergence rates of four estimators under different regimes of $\pi_1$. CLS and NN stand for classical estimators and deep ReLU networks, respectively.

| Regime of $\pi_1$ | CLS-no-TL | NN-no-TL | CLS-TL | NN-TL |
|---|---|---|---|---|
| $\pi_1 \asymp 1$ | $n^{-\frac{1}{6}}$ | $n^{-\frac{1}{6}}$ | $n^{-\frac{1}{6}}$ | $n^{-\frac{1}{6}}$ |
| $n^{-\frac{5}{18}} \ll \pi_1 \ll 1$ | $(n\pi_1)^{-\frac{1}{6}}$ | $(n\pi_1)^{-\frac{1}{6}}$ | $n^{-\frac{1}{6}}$ | $n^{-\frac{1}{6}}$ |
| $n^{-\frac{11}{18}} \lesssim \pi_1 \lesssim n^{-\frac{5}{18}}$ | $(n\pi_1)^{-\frac{1}{6}}$ | $(n\pi_1)^{-\frac{1}{6}}$ | $(n\pi_1)^{-\frac{3}{13}}$ | $n^{-\frac{1}{6}}$ |
| $n^{-1} \ll \pi_1 \ll n^{-\frac{11}{18}}$ | $(n\pi_1)^{-\frac{1}{6}}$ | $(n\pi_1)^{-\frac{1}{6}}$ | $(n\pi_1)^{-\frac{3}{13}}$ | $(n\pi_1)^{-\frac{3}{7}}$ |

## 4. Experiments

### 4.1. Numerical Experiments

We conduct experiments in diverse simulation settings to evaluate our proposed transfer learning strategy (denoted as **2-Stage**). For deep ReLU networks (NN), we compare 2-Stage with several training strategies, including **Pooled**, **Separate**, **Pool-w-L**, and **Top-FT**. Specifically, the Pooled strategy trains a single model $\hat{f}$ on all data while ignoring group labels, following (3), and also serves as the first stage of our procedure. We further consider Separate, which trains independent models for each group; Pool-w-L, which pools the data and appends group labels as additional inputs to train a single model; and Top-FT, which fine-tunes only the output layer of the pooled NN for each group. The method of (Jiao et al., 2024), conceptually similar to Top-FT, was also considered but omitted as it did not show advantages in most of our simulations. Besides NN, we compare with Random Forest (denoted as **RF**, (Breiman, 2001)), implementing the same strategies except for Top-FT. The **ptLasso** (Craig et al., 2026) is also considered in the high-dimensional setting as a parametric competitor.

We generate $n \in \{5000, 10000, 30000, 50000\}$ independent samples for each scenario. Gaussian noise is used in all scenarios. In the low-dimensional settings, we control the noise level via a signal-to-noise ratio (SNR), defined as the ratio between the empirical variance of the noiseless signal computed within each generated dataset and $\text{Var}(\epsilon)$, with $\text{SNR} \in \{2, 5, 10\}$. Each sample is assigned to one of $L = 5$ groups, where group sizes are unbalanced and proportional to the group index $z \in \{1, \ldots, 5\}$. In the high-dimensional scenarios, the noise variance is set to $\text{Var}(\epsilon) \in \{0.1^2, 1\}$. To mimic settings with a large number of different groups, each sample is assigned to one of $L = 30$ groups with equal group sizes. For each generated dataset, 15% of the samples

within each group are randomly held out as a test set. All experiments are repeated over 50 Monte Carlo replications, and performance is evaluated using the mean squared error (MSE) on the test set. Due to space limitations, we present two low-dimensional scenarios with SNR = 5 and two high-dimensional scenario with $\text{Var}(\epsilon) = 1$, and refer readers to Appendix F for other simulation scenarios and detailed experimental configurations.

**Scenario 1.** We consider a low-dimensional setting with an additive structure. Let the input vector be $q = (q_1, \ldots, q_{10}) \overset{\text{i.i.d.}}{\sim} \text{Unif}([0,1]^{10})$, holding for Scenario 1 and 2. We define four intermediate quantities:

$$h_1(q) = \sum_{i=1}^{10} q_i^2, \qquad h_2(q) = \sum_{i=1}^{10} |q_i|,$$

$$h_3(q; z) = \sum_{i=1}^{5} q_i^2 - q_z^2, \quad h_4(q; z) = \sum_{i=1}^{5} |q_i| - |q_z|.$$

The shared and group-specific components are given by:

$$f_0(q) = \log\left(1 + h_1(q) \cdot h_2(q)\right),$$
$$f_{0,z}(q) = \sqrt{1 + |h_3(q; z) + h_4(q; z)|},$$

and the response is generated according to the additive model $y = f_0(x) + f_{0,z}(x) + \epsilon$.

**Scenario 2.** We consider a low-dimensional setting with a nonlinear mixture of shared and group-specific components, without an explicit additive structure. Based on the four intermediate quantities defined in Scenario 1, we further define the following compositions:

$$h_5(q; z) = \sqrt{h_1(q)\, h_2(q) + h_3(q; z)\, h_4(q; z)},$$
$$h_6(q; z) = (h_1(q) + h_3(q; z))^2 / (h_2(q) + h_4(q; z))^2.$$

The group-specific conditional mean is then given by

$$f_z(x) = h_5(q; z) \cdot h_6(q; z),$$

and the response is generated with $y = f_z(x) + \epsilon$.

**Scenario 3.** We consider a high-dimensional setting with shifted latent representations generated from a low-dimensional latent factor model. Let the latent variable be $u = (u_1, \ldots, u_{10}) \sim \mathcal{N}(0, I_{10})$. The response is generated directly from the latent variable as $y = \sum_{j=1}^{10} u_j^2 + \epsilon$. However, the latent variable $u$ is unobserved; instead, for each group $z \in \{1, \ldots, L\}$, the observed covariates $q \in \mathbb{R}^{100}$ are obtained via a group-specific nonlinear transformation of $u$. Specifically, each group is randomly assigned parameters $w_z \in \{1/2, 1/3, \ldots, 1/10\}$,

$\nu_z \in \{1/5, 1/4, 1/3, 1/2, 1, 2, 3, 4, 5\}$, and $\psi_z \in \{2\pi/k : k = 1, \ldots, 6\}$. We define a shifted latent vector $\tilde{u}^{(z)}$:

$$\tilde{u}_j^{(z)} = \begin{cases} u_j, & \text{if } j \in \{1, \ldots, 5\}, \\ u_j + w_z \cdot \sin(\psi_z + \nu_z \cdot u_j), & \text{if } j \in \{6, \ldots, 10\}. \end{cases}$$

The observed covariates are then generated from $\tilde{u}^{(z)}$ via a two-layer neural network mapping $T : \mathbb{R}^{10} \to \mathbb{R}^{100}$, defined as $q = T(\tilde{u}^{(z)}) = V \tanh(W\tilde{u}^{(z)} + b)$, where $W \in \mathbb{R}^{50 \times 10}$, $b \in \mathbb{R}^{50}$, and $V \in \mathbb{R}^{100 \times 50}$ are fixed across all groups and randomly initialized as $W_{ij} \overset{\text{i.i.d.}}{\sim} \mathcal{N}(0, 10^{-1})$, $b_j \overset{\text{i.i.d.}}{\sim} \mathcal{N}(0, 0.1^2)$, and $V_{ij} \overset{\text{i.i.d.}}{\sim} \mathcal{N}(0, 50^{-1})$. Under this construction, the estimation target $f_z(q) = \mathbb{E}[y \mid q, z]$ is a highly nonlinear and implicit function of the observed $q$.

**Scenario 4.** This scenario features a shared low-dimensional latent structure with sparse group-specific signals. Let $u = (u_1, \ldots, u_{10}) \sim \mathcal{N}(0, I_{10})$ be the shared latent variable. The observed covariates $q \in \mathbb{R}^{500}$ consist of two parts: the first 410 coordinates arise from a shared nonlinear transformation $q_{\text{share}} = V \tanh(Wu + b)$, where $W \in \mathbb{R}^{50 \times 10}$, $b \in \mathbb{R}^{50}$, $V \in \mathbb{R}^{410 \times 50}$ are randomly initialized as $W_{ij} \overset{\text{i.i.d.}}{\sim} \mathcal{N}(0, 10^{-1})$, $b_j \overset{\text{i.i.d.}}{\sim} \mathcal{N}(0, 0.1^2)$, $V_{ij} \overset{\text{i.i.d.}}{\sim} \mathcal{N}(0, 50^{-1})$; the remaining 90 coordinates $q_{\text{tail}} \sim \mathcal{N}(0, I_{90})$ form group-specific sparse signals. Each group $z$ is assigned three disjoint indices from $q_{\text{tail}}$, each associated with a randomly selected function from $\{\sin(\nu_z \cdot), \tanh(\nu_z \cdot), (1 + \exp(\nu_z \cdot))^{-1}\}$ with a randomly chosen parameter $\nu_z \in \{1/5, 1/4, 1/3, 1/2, 1, 2, 3, 4, 5\}$. The response is $y = \sum_{j=1}^{10} u_j^2 + \sum_{k=1}^{3} \theta_{z,k}(q_{s_{z,k}}) + \epsilon$, where $\theta_{z,k}$ denotes the function for group $z$ at index $s_{z,k}$.

Based on the numerical simulations, the deep neural network estimator generally outperforms the other considered estimators, demonstrating the advantages of NN as a nonparametric estimator in handling complex functional composite structures across various dimensions. Notably, the NN trained with the proposed two-stage transfer learning strategy consistently outperforms both alternative training strategies and other estimators, highlighting the advantage of the proposed strategy. Moreover, in low-dimensional settings within RF–based estimators, the two-stage transfer learning strategy achieves the lowest MSE, demonstrating the generalizability of our framework to other estimators.

### 4.2. Real Data Experiments

We conduct two real-data experiments: a low-dimensional study based on the Beijing PM2.5 dataset (Chen, 2015) and a high-dimensional study based on the UTKFace dataset (Zhang et al., 2017). In this section, we present partial results for UTKFace and refer readers to Appendix G for the complete results across all experiments.

**Experiments on the UTKFace Dataset.** We conduct real-

*Table 2.* Average MSE for Scenario 1 and 2 with SNR = 5 over 50 independent trials. For each sample size, the lowest MSE is highlighted in bold.

| Model / Size | 5000 | 10000 | 30000 | 50000 |
|---|---|---|---|---|
| | Scenario 1 (MSE $\times 10^2$) | | | |
| Pooled (NN) | 2.28 (0.2) | 2.13 (0.2) | 2.34 (0.7) | 1.93 (0.3) |
| 2-Stage (NN) | **0.93 (0.2)** | **0.64 (0.1)** | **0.34 (0.1)** | **0.27 (0.1)** |
| Separate (NN) | 6.64 (1.2) | 0.97 (0.4) | 0.60 (0.1) | 0.49 (0.1) |
| Top-FT (NN) | 2.12 (0.2) | 1.77 (0.2) | 1.14 (0.2) | 0.92 (0.2) |
| Pool-w-L (NN) | 1.03 (0.2) | 0.74 (0.2) | 1.03 (1.0) | 0.48 (0.2) |
| Pooled (RF) | 7.08 (0.4) | 6.73 (0.3) | 6.55 (0.2) | 6.69 (0.1) |
| 2-Stage (RF) | 4.10 (0.3) | 3.37 (0.2) | 2.52 (0.1) | 2.36 (0.1) |
| Separate (RF) | 7.75 (0.5) | 6.50 (0.3) | 5.32 (0.2) | 5.00 (0.1) |
| Pool-w-L (RF) | 6.92 (0.4) | 6.56 (0.3) | 6.37 (0.2) | 6.52 (0.1) |
| | Scenario 2 (MSE $\times 10^2$) | | | |
| Pooled (NN) | 3.97 (0.5) | 3.51 (0.3) | 3.67 (1.1) | 2.77 (0.2) |
| 2-Stage (NN) | **2.75 (0.3)** | **2.09 (0.2)** | **1.44 (0.1)** | **1.09 (0.1)** |
| Separate (NN) | 8.84 (1.4) | 3.75 (0.8) | 2.00 (0.1) | 1.75 (0.2) |
| Top-FT (NN) | 3.52 (0.3) | 2.92 (0.3) | 1.98 (0.2) | 1.64 (0.2) |
| Pool-w-L (NN) | 3.52 (0.7) | 2.73 (0.5) | 2.53 (1.2) | 1.51 (0.2) |
| Pooled (RF) | 12.37 (1.0) | 11.43 (0.6) | 10.87 (0.3) | 11.03 (0.3) |
| 2-Stage (RF) | 7.96 (0.8) | 6.58 (0.5) | 5.12 (0.2) | 4.76 (0.2) |
| Separate (RF) | 15.70 (1.3) | 13.12 (0.7) | 10.31 (0.3) | 9.57 (0.2) |
| Pool-w-L (RF) | 12.26 (1.0) | 11.33 (0.6) | 10.78 (0.3) | 10.97 (0.3) |

data experiments to image data for facial age estimation, where the goal is to estimate age conditioned on a facial image input. In this task, shared visual attributes such as skin texture and wrinkles provide universal cues for age estimation, while facial morphology varies across different ethnic groups, making it a natural setting for transfer learning.

Specifically, experiments are conducted on the UTKFace dataset (Zhang et al., 2017), which contains 23,705 facial images annotated with ages ranging from 0 to 116, along with ethnicity labels across five groups: White (43%), Black (19%), Asian (14%), Indian (17%), and Other (7%). We treat ethnicity as the group label and perform neural network–based transfer learning across groups. Following (Baumann et al., 2021), we adopt a more practical approach in which rather than learning directly from raw pixels, we first extract 512-dimensional latent features for each image using a pretrained FaceNet model (Schroff et al., 2015), with weights obtained from an open-source implementation trained on the VGGFace2 dataset (Cao et al., 2018). These features then serve as input covariates for both our proposed framework and competing methods.

We evaluate the five learning strategies from the previous section, focusing exclusively on DNN estimators. For data partitioning, we randomly split the data while maintaining the ethnic group proportions across training (70%), validation (15%), and test (15%) sets. Each neural network instance is a four-layer MLP with 64 hidden units per layer and ReLU activations, trained with early stopping on the validation set. Additional configurations are provided in Sec-

*Table 3.* Average MSE for Scenario 3 and 4 with $\mathrm{Var}(\epsilon) = 1$ over 50 independent trials. Random forest–based estimators are omitted due to high computational cost and poor performance. For each sample size, the lowest MSE is highlighted in bold.

| Model / Size | 5000 | 10000 | 30000 | 50000 |
|---|---|---|---|---|
| | Scenario 3 (MSE) | | | |
| Pooled (NN) | 4.08 (0.4) | 3.22 (0.4) | 2.63 (0.4) | 2.12 (0.3) |
| 2-Stage (NN) | **3.95 (0.4)** | **2.97 (0.3)** | **2.12 (0.2)** | **1.77 (0.2)** |
| Separate (NN) | 18.52 (1.4) | 16.84 (1.1) | 8.01 (0.6) | 5.70 (0.4) |
| Top-FT (NN) | 3.97 (0.4) | 3.05 (0.3) | 2.26 (0.3) | 1.87 (0.2) |
| Pool-w-L (NN) | 4.37 (0.5) | 3.19 (0.3) | 2.34 (0.3) | 1.85 (0.2) |
| ptLasso | 20.14 (1.4) | 19.14 (1.2) | 18.76 (0.8) | 18.51 (0.7) |
| | Scenario 4 (MSE) | | | |
| Pooled (NN) | 9.13 (0.9) | 6.46 (0.6) | 4.40 (0.5) | 3.56 (0.3) |
| 2-Stage (NN) | **9.08 (0.9)** | **6.33 (0.6)** | **3.89 (0.3)** | **3.19 (0.2)** |
| Separate (NN) | 22.33 (1.5) | 21.59 (0.9) | 15.16 (0.6) | 13.50 (0.7) |
| Top-FT (NN) | 9.12 (0.9) | 6.40 (0.6) | 4.11 (0.3) | 3.44 (0.3) |
| Pool-w-L (NN) | 9.21 (0.9) | 6.47 (0.6) | 4.59 (0.7) | 3.51 (0.3) |
| ptLasso | 21.76 (1.7) | 19.75 (1.0) | 19.31 (0.8) | 19.20 (0.8) |

*Table 4.* Overall and group-wise average MSE of age regression models over 20 independent trials. Random forest–based estimators are omitted due to poor performance. The best performance within each group is bolded.

| Model | Overall | White | Black | Asian | Indian |
|---|---|---|---|---|---|
| Pooled | 61.8 (1.7) | 72.2 (2.7) | 60.1 (3.8) | 47.9 (7.8) | 49.2 (2.9) |
| 2-Stage | **59.6 (1.9)** | **68.9 (3.2)** | **59.1 (4.1)** | **46.3 (8.4)** | 48.3 (2.7) |
| Separate | 63.1 (2.5) | 72.4 (3.8) | 65.4 (6.0) | 48.2 (8.6) | 49.6 (3.6) |
| Top-FT | 61.4 (1.7) | 71.8 (2.5) | 59.8 (3.7) | 47.8 (7.9) | 48.7 (3.0) |
| Pool-w-L | 60.7 (2.5) | 71.1 (3.9) | 59.5 (3.4) | 46.7 (8.4) | **47.5 (2.7)** |

tion G.2. Table 4 reports the average MSE over 20 random trials on the test set, excluding samples labeled as "Other" ethnicity due to their ambiguous demographic attribution.

From Table 4, the three hybrid strategies combining common and group-specific information consistently outperform the simple Pooled and Separated baselines, demonstrating the value of both leveraging shared patterns across groups and accommodating differences among groups in age estimation. Among all strategies, the proposed two-stage transfer learning achieves the lowest overall MSE, attaining optimal performance on White, Black, and Asian subgroups while yielding substantial improvements on the Indian group relative to Pooled. These results validate the effectiveness of our proposed method. For additional results, including analyses that incorporate gender as an additional grouping factor and experiments using raw image inputs trained directly with MLPs, we refer readers to Appendix G.3.

## 5. Conclusion

This paper investigates transfer learning for nonparametric regression, using fully connected deep neural networks with ReLU activation as the estimator of interest. We assume that each group-specific conditional mean consists of a shared component common across all groups and a group-specific deviation. Our approach employs a two-stage offset learning framework: in the first stage, the overall conditional mean is estimated using pooled data from all groups to capture the shared structure; in the second stage, a group-specific offset is estimated to learn group-level characteristics, which are then combined to form the final estimator. We further discuss scenarios where transfer learning with deep neural networks can be beneficial, including cases where averaging similar groups yields a simpler overall function, where the offset function is simpler than the group-specific conditional mean, and data augmentation with all groups' data improves the estimation of the overall conditional mean.

Several directions remain open for future work. First, our framework formally assumes an additive decomposition; while our simulations do not require this explicit form to hold, extending this to other coupling forms remains interesting. Second, negative transfer is theoretically possible in our setting when the offset $G_\ell$ is highly complex or when the first-stage aggregation produces a complicated $\bar{f}$. Such a case may arise when the groups are substantially different; in practice, however, groups in many real-world datasets exhibit strong cross-group similarity (e.g., our real-data experiments), and formal detection of negative transfer is left as future work. Third, our two-stage offset framework is conceptually related to the practically popular last-layer fine-tuning strategy, and developing analogous theory for fine-tuning is an open and important topic. Beyond these, aligning our theory with current practice motivates extensions to more advanced deep architectures, such as convolutional neural networks (LeCun et al., 2002; Krizhevsky et al., 2012) and Transformers (Vaswani et al., 2017). Finally, while this work assumes discrete group differences, a natural extension would be to model continuous variation across smoothly related tasks.

## Acknowledgements

We greatly thank the anonymous reviewers for their helpful comments and suggestions, which have greatly helped us improve the quality of this work.

## Impact Statement

This paper studies transfer learning for nonparametric regression with data from multiple groups and proposes a general two-stage framework with theoretical guarantees. While the proposed methods may have potential applications in areas such as scientific data analysis and other domains involving grouped data, the primary contributions of this work are focuses on improving statistical efficiency and convergence properties. As such, the societal impact of this work is ex-

pected to be indirect, mainly by advancing the theoretical understanding of transfer learning and nonparametric regression with deep neural networks, and by informing future research in related areas.

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

## A. Formal Definitions of Hierarchical Composition Models

**Definition A.1** $((p, C)$-Smoothness$)$**.** Let $p = q + s$ for some $q \in \mathbb{N} = \mathbb{Z}^+ \cup \{0\}$ and $0 < s \leq 1$. We say that a function $g : \mathbb{R}^d \to \mathbb{R}$ is called $(p, C)$-smooth, if for every $\alpha = (\alpha_1, \dots, \alpha_d) \in \mathbb{N}^d$, with $d \in \mathbb{Z}^+$, where $\sum_{j=1}^{d} \alpha_j = q$ the partial derivative $\partial^q g / (\partial a_1^{\alpha_1} \dots \partial a_d^{\alpha_d})$ exists and satisfies

$$\left| \frac{\partial^q g}{\partial a_1^{\alpha_1} \dots \partial a_d^{\alpha_d}} (a) - \frac{\partial^q g}{\partial a_1^{\alpha_1} \dots \partial a_d^{\alpha_d}} (b) \right| \leq C \|a - b\|^s$$

for all $a, b \in \mathbb{R}^d$.

Building on the definition of $(p, C)$-smoothness, we now introduce the class of hierarchical composition models.

**Definition A.2** (Space of Hierarchical Composition Models, (Kohler & Langer, 2021))**.** For $l = 1$ and smoothness constraint $\mathcal{P} \subseteq (0, \infty) \times \mathbb{N}$ the space of hierarchical composition models is given as

$$\mathcal{H}(1, \mathcal{P}) \quad := \quad \left\{ h : \mathbb{R}^d \to \mathbb{R} : h(a) = m\left(a_{(\pi(1))}, \dots, a_{(\pi(K))}\right), \text{ where } m : \mathbb{R}^K \to \mathbb{R} \text{ is} \right.$$
$$\left. (p, C)\text{-smooth for some } (p, K) \in \mathcal{P} \text{ and } \pi : \{1, \dots, K\} \to \{1, \dots, d\}\right\}.$$

For $l > 1$, we recursively construct

$$\mathcal{H}(l, \mathcal{P}) \quad := \quad \left\{ h : \mathbb{R}^d \to \mathbb{R} : h(a) = m\left(f_1(a), \dots, f_K(a)\right), \text{ where } m : \mathbb{R}^K \to \mathbb{R} \text{ is} \right.$$
$$\left. (p, C)\text{-smooth for some } (p, K) \in \mathcal{P} \text{ and } f_i \in \mathcal{H}(l - 1, \mathcal{P})\right\}.$$

Figure 1 gives an illustration of a hierarchical composition model from the class $\mathcal{H}(2, \mathcal{P})$, where a complex multivariate function is constructed via a two-level composition of low-dimensional smooth functions acting on subsets of the 10-dimensional input variables.

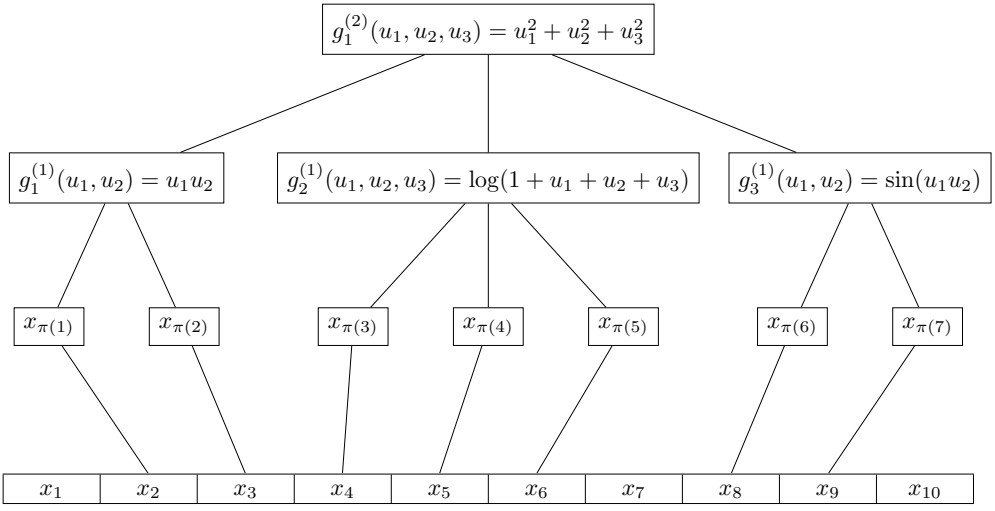

*Figure 1.* Illustration of a hierarchical composition model of the class $\mathcal{H}(2, \mathcal{P})$.

## B. Formal Statements of Theorems

**Theorem B.1.** *[General Error Bound for First-Stage Estimation (Theorem 1 from (Padilla et al., 2024a)])]. Let $\mathcal{U}_n > 0$ and suppose that $\tilde{f} \in \mathcal{F}$ is such that*

$$\|\bar{f} - \tilde{f}\|_\infty \leq \sqrt{\phi_n},$$

*so that $\phi_n$ is the approximating error. Suppose that $\mathcal{A}_n$ is chosen to satisfy*

$$\mathcal{A}_n \geq 8 \max\{\|\bar{f}\|_\infty + \mathcal{U}_n, 8\|\bar{f}\|_\infty, 8\sqrt{\phi_n}\}.$$

*Moreover, let $\mathcal{F}_{\mathcal{A}_n} := \{f_{\mathcal{A}_n}/(2\mathcal{A}_n) : f \in \mathcal{F}\}$ and assume that*

$$\sup_{x_1,\ldots,x_n \in [0,1]^d} \log N\left(\delta, \mathcal{F}_{\mathcal{A}_n}, \|\cdot\|_n\right) \leq \eta_n(\delta)$$

*for some decreasing function $\eta_n : (0,1) \to \mathbb{R}_{\geq 0}$. If*

$$\lim_{n\to\infty} \left[ \sum_{k=0}^{\infty}\sum_{k'=1}^{\infty} \exp\left(-C_1\eta_n(2^{-k-k'-1})\right) + \sum_{k=0}^{\infty} \exp\left(-C_2\eta_n(2^{-k-1})\right) + \mathbb{P}(\|\bar{\epsilon}\|_\infty > \mathcal{U}_n) \right] = 0, \tag{14}$$

*for some constants $C_1, C_2 > 0$ and*

$$\sup_{k\in\mathbb{N}} \sum_{k'=1}^{\infty} \frac{\eta_n(2^{-k-k'})}{2^{2k'}\eta_n(2^{-k})} \leq 1, \tag{15}$$

*then*

$$\max\{\|\bar{f} - \hat{f}_{\mathcal{A}_n}\|_n^2, \|\bar{f} - \hat{f}_{\mathcal{A}_n}\|_{\mathcal{L}_2}^2\} = O_{\mathbb{P}}\left(\phi_n + \frac{\mathcal{U}_n^2\eta_n(\delta_n)}{n} + \mathcal{A}_n\delta_n^2\right), \tag{16}$$

*where $\delta_n$ is a critical radius of $\mathcal{F}_{\mathcal{A}_n}$, see Definition C.1.*

**Theorem B.2.** *[Detailed Statement of Theorem 3.2]. With the notation and conditions from Theorem B.1, let $r_n$ be the rate of convergence of $\hat{f}_{\mathcal{A}_n}$ towards $\bar{f}$, namely*

$$r_n := \phi_n + \frac{\mathcal{U}_n^2\eta_n(\delta_n)}{n} + \mathcal{A}_n\delta_n^2,$$

*and where $\hat{f}$ has been constructed as in (3). Suppose for $\ell \in \{1,\ldots,L\}$ there exists $\widetilde{G}_\ell \in \mathcal{F}_\ell$ such that*

$$\|G_\ell - \widetilde{G}_\ell\|_\infty \leq \sqrt{\phi_{\ell,n}},$$

*where $\phi_{\ell,n}$ denotes the approximation error. Let $\eta_{\ell,n} : \mathbb{R}_+ \to \mathbb{R}_+$, be functions such that*

$$\max_{\frac{n\underline{\pi}_\ell}{2} \leq n_\ell \leq 2n\overline{\pi}_\ell} \sup_{\{x_i\}_{i\in\mathcal{I}_\ell}\subset[0,1]^d} \log N(\delta, \mathcal{F}_{\ell,\mathcal{B}_n}, \|\cdot\|_{\mathcal{I}_\ell}) \leq \eta_{\ell,n}(\delta) \ \forall\delta\in(0,1), \tag{17}$$

*where $n_\ell = |\mathcal{I}_\ell|$, where $\mathcal{F}_{\ell,\mathcal{B}_n} := \{f_{\mathcal{B}_n}/(2\mathcal{B}_n) : f \in \mathcal{F}_\ell\}$, and $\mathcal{B}_n$ is chosen such that*

$$\mathcal{B}_n \geq \max\{\mathcal{U}_n + \|\bar{f}\|_\infty + \mathcal{A}_n + 2\max_{\ell=1,\ldots,L}\|f_{0,\ell}\|_\infty, \|G_\ell\|_\infty + \sqrt{\phi_{\ell,n}}\}. \tag{18}$$

*Suppose that $\delta \in (0,1)$ satisfies*

$$\max_{m:\frac{n\underline{\pi}_\ell}{2}\leq m\leq 2n\overline{\pi}_\ell} \sup_{\{x_j\}_{j=1}^m\subset[0,1]^d} \left\{ \frac{1}{\sqrt{m}}\int_{\delta^2/48}^{\delta} \sqrt{\log N\left(t/4, \mathcal{F}_{\ell,\mathcal{B}_n}, \|\cdot\|_m\right)}dt + \frac{\delta}{\sqrt{m}}\sqrt{c_1\log(48/\delta^2)} \right\} \lesssim \delta^2, \tag{19}$$

*and*

$$\max_{m:\frac{n\underline{\pi}_\ell}{2}\leq m\leq 2n\overline{\pi}_\ell} \sup_{\{x_j\}_{j=1}^m\subset[0,1]^d} \left\{ \frac{1}{\sqrt{m}}\int_{\delta^2/48}^{\delta} \sqrt{\log N\left(t/4, \mathcal{F}_{\mathcal{A}_n}, \|\cdot\|_m\right)}dt + \frac{\delta}{\sqrt{m}}\sqrt{c_1\log(48/\delta^2)} \right\} \lesssim \delta^2, \tag{20}$$

*then*

$$\|g_{0,\ell} - \hat{g}_\ell\|_{\mathcal{L}_2}^2 = O_{\mathbb{P}}\left( \phi_{\ell,n} + r_n + \frac{\mathcal{U}_n^2\eta_{\ell,n}(\delta_n)}{n\underline{\pi}_\ell} + \mathcal{B}_n\delta_n^2 \right) \tag{21}$$

*provided that $n\underline{\pi}_\ell^3/\overline{\pi}_\ell^2 \to \infty$, $\delta_n^2 n\underline{\pi}_\ell \to \infty$, and*

$$\lim_{n\to\infty} \left[ \sum_{k=0}^{\infty}\sum_{k'=1}^{\infty} \exp\left(-C_1\eta_{\ell,n}(2^{-k-k'-1})\right) + \sum_{k=0}^{\infty} \exp\left(-C_2\eta_{\ell,n}(2^{-k-1})\right) + \mathbb{P}(\|\epsilon\|_\infty > \mathcal{U}_n) \right] = 0 \tag{22}$$

*for some constants $C_1, C_2 > 0$, and*

$$\sup_{k\in\mathbb{N}} \sum_{k'=1}^{\infty} \frac{\eta_{\ell,n}(2^{-k-k'})}{2^{2k'}\eta_{\ell,n}(2^{-k})} \leq 1. \tag{23}$$

**Theorem B.3.** *[Upper Bound for First-Stage Estimation with Deep ReLU Network (Theorem 2 in (Padilla et al., 2024a))].* *Suppose that $\bar{f} \in \mathcal{H}(l_0, \mathcal{P}_0)$ for some $l_0 \in \mathbb{N}$ and $\mathcal{P}_0 \subset [1, \infty) \times \mathbb{N}$. Furthermore, assume that each function $m$ in the definition of $\bar{f}$ can have different smoothness $p_m = q_m + s_m$, for $q_m \in \mathbb{N}$, $s_m \in (0, 1]$, and of potentially different input dimension $K_m$, so that $(p_m, K_m) \in \mathcal{P}_0$. Let $K_{\max}$ be the largest input dimension and $p_{\max}$ the largest smoothness of any of the functions $m$. Suppose that all the partial derivatives of order less than or equal to $q_m$ are uniformly bounded by constant $c_2$, and each function $m$ is Lipschitz continuous with Lipschitz constant $C_{\text{Lip}} \geq 1$. Also, assume that $\max\{p_{\max}, K_{\max}\} = O(1)$. Let*

$$\phi_n = \max_{(p,K)\in\mathcal{P}_0} n^{\frac{-2p}{(2p+K)}}. \tag{24}$$

*Then there exists sufficiently large positive constants $c_3$ and $c_4$ such that if*

$$M = \lceil c_3 \log n \rceil \quad and \quad \nu = \left\lceil c_4 \max_{(p,K)\in\mathcal{P}_0} n^{\frac{K}{2(2p+K)}} \right\rceil \tag{25}$$

*or*

$$M = \left\lceil c_3 \max_{(p,K)\in\mathcal{P}_0} n^{\frac{K}{2(2p+K)}} \log n \right\rceil \quad and \quad \nu = \lceil c_4 \rceil, \tag{26}$$

*then $\hat{f}_{\mathcal{A}_n}$, with $\hat{f}$ as defined in (3) with $\mathcal{F} := \mathcal{F}(M, \nu)$, satisfies,*

$$\|\bar{f} - \hat{f}_{\mathcal{A}_n}\|_{\mathcal{L}_2}^2 = O_{\mathbb{P}}(r_n) \tag{27}$$

*where*

$$r_n := \frac{\max\{\mathcal{A}_n, \mathcal{U}_n^2\} \log n}{n} + \phi_n \log^3(n) \log(\mathcal{A}_n) \max\{\mathcal{A}_n, \mathcal{U}_n^2\}, \tag{28}$$

*provided that*

$$\lim_{n\to\infty} \mathbb{P}(\|\epsilon\|_\infty > \mathcal{U}_n) = 0$$

*and*

$$\mathcal{A}_n \geq 8 \max\{\|\bar{f}\|_\infty + \mathcal{U}_n, 8\|\bar{f}\|_\infty, 8\sqrt{\phi_n}\}$$

*hold.*

**Theorem B.4.** *[Detailed Statement of Theorem 3.4.]* *Suppose that the conditions of Theorem B.3 hold. Also, for $\ell \in \{1, \ldots, L\}$, let*

$$G_\ell(x) := f_{0,\ell}(x) - \sum_{k=1}^{L} f_{0,k}(x) \, \mathbb{P}(Z = k | X = x).$$

*Suppose that $G_\ell \in \mathcal{H}(R_\ell, \mathcal{P}_\ell)$ for some $R_\ell \in \mathbb{N}$ and $\mathcal{P}_\ell \subset [1, \infty) \times \mathbb{N}$. Furthermore, assume that each function $m$ in the definition of $G_\ell$ can have different smoothness $p_m = q_m + s_m$, for $q_m \in \mathbb{N}$, $s_m \in (0, 1]$, and of potentially different input dimension $K_m$, so that $(p_m, K_m) \in \mathcal{P}_\ell$. Let $K_{\ell,\max}$ be the largest input dimension and $p_{\ell,\max}$ the largest smoothness of any of the functions $m$. Suppose that all the partial derivatives of order less than or equal to $q_m$ are uniformly bounded by constant $c_2$, and each function $m$ is Lipschitz continuous with Lipschitz constant $C_{\text{Lip}} \geq 1$. Also, assume that $\max\{p_{\ell,\max}, K_{\ell,\max}\} = O(1)$. Let*

$$\phi_{\ell,n} = \max_{(p,K)\in\mathcal{P}_\ell} (n\underline{\pi}_\ell)^{\frac{-2p}{(2p+K)}}. \tag{29}$$

*Then there exists sufficiently large positive constants $c_3$ and $c_4$ such that if*

$$M_\ell = \lceil c_3 \log(n\underline{\pi}_\ell) \rceil \quad and \quad \nu_\ell = \left\lceil c_4 \max_{(p,K)\in\mathcal{P}_\ell} (n\underline{\pi}_\ell)^{\frac{K}{2(2p+K)}} \right\rceil \tag{30}$$

*or*

$$M_\ell = \left\lceil c_3 \max_{(p,K)\in\mathcal{P}_\ell} (n\underline{\pi}_\ell)^{\frac{K}{2(2p+K)}} \log(n\underline{\pi}_\ell) \right\rceil \quad and \quad \nu_\ell = \lceil c_4 \rceil, \tag{31}$$

*then $\hat{g}_\ell$ as defined in (5) with $\mathcal{F}_\ell := \mathcal{F}(M_\ell, \nu_\ell)$, satisfies,*

$$\|g_{0,\ell} - \hat{g}_\ell\|_{\mathcal{L}_2}^2 = O_{\mathbb{P}}\left( \frac{\phi_n \max\{\mathcal{A}_n, \mathcal{U}_n^2\} \log^3(n) \log(\mathcal{A}_n^2 n \overline{\pi}_\ell)}{\underline{\pi}_\ell} + \phi_{\ell,n} \max\{\mathcal{U}_n^2, \mathcal{B}_n\} \log^4(\mathcal{B}_n^2 n \overline{\pi}_\ell) \right) \tag{32}$$

*provided that $n\underline{\pi}_\ell^3 / \overline{\pi}_\ell^2 \to \infty$, $\mathcal{B}_n$ is chosen to satisfy (18).*

# C. Proofs

## C.1. Preliminaries

**Definition C.1.** Define $B_\infty(1) := \{f : [0,1]^d \to \mathbb{R} : \|f\|_\infty \le 1\}$. Given a function class $\mathcal{F}$ with $\mathcal{F} \subset B_\infty(1)$, we call $\delta_n > 0$ a critical radius for $\mathcal{F}$ if

$$\mathbb{E}\left(\sup_{f \in \mathrm{star}(\mathcal{F}) \,:\, \|f\|_n \le \delta_n} \frac{1}{n} \sum_{i=1}^{n} \xi_i f(x_i) \,\bigg|\, \{x_i\}_{i=1}^{n}\right) \le \delta_n^2,$$

where $\xi_1, \ldots, \xi_n$ are independent Rademacher variables independent of $\{x_i\}_{i=1}^{n}$ and where $\mathrm{star}(\mathcal{F})$ is defined as

$$\mathrm{star}(\mathcal{F}) := \{\lambda f \,:\, \lambda \in [0,1], f \in \mathcal{F}\}.$$

## C.2. Auxiliary Lemmas

**Lemma C.2.** *Let* $\tilde{y}_i = y_i - \hat{f}_{\mathcal{A}_n}(x_i)$ *for* $i = 1, \ldots, n$. *Then*

$$\tilde{y}_i = f_{0,z_i}(x_i) - \sum_{\ell=1}^{L} f_{0,\ell}(x_i)\,\mathbb{P}(Z = \ell | X = x_i) + (\bar{f}(x_i) - \hat{f}_{\mathcal{A}_n}(x_i)) + \epsilon_i,$$

*for* $i = 1, \ldots, n$.

*Proof.* Notice that

$$
\begin{aligned}
\tilde{y}_i &= y_i - \hat{f}_{\mathcal{A}_n}(x_i) \\
&= y_i + (\bar{f}(x_i) - \hat{f}_{\mathcal{A}_n}(x_i)) - \bar{f}(x_i) \\
&= f_0(x_i) + f_{0,z_i}(x_i) + \epsilon_i + (\bar{f}(x_i) - \hat{f}_{\mathcal{A}_n}(x_i)) - \bar{f}(x_i) \\
&= f_0(x_i) + f_{0,z_i}(x_i) + \epsilon_i + (\bar{f}(x_i) - \hat{f}_{\mathcal{A}_n}(x_i)) - [f_0(x_i) + \sum_{\ell=1}^{L} f_{0,\ell}(x_i)\,\mathbb{P}(Z = \ell | X = x_i).] \\
&= f_{0,z_i}(x_i) - \sum_{\ell=1}^{L} f_{0,\ell}(x_i)\,\mathbb{P}(Z = \ell | X = x_i) + (\bar{f}(x_i) - \hat{f}_{\mathcal{A}_n}(x_i)) + \epsilon_i.
\end{aligned}
$$

$\square$

**Lemma C.3.** *Suppose that the event* $\Omega_1 = \{\|\epsilon\|_\infty \le \mathcal{U}_n\}$ *holds and* $\mathcal{B}_n$ *satisfies*

$$\mathcal{B}_n \ge \mathcal{U}_n + \|\bar{f}\|_\infty + \mathcal{A}_n + 2\max_{\ell=1,\ldots,L} \|f_{0,\ell}\|_\infty. \tag{33}$$

*Then* $\|\tilde{y}\|_\infty \le \mathcal{B}_n$.

*Proof.* Notice that by Lemma C.2,

$$|\tilde{y}_i| \le \|f_{0,z_i}\|_\infty + \max_{\ell=1,\ldots,L} \|f_{0,\ell}\|_\infty + \|\bar{f}\|_\infty + \mathcal{A}_n + \mathcal{U}_n$$

and the claim follows.

$\square$

**Lemma C.4.** *Let*

$$G_\ell(x) := f_{0,z_i}(x) - \sum_{k=1}^{L} f_{0,k}(x)\,\mathbb{P}(Z = k | X = x)$$

*and* $\widetilde{G}_\ell \in \mathcal{F}_\ell$ *be such that* $\|G_\ell - \widetilde{G}_\ell\|_\infty \le \sqrt{\phi_{\ell,n}}$. *Such* $\widetilde{G}_\ell$ *exists by Theorem 3 in (Kohler & Langer, 2021). Then if* $\mathcal{B}_n \ge \|G_\ell\|_\infty + \sqrt{\phi_{\ell,n}}$ *we have that* $\widetilde{G}_\ell = \widetilde{G}_{\ell,\mathcal{B}_n} := (\widetilde{G}_\ell)_{\mathcal{B}_n}$.

*Proof.* Simply observe that

$$\|\widetilde{G}_\ell\|_\infty \leq \|\widetilde{G}_\ell - G_\ell\|_\infty + \|G_\ell\|_\infty \leq \sqrt{\phi_{\ell,n}} + \|G_\ell\|_\infty \leq \mathcal{B}_n$$

and the claim follows. $\square$

**Lemma C.5.** *Suppose that Assumption 3.1 holds. Then letting* $n_\ell = |\{i \in [n] : z_i = \ell\}|$, *for* $\ell \in \{1, \ldots, L\}$ *the event*

$$\Omega_2(\ell) := \left\{ \frac{n\underline{\pi}_\ell}{2} \leq n_\ell \leq 2n\overline{\pi}_\ell \right\}$$

*happens with probability at least*

$$1 - \exp\left(-\frac{n\underline{\pi}_\ell^3}{27\overline{\pi}_\ell^2}\right).$$

*Proof.* First, notice that $n_\ell$ is Binomial$(n, p_\ell)$ where

$$p_\ell = \int \mathbb{P}(z_i = \ell | x_i = x) dF_X(dx) \in [\underline{\pi}_\ell, \overline{\pi}_\ell],$$

by Assumption 3.1.

Hence, by the binomial concentration inequality, for instance see Lemma 5 in (Madrid Padilla et al., 2020), letting $\kappa_\ell = \underline{\pi}_\ell/(3\overline{\pi}_\ell) \in [0, 1]$,

$$
\begin{aligned}
\mathbb{P}(|np_\ell - n_\ell| \geq \kappa_\ell n\overline{\pi}_\ell) &\leq \mathbb{P}(|np_\ell - n_\ell| \geq \kappa_\ell np_\ell) \\
&\leq \exp\left(-\frac{\underline{\pi}_\ell^2 np_\ell}{27\overline{\pi}_\ell^2}\right) \\
&\leq \exp\left(-\frac{n\underline{\pi}_\ell^3}{27\overline{\pi}_\ell^2}\right).
\end{aligned}
$$

However, $|np_\ell - n_\ell| < \kappa_\ell n\overline{\pi}_\ell$ holds if and only if

$$np_\ell - \kappa_\ell n\overline{\pi}_\ell < n_\ell < np_\ell + \kappa_\ell n\overline{\pi}_\ell$$

which implies

$$\frac{n\underline{\pi}_\ell}{2} < n_\ell < 2n\overline{\pi}_\ell$$

given our choice of $\kappa_\ell$. The claim then follows. $\square$

### C.3. Proof of Theorem B.2

*Proof.* For a fixed $\ell \in \{1, \ldots, L\}$ let $\mathcal{I}_\ell := \{i \in [n] : z_i = \ell\}$ and $n_\ell := |\mathcal{I}_\ell|$, and we condition on $z$ being in the set

$$\{z \in \{1, \ldots, L\}^n : \frac{n\underline{\pi}_\ell}{2} \leq |\{i : z_i = \ell\}| \leq 2n\overline{\pi}_\ell\}. \tag{34}$$

By Lemma C.5 this happens with probability at least

$$1 - \exp\left(-\frac{n\underline{\pi}_\ell^3}{27\overline{\pi}_\ell^2}\right).$$

Let $\delta > 0$ to be defined later. Suppose that the event $\Omega_1$ defined in Lemma C.3 holds. Then, given our choice of $\mathcal{B}_n$, from Lemmas C.3 and C.4 we have $\widetilde{G}_\ell = \widetilde{G}_{\ell,\mathcal{B}_n}$ and $\|\tilde{y}\|_\infty \leq \mathcal{B}_n$. Hence,

$$\frac{1}{n_\ell} \sum_{i \in \mathcal{I}_\ell} (\tilde{y}_i - \hat{f}_{\ell,\mathcal{B}_n}(x_i))^2 \leq \frac{1}{n_\ell} \sum_{i \in \mathcal{I}_\ell} (\tilde{y}_i - \hat{f}_\ell(x_i))^2$$

where $\hat{f}_{l,\mathcal{B}_n} = (\hat{f}_l)_{\mathcal{B}_n}$. Also, by the basic inequality,

$$\frac{1}{n_\ell} \sum_{i \in \mathcal{I}_\ell} (\tilde{y}_i - \hat{f}_\ell(x_i))^2 \leq \frac{1}{n_\ell} \sum_{i \in \mathcal{I}_\ell} (\tilde{y}_i - \widetilde{G}_\ell(x_i))^2 = \frac{1}{n_\ell} \sum_{i \in \mathcal{I}_\ell} (\tilde{y}_i - \widetilde{G}_{\ell,\mathcal{B}_n}(x_i))^2.$$

As a result,

$$\frac{1}{n_\ell} \sum_{i \in \mathcal{I}_\ell} (\tilde{y}_i - \hat{f}_{\ell,\mathcal{B}_n}(x_i))^2 \leq \frac{1}{n_\ell} \sum_{\in \mathcal{I}_\ell} (\tilde{y}_i - \widetilde{G}_{\ell,\mathcal{B}_n}(x_i))^2.$$

However, by Lemma C.2 and the notation of Lemma C.4, $\tilde{y}_i = G_\ell(x_i) + (\bar{f}(x_i) - \hat{f}_{\mathcal{A}_n}(x_i)) + \epsilon_i$ for $i$ with $z_i = \ell$, then

$$\frac{1}{n_\ell} \sum_{i \in \mathcal{I}_\ell} [G_\ell(x_i) - \hat{f}_{\ell,\mathcal{B}_n}(x_i) + (\bar{f}(x_i) - \hat{f}_{\mathcal{A}_n}(x_i)) + \epsilon_i]^2 \leq \frac{1}{n_\ell} \sum_{i \in \mathcal{I}_\ell} [G_\ell(x_i) - \widetilde{G}_{\ell,\mathcal{B}_n} + (\bar{f}(x_i) - \hat{f}_{\mathcal{A}_n}(x_i)) + \epsilon_i]^2$$

which implies

$$
\begin{aligned}
\frac{1}{n_\ell} \sum_{i \in \mathcal{I}_\ell} (G_\ell(x_i) - \hat{f}_{\ell,\mathcal{B}_n}(x_i))^2 &\leq \frac{1}{n_\ell} \sum_{i \in \mathcal{I}_\ell} (G_\ell(x_i) - \widetilde{G}_{\ell,\mathcal{B}_n}(x_i))^2 + \frac{2}{n_\ell} \sum_{i \in \mathcal{I}_\ell} \epsilon_i(\hat{f}_{\ell,\mathcal{B}_n}(x_i) - \widetilde{G}_{\ell,\mathcal{B}_n}(x_i)) + \\
&\quad \frac{2}{n_\ell} \sum_{i \in \mathcal{I}_\ell} (\hat{f}_{\ell,\mathcal{B}_n}(x_i) - \widetilde{G}_{\ell,\mathcal{B}_n}(x_i))(\bar{f}(x_i) - \hat{f}_{\mathcal{A}_n}(x_i)) \\
&\leq \|\widetilde{G}_\ell - G_\ell\|_\infty^2 + \frac{2}{n_\ell} \sum_{i \in \mathcal{I}_\ell} \epsilon_i(\hat{f}_{\ell,\mathcal{B}_n}(x_i) - \widetilde{G}_{\ell,\mathcal{B}_n}(x_i)) + \\
&\quad \frac{2}{n_\ell} \sum_{i \in \mathcal{I}_\ell} (\hat{f}_{\ell,\mathcal{B}_n}(x_i) - \widetilde{G}_{\ell,\mathcal{B}_n}(x_i))(\bar{f}(x_i) - \hat{f}_{\mathcal{A}_n}(x_i)) \\
&\leq \phi_{\ell,n} + \frac{2}{n_\ell} \sum_{i \in \mathcal{I}_\ell} \epsilon_i(\hat{f}_{\ell,\mathcal{B}_n}(x_i) - \widetilde{G}_{\ell,\mathcal{B}_n}(x_i)) + \\
&\quad \frac{2}{n_\ell} \left[ \frac{1}{32} \sum_{i \in \mathcal{I}_\ell} (\hat{f}_{\ell,\mathcal{B}_n}(x_i) - \widetilde{G}_{\ell,\mathcal{B}_n}(x_i))^2 + 8 \sum_{i \in \mathcal{I}_\ell} (\bar{f}(x_i) - \hat{f}_{\mathcal{A}_n}(x_i))^2 \right] \\
&\leq \phi_{\ell,n} + \frac{2}{n_\ell} \sum_{i \in \mathcal{I}_\ell} \epsilon_i(\hat{f}_{\ell,\mathcal{B}_n}(x_i) - \widetilde{G}_{\ell,\mathcal{B}_n}(x_i)) + \frac{1}{8n_\ell} \sum_{i \in \mathcal{I}_\ell} (G_\ell(x_i) - \hat{f}_{\ell,\mathcal{B}_n}(x_i))^2 \\
&\quad \frac{1}{8n_\ell} \sum_{i \in \mathcal{I}_\ell} (G_\ell(x_i) - \widetilde{G}_\ell(x_i))^2 + \frac{16}{n_\ell} \sum_{i \in \mathcal{I}_\ell} (\bar{f}(x_i) - \hat{f}_{\mathcal{A}_n}(x_i))^2 \\
&\leq \frac{9\phi_{\ell,n}}{8} + \frac{2}{n_\ell} \sum_{i \in \mathcal{I}_\ell} \epsilon_i(\hat{f}_{\ell,\mathcal{B}_n}(x_i) - \widetilde{G}_{\ell,\mathcal{B}_n}(x_i)) + \frac{1}{8n_\ell} \sum_{i \in \mathcal{I}_\ell} (G_\ell(x_i) - \hat{f}_{\ell,\mathcal{B}_n}(x_i))^2 \\
&\quad \frac{16}{n_\ell} \sum_{i \in \mathcal{I}_\ell} (\bar{f}(x_i) - \hat{f}_{\mathcal{A}_n}(x_i))^2.
\end{aligned}
$$

Therefore,

$$\frac{7}{8n_\ell} \sum_{i \in \mathcal{I}_\ell} (G_\ell(x_i) - \hat{f}_{\ell,\mathcal{B}_n}(x_i))^2 \leq \frac{9\phi_{\ell,n}}{8} + \frac{2}{n_\ell} \sum_{i \in \mathcal{I}_\ell} \epsilon_i(\hat{f}_{\ell,\mathcal{B}_n}(x_i) - \widetilde{G}_{\ell,\mathcal{B}_n}(x_i)) + \frac{16}{n_\ell} \sum_{i \in \mathcal{I}_\ell} (\bar{f}(x_i) - \hat{f}_{\mathcal{A}_n}(x_i))^2. \tag{35}$$

Next let $\mathcal{H}_\ell := \{(f_{\mathcal{B}_n} - g_{\mathcal{B}_n})/(2\mathcal{B}_n) : f, g \in \mathcal{F}_{\ell,\mathcal{B}_n}\}$. Then $f \in \mathcal{H}_\ell$ implies $\|f\|_\infty \leq 1$ and

$$\log N(\delta, \mathcal{H}_\ell, \|\cdot\|_{\mathcal{I}_\ell}) \leq 2 \log N(\delta/2, \mathcal{F}_{\ell,\mathcal{B}_n}, \|\cdot\|_{\mathcal{I}_\ell}) \leq 2\eta_{\ell,n}(\delta/2), \tag{36}$$

where the second inequality follows since we are conditioning on $z$ satisfiying (34) and due to our assumption that (17) holds.

Hence, by Lemma 4 from (Padilla et al., 2024a), for some positive constant $C > 0$,

$$
\begin{aligned}
\frac{2\mathcal{B}_n}{n_\ell} \sum_{i \in \mathcal{I}_\ell} \frac{(\hat{f}_{\ell,\mathcal{B}_n}(x_i) - \widetilde{G}_{\ell,\mathcal{B}_n}(x_i))}{2\mathcal{B}_n} \epsilon_i &\leq 2C\mathcal{B}_n \mathcal{U}_n \|(\hat{f}_{\ell,\mathcal{B}_n} - \widetilde{G}_{\ell,\mathcal{B}_n})/(2\mathcal{B}_n)\|_{\mathcal{I}_\ell} \cdot \\
&\quad \sqrt{\eta_{\ell,n}(\|(\hat{f}_{\ell,\mathcal{B}_n} - \widetilde{G}_{\ell,\mathcal{B}_n})/(4\mathcal{B}_n)\|_{\mathcal{I}_\ell})/n_\ell} \\
&\leq C\mathcal{U}_n \|\hat{f}_{\ell,\mathcal{B}_n} - \widetilde{G}_{\ell,\mathcal{B}_n}\|_{\mathcal{I}_\ell} \cdot \sqrt{\eta_{\ell,n}(\|(\hat{f}_{\ell,\mathcal{B}_n} - \widetilde{G}_{\ell,\mathcal{B}_n})/(4\mathcal{B}_n)\|_{\mathcal{I}_\ell})/n_\ell}
\end{aligned} \tag{37}
$$

with probability at least

$$1 - 4\sum_{k=0}^{\infty}\sum_{k'=1}^{\infty} \exp\left(-C_1 \eta_{\ell,n}(2^{-k-k'-1})\right) - 4\sum_{k=0}^{\infty}\exp\left(-C_2\eta_{\ell,n}(2^{-k-1})\right) - \mathbb{P}(\|\epsilon\|_\infty > \mathcal{U}_n).$$

Define $\Omega_3$, the event such that (37) holds and let $\delta > 0$. From now on suppose that $\Omega_1 \cap \Omega_3$ holds. If

$$\|(\hat{f}_{\ell,\mathcal{B}_n} - \widetilde{G}_{\ell,\mathcal{B}_n})/(4\mathcal{B}_n)\|_{\mathcal{I}_\ell} \leq \delta$$

we obtain

$$\|(\hat{f}_{\ell,\mathcal{B}_n} - \widetilde{G}_{\ell,\mathcal{B}_n})\|_{\mathcal{I}_\ell}^2 \leq 16\mathcal{B}_n^2 \delta^2,$$

which implies

$$\|\hat{f}_{\ell,\mathcal{B}_n} - G_\ell\|_{\mathcal{I}_\ell}^2 \leq 32\mathcal{B}_n^2\delta^2 + 2\phi_{\ell,n},$$

Suppose now that

$$\|(\hat{f}_{\ell,\mathcal{B}_n} - \widetilde{G}_{\ell,\mathcal{B}_n})/(4\mathcal{B}_n)\|_{\mathcal{I}_\ell} > \delta.$$

Then (35) and (37) imply

$$
\begin{aligned}
\frac{7}{8n_\ell}\sum_{i\in\mathcal{I}_\ell}(G_\ell(x_i) - \hat{f}_{\ell,\mathcal{B}_n}(x_i))^2 &\leq& \frac{9\phi_{\ell,n}}{8} + \frac{16}{n_\ell}\sum_{i\in\mathcal{I}_\ell}(\bar{f}(x_i) - \hat{f}_{\mathcal{A}_n}(x_i))^2 \\
&& + 2C\mathcal{U}_n\|\hat{f}_{\ell,\mathcal{B}_n} - \widetilde{G}_{\ell,\mathcal{B}_n}\|_{\mathcal{I}_\ell}\cdot\sqrt{\eta_{\ell,n}(\delta)/n_\ell} \\
&\leq& \frac{9\phi_{\ell,n}}{8} + \frac{16}{n_\ell}\sum_{i\in\mathcal{I}_\ell}(\bar{f}(x_i) - \hat{f}_{\mathcal{A}_n}(x_i))^2 + \\
&& \frac{\|\hat{f}_{\ell,\mathcal{B}_n} - \widetilde{G}_{\ell,\mathcal{B}_n}\|_{\mathcal{I}_\ell}^2}{4} + \frac{4C^2\mathcal{U}_n^2\eta_{\ell,n}(\delta)}{n_\ell} \\
&\leq& \frac{9\phi_{\ell,n}}{8} + \frac{16}{n_\ell}\sum_{i\in\mathcal{I}_\ell}(\bar{f}(x_i) - \hat{f}_{\mathcal{A}_n}(x_i))^2 + \\
&& \frac{\|\hat{f}_{\ell,\mathcal{B}_n} - G_\ell\|_{\mathcal{I}_\ell}^2}{2} + \frac{\|G_\ell - \widetilde{G}_{\ell,\mathcal{B}_n}\|_{\mathcal{I}_\ell}^2}{2} + \frac{4C^2\mathcal{U}_n^2\eta_{\ell,n}(\delta)}{n_\ell}.
\end{aligned}
\tag{38}
$$

Hence,

$$
\begin{aligned}
\frac{3}{8}\|G_\ell - \hat{f}_{\ell,\mathcal{B}_n}\|_{\mathcal{I}_\ell}^2 &\leq& \frac{17\phi_{\ell,n}}{8} + 16\|\bar{f} - \hat{f}_{\mathcal{A}_n}\|_{\mathcal{I}_\ell}^2 + \frac{4C^2\mathcal{U}_n^2\eta_{\ell,n}(\delta)}{n_\ell} \\
&\leq& \frac{17\phi_{\ell,n}}{8} + 32\|\tilde{f}_{\mathcal{A}_n} - \hat{f}_{\mathcal{A}_n}\|_{\mathcal{I}_\ell}^2 + 32\|\bar{f} - \tilde{f}\|_{\mathcal{I}_\ell}^2 + \frac{4C^2\mathcal{U}_n^2\eta_{\ell,n}(\delta)}{n_\ell} \\
&\leq& \frac{17\phi_{\ell,n}}{8} + 32\|\tilde{f}_{\mathcal{A}_n} - \hat{f}_{\mathcal{A}_n}\|_{\mathcal{I}_\ell}^2 + 32\phi_n + \frac{4C^2\mathcal{U}_n^2\eta_{\ell,n}(\delta)}{n_\ell}
\end{aligned}
\tag{39}
$$

with $\tilde{f}$ and $\phi_n$ as in the notation of Theorem B.1.

Moreover, by choosing $\delta$ to satisfy

$$\max_{m\,:\,\frac{n\pi_\ell}{2}\leq m\leq 2n\overline{\pi}_\ell} \sup_{\{x_j\}_{j=1}^m \subset [0,1]^d} \left\{\frac{1}{\sqrt{m}}\int_{\delta^2/48}^\delta \sqrt{\log N\left(t/4, \mathcal{F}_{\ell,\mathcal{B}_n}, \|\cdot\|_m\right)}dt + \frac{\delta}{\sqrt{m}}\sqrt{c_1\log(48/\delta^2)}\right\} \lesssim \delta^2, \tag{40}$$

we ensure that

$$\frac{1}{\sqrt{n_\ell}}\int_{\delta^2/48}^\delta \sqrt{\log N\left(t/4, \mathcal{F}_{\ell,\mathcal{B}_n}, \|\cdot\|_{\mathcal{I}_\ell}\right)}dt + \frac{\delta}{\sqrt{n_\ell}}\sqrt{c_1\log(48/\delta^2)} \lesssim \delta^2,$$

Hence, from (36) and Lemma 6 in (Padilla et al., 2024a) it follows that the event

$$\Omega_4 := \left\{\sup_{h\in\mathcal{H}_\ell}\left|\frac{|\|h\|_{\mathcal{I}_\ell}^2 - \|h\|_{\mathcal{L}_2}^2|}{\frac{1}{2}\|h\|_{\mathcal{L}_2}^2 + \frac{\delta^2}{2}}\right| \leq 1\right\}$$

holds with probability at least

$$1 - c_2 \exp(-c_1 \delta^2 n_{\underline{\pi}_\ell})$$

for constants $c_1, c_2 > 0$.

Therefore, from (39), in the event $\Omega_1 \cap \Omega_3 \cap \Omega_4$, we obtain that

$$
\begin{aligned}
\|G_\ell - \hat{f}_{\ell,\mathcal{B}_n}\|_{\mathcal{L}_2}^2 &\leq 2\|G_\ell - \widetilde{G}_\ell\|_{\mathcal{L}_2}^2 + 2\|\widetilde{G}_\ell - \hat{f}_{\ell,\mathcal{B}_n}\|_{\mathcal{L}_2}^2 \\
&\leq 2\|G_\ell - \widetilde{G}_\ell\|_\infty^2 + 2[2\|\widetilde{G}_\ell - \hat{f}_{\ell,\mathcal{B}_n}\|_{\mathcal{I}_\ell}^2 + \delta^2] \\
&\leq 2\|G_\ell - \widetilde{G}_\ell\|_\infty^2 + \frac{32}{3}\Big[\frac{17\phi_{\ell,n}}{8} + 32\|\tilde{f}_{\mathcal{A}_n} - \hat{f}_{\mathcal{A}_n}\|_{\mathcal{I}_\ell}^2 + 32\phi_n + \frac{4C^2\mathcal{U}_n^2\eta_{\ell,n}(\delta)}{n_\ell}\Big] + 2\delta^2 \\
&\lesssim \phi_{\ell,n} + \|\tilde{f}_{\mathcal{A}_n} - \hat{f}_{\mathcal{A}_n}\|_{\mathcal{I}_\ell}^2 + \phi_n + \frac{\mathcal{U}_n^2\eta_{\ell,n}(\delta)}{n_\ell} + \delta^2. \\
&\lesssim \phi_{\ell,n} + \|\tilde{f}_{\mathcal{A}_n} - \hat{f}_{\mathcal{A}_n}\|_{\mathcal{I}_\ell}^2 + \phi_n + \frac{\mathcal{U}_n^2\eta_{\ell,n}(\delta)}{n_{\underline{\pi}_\ell}} + \delta^2.
\end{aligned}
\tag{41}
$$

Furthermore, let $\mathcal{F}_{\mathcal{A}_n} := \{f_{\mathcal{A}_n}/(2\mathcal{A}_n) : f \in \mathcal{F}\}$ and

$$\mathcal{H} := \{f - g : f, g \in \mathcal{F}_{\mathcal{A}_n}\}.$$

Then, by choosing $\delta$ to satisfy

$$
\max_{m : \frac{n_{\underline{\pi}_\ell}}{2} \leq m \leq 2n\overline{\pi}_\ell} \sup_{\{x_j\}_{j=1}^m \subset [0,1]^d} \left\{ \frac{1}{\sqrt{m}} \int_{\delta^2/48}^\delta \sqrt{\log N\left(t/4, \mathcal{F}_{\mathcal{A}_n}, \|\cdot\|_m\right)} dt + \frac{\delta}{\sqrt{m}} \sqrt{c_1 \log(48/\delta^2)} \right\} \lesssim \delta^2,
\tag{42}
$$

Lemma 6 in (Padilla et al., 2024a) also implies that the event

$$
\Omega_5 := \left\{ \sup_{h \in \mathcal{H}} \left| \frac{|\|h\|_{\mathcal{I}_\ell}^2 - \|h\|_{\mathcal{L}_2}^2|}{\frac{1}{2}\|h\|_{\mathcal{L}_2}^2 + \frac{\delta^2}{2}} \right| \leq 1 \right\}
$$

holds with probability at least

$$1 - c_2 \exp(-c_1 \delta^2 n\pi_\ell).$$

As a result, from (41), in the event $\Omega_1 \cap \Omega_3 \cap \Omega_4 \cap \Omega_5$, we have that

$$
\begin{aligned}
\|G_\ell - \hat{f}_{\ell,\mathcal{B}_n}\|_{\mathcal{L}_2}^2 &\lesssim \phi_{\ell,n} + \|\tilde{f}_{\mathcal{A}_n} - \hat{f}_{\mathcal{A}_n}\|_{\mathcal{I}_\ell}^2 + \phi_n + \frac{\mathcal{U}_n^2\eta_{\ell,n}(\delta)}{n_\ell} + \delta^2 \\
&\lesssim \phi_{\ell,n} + \|\tilde{f}_{\mathcal{A}_n} - \hat{f}_{\mathcal{A}_n}\|_{\mathcal{L}_2}^2 + \phi_n + \frac{\mathcal{U}_n^2\eta_{\ell,n}(\delta)}{n_\ell} + \delta^2 \\
&\lesssim \phi_{\ell,n} + r_n + \frac{\mathcal{U}_n^2\eta_{\ell,n}(\delta)}{n_{\underline{\pi}_\ell}} + \delta^2.
\end{aligned}
\tag{43}
$$

Finally, the claim follows from the inequality

$$\|g_{0,\ell} - \hat{g}_\ell\|_{\mathcal{L}_2}^2 \lesssim \|G_\ell - \hat{f}_{\ell,\mathcal{B}_n}\|_{\mathcal{L}_2}^2 + \|\bar{f} - \hat{f}_{\mathcal{A}_n}\|_{\mathcal{L}_2}^2.$$

$\square$

## C.4. Proof of Theorem B.4

We apply Theorem B.2 and proceed as in the proof of Theorem 2 in (Padilla et al., 2024a). Specifically, let $\mathcal{F}_{\mathcal{B}_n}(M_\ell, \nu_\ell) := \{f_{\mathcal{B}_n} : f \in \mathcal{F}(M_\ell, \nu_\ell)\}$. Then, by Theorem 6 from (Bartlett et al., 2019), it holds that the VC dimension of $\mathcal{C}(M_\ell, \nu_\ell)$ satisfies

$$\text{VC}(\mathcal{F}(M_\ell, \nu_\ell)) \lesssim M_\ell^2 \nu_\ell^2 \log(M_\ell \nu_\ell).$$

As a result, by Lemma 9.2 and Theorem 9.4 from (Györfi et al., 2002), for any $\delta \in (0,1)$ and for any $\{x_j\}_{j=1}^m$ with induced norm $\|\cdot\|_m$, it holds that

$$
\begin{aligned}
\log N(\delta, \mathcal{F}_{\mathcal{B}_n}(M_\ell, \nu_\ell), \|\cdot\|_m) &\lesssim M^2\nu^2 \log(M_\ell\nu_\ell) \cdot \left[\log(\mathcal{B}_n^2\delta^{-2}) + \log\log(\mathcal{B}_n^2\delta^{-2})\right] \\
&\lesssim M_\ell^2\nu_\ell^2 \log(M_\ell\nu_\ell) \log(\mathcal{B}_n^2\delta^{-1}) \\
&\leq C_0(n\underline{\pi}_\ell)\phi_{\ell,n} \log^3(n\underline{\pi}_\ell) \log(\mathcal{B}_n^2\delta^{-1})
\end{aligned}
\tag{44}
$$

where $C_0 > 0$ is a constant, and the last inequality holds from our choice of $M_\ell$ and $\nu_\ell$. Here, $m \in \mathbb{N}$ is arbitray.

Therefore, with $\mathcal{F}_\ell := \mathcal{F}(M_\ell, \nu_\ell)$, we have that

$$
\max_{\frac{n\underline{\pi}_\ell}{2} \leq n_\ell \leq 2n\overline{\pi}_\ell} \sup_{\{x_i\}_{i\in\mathcal{I}_\ell}\subset[0,1]^d} \log N(\delta, \mathcal{F}_{\ell,\mathcal{B}_n}, \|\cdot\|_{\mathcal{I}_\ell}) \leq \eta_{\ell,n}(\delta) \ \ \forall \delta \in (0,1),
$$

for

$$
\eta_{\ell,n}(\delta) := C_0(n\underline{\pi}_\ell)\phi_{\ell,n} \log^3(n\underline{\pi}_\ell) \log(\mathcal{B}_n^2\delta^{-1}).
\tag{45}
$$

Thus, (17) holds with $\eta_{\ell,n}(\delta)$ as in (45).

Furthermore, for any positive constants $C_1$ and $C_2$, we have that

$$
\begin{aligned}
&\sum_{l=0}^\infty \sum_{l'=1}^\infty \exp\left(-C_1\eta_{\ell,n}(2^{-l-l'-1})\right) + \sum_{l=0}^\infty \exp\left(-C_2\eta_{\ell,n}(2^{-l-1})\right) \\
&= \sum_{l=0}^\infty \sum_{l'=1}^\infty \exp\left(-C_1C_0(l+l'+1)n\underline{\pi}_\ell\phi_{\ell,n} \log^3(n\underline{\pi}_\ell) \log(2\mathcal{B}_n^2)\right) \\
&\quad + \sum_{l=0}^\infty \exp\left(-C_2C_0 n\underline{\pi}_\ell\phi_{\ell,n} \log^3(n\underline{\pi}_\ell) \log(2\mathcal{B}_n^2)\right) \\
&= \left[\exp\left(-C_1C_0 n\underline{\pi}_\ell\phi_{\ell,n} \log^3(n\underline{\pi}_\ell) \log(2\mathcal{B}_n^2)\right) / \left[1 - \exp\left(-C_1C_0 n\underline{\pi}_\ell\phi_{\ell,n} \log^3(n\underline{\pi}_\ell) \log(2\mathcal{B}_n^2)\right)\right]\right]^2 + \\
&\quad \exp\left(-C_2C_0 n\underline{\pi}_\ell\phi_{\ell,n} \log^3(n\underline{\pi}_\ell) \log(2\mathcal{B}_n^2)\right) / \left[1 - \exp\left(-C_2C_0 n\underline{\pi}_\ell\phi_{\ell,n} \log^3(n\underline{\pi}_\ell) \log(2\mathcal{B}_n^2)\right)\right] \\
&\underset{n\to\infty}{\to} 0,
\end{aligned}
\tag{46}
$$

since

$$
\lim_{n\to\infty} n\underline{\pi}_\ell\phi_{\ell,n} \log^3(n\underline{\pi}_\ell) \log(2\mathcal{B}_n^2) = \infty
$$

holds provided that

$$
\lim_{n\to\infty} n\underline{\pi}_\ell = \infty,
$$

which holds by assumption. Thus, we have verified (22).

Next, notice that

$$
\sup_{k\in\mathbb{N}} \sum_{k'=1}^\infty \frac{\eta_{\ell,n}(2^{-k-k'})}{2^{2k'}\eta_{\ell,n}(2^{-k})} \leq \sup_{k\in\mathbb{N}} \sum_{k'=1}^\infty \frac{(k+k')}{2^{2k'}k} \leq 1,
$$

which verifies (23).

Now, it remains to verify (19) and (20) for an appropriate $\delta > 0$. Towards that end, notice that by the proof of Lemma 6 in (Padilla et al., 2024a), (19) holds if $\delta$ is chosen to satisfy

$$
\delta \geq \max_{m : \frac{n\underline{\pi}_\ell}{2} \leq m \leq 2n\overline{\pi}_\ell} c_1 \left[\sqrt{\frac{\log m}{m}} + \sqrt{\frac{\log N\left(1/(48m), \mathcal{F}_{\ell,\mathcal{B}_n}, \|\cdot\|_m\right)}{m}}\right]
\tag{47}
$$

for a positive constant $c_1$. However, by (44), we can take

$$
\sqrt{\frac{\log(n\overline{\pi}_\ell)}{n\underline{\pi}_\ell}} + \sqrt{\phi_{\ell,n} \log^3(n\underline{\pi}_\ell) \log(\mathcal{B}_n^2 n\overline{\pi}_\ell)} \lesssim \delta
$$

to ensure that (19) holds. Similarly, by the proof of Lemma 6 in (Padilla et al., 2024a), (20) holds if $\delta$ is chosen to satisfy

$$\delta \geq \max_{m: \frac{n\pi_\ell}{2} \leq m \leq 2n\overline{\pi}_\ell} c_1 \left[ \sqrt{\frac{\log m}{m}} + \sqrt{\frac{\log N\left(1/(48m), \mathcal{F}_{\mathcal{A}_n}, \|\cdot\|_m\right)}{m}} \right] \tag{48}$$

which holds if we take

$$\sqrt{\frac{\log(n\overline{\pi}_\ell)}{n\underline{\pi}_\ell}} + \sqrt{\frac{\phi_n \log^3(n) \log(\mathcal{A}_n^2 n\overline{\pi}_\ell)}{\underline{\pi}_\ell}} \lesssim \delta,$$

by our choice of $\mathcal{F}$ in Theorem B.2. Therefore, taking

$$\sqrt{\frac{\log(n\overline{\pi}_\ell)}{n\underline{\pi}_\ell}} + \sqrt{\frac{\phi_n \log^3(n) \log(\mathcal{A}_n^2 n\overline{\pi}_\ell)}{\underline{\pi}_\ell}} + \sqrt{\phi_{\ell,n} \log^3(n\underline{\pi}_\ell) \log(\mathcal{B}_n^2 n\overline{\pi}_\ell)} \lesssim \delta,$$

it follows that (19) and (20) both hold.

### C.5. Proof of Theorem 3.7

*Proof.* Proceeding as in the proof of Theorem B.2, supposing that

$$\|(\hat{f}_{\ell,\mathcal{B}_n} - \widetilde{G}_{\ell,\mathcal{B}_n})/(4\mathcal{B}_n)\|_{\mathcal{I}_\ell} > \delta,$$

we obtain that, in the event $\Omega_1 \cap \Omega_3 \cap \Omega_4$,

$$\|G_\ell - \hat{f}_{\ell,\mathcal{B}_n}\|_{\mathcal{L}_2}^2 \lesssim \phi_{\ell,n} + \|\tilde{f}_{\mathcal{A}_n} - \hat{f}_{\mathcal{A}_n}\|_{\mathcal{I}_\ell}^2 + \phi_n + \frac{\mathcal{U}_n^2 \eta_{\ell,n}(\delta)}{n\underline{\pi}_\ell} + \delta^2. \tag{49}$$

Next, notice that since $\hat{f}_{\mathcal{A}_n}$ is independent of $\{(x_i)\}_{i \in \mathcal{I}_\ell}$, we obtain that

$$\mathbb{E}(\|\tilde{f}_{\mathcal{A}_n} - \hat{f}_{\mathcal{A}_n}\|_{\mathcal{I}_\ell}^2 \,|\, \hat{f}_{\mathcal{A}_n}) = \|\tilde{f}_{\mathcal{A}_n} - \hat{f}_{\mathcal{A}_n}\|_{\mathcal{L}_2}^2.$$

Furthermore, for all $i \in \mathcal{I}_\ell$, we have

$$0 \leq (\tilde{f}_{\mathcal{A}_n}(x_i) - \hat{f}_{\mathcal{A}_n}(x_i))^2 \leq 2\mathcal{A}_n^2.$$

Hence, by Hoeffding's inequality, there exist constants $c_1, c_2 > 0$ such that

$$\|\tilde{f}_{\mathcal{A}_n} - \hat{f}_{\mathcal{A}_n}\|_{\mathcal{I}_\ell}^2 \leq \|\tilde{f}_{\mathcal{A}_n} - \hat{f}_{\mathcal{A}_n}\|_{\mathcal{L}_2}^2 + c_1 \mathcal{A}_n^2 \sqrt{\frac{\log n}{n\underline{\pi}_\ell}}, \tag{50}$$

with probability at least

$$1 - \exp\left(-c_2 \log n\right).$$

Let $\Omega_5$ be the event that (50) holds. Then with $\Omega_1, \ldots, \Omega_4$ as in the proof of Theorem B.2, we obtain that in $\Omega_1 \cap \Omega_2 \cap \Omega_3 \cap \Omega_4 \cap \Omega_5$,

$$
\begin{aligned}
\|G_\ell - \hat{f}_{\ell,\mathcal{B}_n}\|_{\mathcal{L}_2}^2 &\lesssim \phi_{\ell,n} + \|\tilde{f}_{\mathcal{A}_n} - \hat{f}_{\mathcal{A}_n}\|_{\mathcal{L}_2}^2 + \mathcal{A}_n^2 \sqrt{\frac{\log n}{n\underline{\pi}_\ell}} + \phi_n + \frac{\mathcal{U}_n^2 \eta_{\ell,n}(\delta)}{n_\ell} + \delta^2 \\
&\lesssim \phi_{\ell,n} + r_n + \mathcal{A}_n^2 \sqrt{\frac{\log n}{n\underline{\pi}_\ell}} + \frac{\mathcal{U}_n^2 \eta_{\ell,n}(\delta)}{n\underline{\pi}_\ell} + \delta^2.
\end{aligned}
\tag{51}
$$

Finally, the claim follows as in the proof of Theorem B.2 and the inequality

$$\|g_{0,\ell} - \hat{g}_\ell\|_{\mathcal{L}_2}^2 \lesssim \|G_\ell - \hat{f}_{\ell,\mathcal{B}_n}\|_{\mathcal{L}_2}^2 + \|\bar{f} - \hat{f}_{\mathcal{A}_n}\|_{\mathcal{L}_2}^2.$$

$\square$

## C.6. Proof of Corollary 3.8

*Proof.* The proof follows as that of Theorem B.4. The only difference is that we do not need to enforce (20). Hence, $\delta$ can be chosen simply to satisfy (48) which as in the proof of Theorem B.4 can be done with $\delta$ satisfying

$$\sqrt{\frac{\log(n\overline{\pi}_\ell)}{n\underline{\pi}_\ell}} + \sqrt{\phi_{\ell,n}\log^3(n\underline{\pi}_\ell)\log(\mathcal{B}_n^2 n\overline{\pi}_\ell)} \lesssim \delta.$$

Then the conclusion follows from Theorem B.2 and by noticing that $K^\ell > 2p^\ell$ implies

$$\sqrt{\frac{\log n}{n\underline{\pi}_\ell}} << \phi_{\ell,n}.$$

$\square$

# D. Applications to Other Nonparametric Estimators

The general theoretical framework developed in Section 3.1 applies to a broad class of nonparametric estimators beyond deep ReLU networks. In this section, we present two additional examples: orthogonal series regression and trend filtering. For trend filtering, we instantiate Theorem B.2 (our general result under shared two-stage estimation), and to the best of our knowledge, the transfer learning setting has not been previously studied for this estimator, with our results yielding the first convergence guarantees in this regime. For orthogonal series regression, we instantiate Theorem 3.7 (our result under independent two-stage estimation) and recover, up to logarithmic factors, the existing convergence rates established in (Wang et al., 2016), providing a unified theoretical perspective on offset-based transfer learning.

## D.1. Trend Filtering

We now apply Theorem B.1 to total variation–based estimators, commonly referred to as *trend filtering* (Mammen & Van De Geer, 1997; Tibshirani, 2014). Trend filtering estimates a univariate regression function by penalizing the discrete total variation of its $r$th-order derivative, and is known to adapt to spatially varying smoothness. To the best of our knowledge, the pretraining setting has not been previously studied for this class of estimators.

Let $\mathcal{S}$ be a space of univariate functions invariant to scalar multiplication and $R : \mathcal{S} \to \mathbb{R}$ a regularizer that acts in $\mathcal{S}$ and it is a seminorm.

**Assumption D.1.** Suppose that there exists constants $K > 0$ and $0 < w \leq 1$ such that for all $m \in \mathbb{N}$ and $\delta \in (0,1)$, it holds that

$$\sup_{x_1,\ldots,x_m \subset [0,1]} \log N(\delta, B_R(1) \cap B_\infty(1), \|\cdot\|_m) \leq K\delta^{-w},$$

where $\|\cdot\|_m$ is the empirical norm induced by $x_1,\ldots,x_m$, and $B_R(1)$ is the unit ball defined by $R$:

$$B_R(1) := \{f \in \mathcal{S} : R(f) \leq 1\}$$

and $B_\infty(1) = \{f \in \mathcal{S} : \|f\|_\infty \leq 1\}$.

**Assumption D.2.** Suppose that $\mathbb{P}(Z = \ell | X = x) = \pi_\ell \in (0,1)$ for all $x \in [0,1]$ and $\ell \in \{1,\ldots,L\}$.

Assumption D.2 restricts the probabilities to be constant for notational simplicity. Consider the model in (1) with $d = 1$ and, for simplicity, with $f_0 = 0$. Our goal is to estimate the functions $f_{0,1},\ldots,f_{0,L}$ which now coincide with $g_{0,1},\ldots,g_{0,L}$, respectively.

As an estimator we consider $\hat{g}_\ell$ defined in (3) and (4), and with

$$\mathcal{F} := \{f \in \mathcal{S} : R(f) \leq V\} \quad \text{and} \quad \mathcal{F}_\ell := \{f \in \mathcal{S} : R(f) \leq V_\ell\}. \tag{52}$$

**Corollary D.3.** *Suppose that Assumptions D.1 and D.2 hold and $V \geq V_0$ and $V_\ell \geq V_{0,\ell}$ where*

$$V_0 := R\left(\sum_{k=1}^L \pi_k g_{0,k}\right), \quad V_{0,\ell} := R\left(g_{0,\ell} - \sum_{k=1}^L \pi_k g_{0,k},\right),$$

*and* $\min\{V_0, V_{0,\ell}\} \gtrsim \log n / n^{1/2 - 1/(2+w)}$. *Suppose that $\mathcal{A}_n$ is chosen to satisfy $\mathcal{A}_n \to \infty$ and*

$$\mathcal{A}_n \geq 8 \max\{\|\bar{f}\|_\infty + \mathcal{U}_n, 8\|\bar{f}\|_\infty, 8\sqrt{\phi_n}, 1\}.$$

*Moreover, assume that $\mathcal{B}_n$ is taken to satisfy $\mathcal{B}_n$ and (18). Then the estimator $\hat{g}_\ell$ defined in (5) satisfies*

$$\|g_{0,\ell} - \hat{g}_\ell\|_{\mathcal{L}_2}^2 = O_\mathbb{P}\left(\frac{\max\{\mathcal{U}_n^2, \mathcal{A}_n, \mathcal{B}_n\}(V^{2w/(2+w)}\mathcal{A}_n^{2w/(2+w)} + V_\ell^{2w/(2+w)}\mathcal{B}_n^{2w/(2+w)})}{(n\pi_\ell)^{2/(2+w)}}\right) \tag{53}$$

*provided that $n\pi_\ell \to \infty$.*

*Remark* D.4. In the natural setting where $\max\left\{\mathcal{U}_n, \mathcal{A}_n, \mathcal{B}_n, \|\bar{f}\|_\infty, \max_{\ell=1,\dots,L}\|f_{0,\ell}\|_\infty, \|G_\ell\|_\infty\right\} = O(\text{poly}(\log n))$, *and ignoring logarithmic factors, the convergence rate in (53) simplifies to*

$$\left(V_0^{2w/(2+w)} + V_{0,\ell}^{2w/(2+w)}\right)(n\pi_\ell)^{-2/(2+w)},$$

*provided that $V \asymp V_0$ and $V_\ell \asymp V_{0,\ell}$. In contrast, by Theorem B.1, the naive estimator $\hat{h}_\ell$—which only uses data from the $\ell$th group and is defined as in (13) with $\mathcal{F}_\ell$ from (52)—satisfies*

$$\|g_{0,\ell} - \hat{h}_\ell\|_{\mathcal{L}_2}^2 = O_\mathbb{P}\left(\frac{\widetilde{V}_\ell^{2w/(2+w)}}{(n\pi_\ell)^{2/(2+w)}}\right)$$

*ignoring logarithmic factors, and where $\widetilde{V}_\ell \asymp R(g_{0,\ell})$. Thus, the pretraining estimator $\hat{g}_\ell$ will achieve a faster rate of convergence than the naive estimator $\hat{h}_\ell$ provided that*

$$R\left(\sum_{k=1}^{L}\pi_k g_{0,k}\right) + R\left(g_{0,\ell} - \sum_{k=1}^{L}\pi_k g_{0,k}\right) << R(g_{0,\ell}),$$

*which can occur when the functions $g_{0,k}$ are relatively similar, and the averaging of these functions results in a representation with lower complexity, in terms of $R$, than that of $g_{0,\ell}$.*

We now apply Corollary D.3 to total variation–based estimators, commonly referred to as trend filtering (see (Mammen & Van De Geer, 1997; Tibshirani, 2014)). Towards that end, for a function $f : [0,1] \to \mathbb{R}$, we define its total variation as

$$\mathrm{TV}(f) = \sup_{M\in\mathbb{N}\,:\,M\geq 1} \mathrm{TV}(f, M), \tag{54}$$

where

$$\mathrm{TV}(f, M) = \sup_{0\leq a_1 \leq \dots \leq a_M \leq 1} \sum_{j=1}^{M-1} |f(a_j) - f(a_{j+1})|.$$

Then, we let $\mathcal{S}$ be the class of functions

$$\mathcal{S} := \{f : [0,1] \to \mathbb{R} \,:\, \mathrm{TV}(f^{(r-1)}) < \infty\}$$

where $r \in \mathbb{N}\backslash\{0\}$ is fixed and $f^{(r-1)}$ denotes the $(r-1)$th weak derivative of $f$. With this notation, we are now ready to state our next result.

**Corollary D.5.** *Let the $(r-1)$th order total variation of function be defined as $R(f) = \mathrm{TV}(f^{(r-1)})$ and the corresponding function classes $\mathcal{F}$ and $\mathcal{F}_\ell$ as in (52). Suppose that the notation and conditions of Corollary D.3 hold. Then the estimator $\hat{g}_\ell$ defined in (5) satisfies*

$$\|g_{0,\ell} - \hat{g}_\ell\|_{\mathcal{L}_2}^2 = O_\mathbb{P}\left(\frac{\max\{\mathcal{U}_n^2, \mathcal{A}_n, \mathcal{B}_n\}(V^{2/(2r+1)}\mathcal{A}_n^{2/(2r+1)} + V_\ell^{2/(2r+1)}\mathcal{B}_n^{2/(2r+1)})}{(n\pi_\ell)^{2r/(2r+1)}}\right) \tag{55}$$

*Remark* D.6. Let $\max\left\{\mathcal{U}_n, \mathcal{A}_n, \mathcal{B}_n, \|\bar{f}\|_\infty, \max_{\ell=1,\ldots,L}\|f_{0,\ell}\|_\infty, \|G_\ell\|_\infty\right\} = O(\text{poly}(\log n))$ just as in Remark D.4, then the rate achieved by trend filtering with pretraining—namely, the estimator $\hat{g}_\ell$—is

$$\left(V_0^{2/(2r+1)} + V_{0,\ell}^{2/(2r+1)}\right)(n\pi_\ell)^{-2r/(2r+1)},$$

while the rate attained by the usual trend filtering estimator is

$$\left(\text{TV}(g_{0,\ell}^{(r-1)})\right)^{2/(2r+1)}(n\pi_\ell)^{-2r/(2r+1)}.$$

Therefore, the pretraining estimator achieves a faster convergence rate than the usual trend filtering estimator whenever the functions

$$\sum_{k=1}^{L}\pi_k g_{0,k} \quad \text{and} \quad g_{0,\ell} - \sum_{k=1}^{L}\pi_k g_{0,k}$$

are substantially smoother than $g_{0,\ell}$, in terms of their total variation.

**Proof of Corollary D.3**

*Proof.* With the notation from Theorem B.1, we let $\mathcal{F}_{\mathcal{A}_n} := \{f_{\mathcal{A}_n}/(2\mathcal{A}_n) : f \in \mathcal{F}\}$. Then

$$\begin{aligned}
\log N\left(\delta, \mathcal{F}_{\mathcal{A}_n}, \|\cdot\|_n\right) &\leq \log N\left(\delta/V, \mathcal{F}_{\mathcal{A}_n}/V, \|\cdot\|_n\right) \\
&\leq \log N\left(\delta/V, B_R(1) \cap B_\infty(1), \|\cdot\|_n\right) \\
&\leq KV^w\delta^{-w} \\
&\leq K\mathcal{A}_n^wV^w\delta^{-w}
\end{aligned} \tag{56}$$

Hence, we define

$$\eta_n(\delta) := K\mathcal{A}_n^wV^w\delta^{-w},$$

and so by Assumption D.2 we proceed to check the conditions of Theorem B.1.

To verify (14), we observe that

$$\begin{aligned}
&\sum_{l=0}^{\infty}\sum_{l'=1}^{\infty}\exp\left(-C_1\eta_n(2^{-l-l'-1})\right) + \sum_{l=0}^{\infty}\exp\left(-C_2\eta_n(2^{-l-1})\right) \\
&= \sum_{l=0}^{\infty}\sum_{l'=1}^{\infty}\exp\left(-C_1K^w\mathcal{A}_n^wV^w2^{(l+l'+1)w}\right) + \sum_{l=0}^{\infty}\exp\left(-C_2K^w\mathcal{A}_n^wV^w2^{(l+1)w}\right) \\
&\leq \sum_{l=0}^{\infty}\sum_{l'=1}^{\infty}\exp\left(-C_1K^w\mathcal{A}_n^wV^w(l+l'+1)w\right) + \sum_{l=0}^{\infty}\exp\left(-C_2K^w\mathcal{A}_n^wV^w(l+1)w\right) \\
&= \sum_{l=0}^{\infty}\left[\exp\left(-C_1K^w\mathcal{A}_n^wV^w(l+1)w\right)\cdot\sum_{l'=1}^{\infty}\left[\exp\left(-C_1K^w\mathcal{A}_n^wV^ww\right)\right]^{l'}\right] + \\
&\quad \sum_{l=0}^{\infty}\left[\exp\left(-C_2K^w\mathcal{A}_n^wV^ww\right)\right]^{l+1} \\
&= \left[\exp\left(-C_1K^w\mathcal{A}_n^wV^ww\right)/\left[1-\exp\left(-C_1K^w\mathcal{A}_n^wV^w/w\right)\right]\right]^2 + \\
&\quad \exp\left(-C_2K^w\mathcal{A}_n^wV_1^ww\right)/\left[1-\exp\left(-C_2K^w\mathcal{A}_n^wV^ww\right)\right] \\
&\underset{n\to\infty}{\to} 0,
\end{aligned} \tag{57}$$

since by construction $\mathcal{A}_n \to \infty$. Thus, (14) holds.

To verify (15), notice that

$$\sup_{k\in\mathbb{N}}\sum_{k'=1}^{\infty}\frac{\eta_n(2^{-k-k'})}{2^{2k'}\eta_n(2^{-k})} = \sup_{k\in\mathbb{N}}\sum_{k'=1}^{\infty}\frac{1}{2^{(2-w)k'}} \leq 1. \tag{58}$$

Next, we construct a $\delta$ that is a critical radius. Towards that end, we notice that as in Lemma 6 of (Padilla et al., 2024a), it is enough to have $\delta$ satisfy

$$\frac{1}{\sqrt{n}}\int_{\delta^2/4}^{\delta}\sqrt{\log N\left(t/2, \mathcal{F}_{\mathcal{A}_n}, \|\cdot\|_n\right)}dt + \frac{\delta}{\sqrt{n}}\sqrt{\log(48/\delta^2)} \lesssim \delta^2. \tag{59}$$

However,

$$
\begin{aligned}
&\frac{1}{\sqrt{n}} \int_{\delta^2/4}^{\delta} \sqrt{\log N\left(t/2, \mathcal{F}_{\mathcal{A}_n}, \|\cdot\|_n\right)} dt \;+\; \frac{\delta}{\sqrt{n}} \sqrt{\log(48/\delta^2)} \\
&\leq \frac{1}{\sqrt{n}} \int_0^{\delta} \sqrt{\log N\left(t/2, \mathcal{F}_{\mathcal{A}_n}, \|\cdot\|_n\right)} dt \;+\; \frac{\delta_n}{\sqrt{n}} \sqrt{\log(48/\delta^2)} \\
&\lesssim \frac{1}{\sqrt{n}} \int_0^{\delta} \sqrt{\mathcal{A}_n^w V^w t^{-w}} dt \;+\; \frac{\delta}{\sqrt{n}} \sqrt{\log(48/\delta^2)} \\
&\lesssim \frac{V^{w/2} \mathcal{A}_n^{w/2} \delta^{1-w/2}}{\sqrt{n}} \;+\; \frac{\delta}{\sqrt{n}} \sqrt{\log(48/\delta^2)}.
\end{aligned}
\tag{60}
$$

Hence, (59) holds for $\delta > 0$ satisfying

$$
\delta^2 \;\asymp\; n^{-2/(2+w)} V^{2w/(2+w)} \mathcal{A}_n^{2w/(2+w)}.
\tag{61}
$$

Thus, $\delta$ is a critical radius for $\mathcal{F}_{\mathcal{A}_n}$. Moreover, in this case, we also have that

$$
\frac{\eta_n(\delta)}{n} \;\lesssim\; \delta^2.
$$

Therefore, from Theorem B.1, we obtain that

$$
\max\{\|\bar{f} - \hat{f}_{\mathcal{A}_n}\|_n^2, \|\bar{f} - \hat{f}_{\mathcal{A}_n}\|_{\mathcal{L}_2}^2\} \;=\; O_{\mathbb{P}}\left(r_n\right),
\tag{62}
$$

where

$$
r_n \;:=\; \frac{\max\{\mathcal{U}_n^2, \mathcal{A}_n\} \mathcal{A}_n^{2w/(2+w)} V^{2w/(2+w)}}{n^{2/(2+w)}}.
$$

Next, we analyze the second stage estimator using Theorem B.2. Let $\mathcal{F}_{\ell,\mathcal{B}_n} := \{f_{\mathcal{B}_n}/(2\mathcal{B}_n) : f \in \mathcal{F}_\ell\}$. Then, taking

$$
\eta_{\ell,n}(\delta) \;:=\; K \mathcal{B}_n^w V_\ell^w \delta^{-w},
$$

proceeding as in (56), we obtain that

$$
\max_{\frac{n\pi_\ell}{2} \leq n_\ell \leq 2n\pi_\ell} \;\sup_{\{x_i\}_{i \in I_\ell} \subset [0,1]^d} \log N(\delta, \mathcal{F}_{\ell,\mathcal{B}_n}, \|\cdot\|_{I_\ell}) \;\leq\; \eta_{\ell,n}(\delta).
$$

which shows (17). Moreover, we can verify (22) proceeding as in (57), and (23) as in (58).

Next, we proceed to find $\delta$ for which (19) and (20) hold. However, based on our previous calculations (19) and (20) are equivalent to

$$
\frac{1}{\sqrt{n\pi_\ell}} \int_{\delta^2/48}^{\delta} \sqrt{\mathcal{B}_n^w V_\ell^w t^{-w}} dt \;+\; \frac{\delta}{\sqrt{n\pi_\ell}} \sqrt{\log(48/\delta^2)} \;\lesssim\; \delta^2,
$$

and

$$
\frac{1}{\sqrt{n\pi_\ell}} \int_{\delta^2/48}^{\delta} \sqrt{\mathcal{A}_n^w V^w t^{-w}} dt \;+\; \frac{\delta}{\sqrt{n\pi_\ell}} \sqrt{\log(48/\delta^2)} \;\lesssim\; \delta^2,
$$

respectively. Hence, (19) and (20) hold for a choice of $\delta$ satisfying

$$
\delta^2 \;\asymp\; (n\pi_\ell)^{-2/(2+w)} \left[ V^{2w/(2+w)} \mathcal{A}_n^{2w/(2+w)} \;+\; V_\ell^{2w/(2+w)} \mathcal{B}_n^{2w/(2+w)} \right].
$$

Therefore, the claim follows from Theorem B.2.

$\square$

## Proof of Corollary D.5

*Proof.* The statement of Corollary D.5 follows from Corollary D.3, since Assumption D.1 is satisfied for the trend filtering regularizer. In particular, by Corollary 1 of (Sadhanala & Tibshirani, 2019), Assumption D.1 holds with $w = 1/r$ for $R(f) = \mathrm{TV}(f^{(r-1)})$, corresponding to $k = r - 1$ in their notation. $\square$

## D.2. Orthogonal Series Regression

We now apply Theorem 3.7 to orthogonal series estimator, instantiating the function classes $\mathcal{F}$ and $\mathcal{F}_\ell$ as Sobolev ellipsoids on $[0,1]^d$. This setting recovers, up to logarithmic factors, the two-task transfer learning rate established in (Wang et al., 2016), whose offset-based decomposition (Model 4.1 of (Wang et al., 2016)) parallels our two-stage procedure.

**Setup.** Following the set up in (Wang et al., 2016), we specialize the model in (1) to $L = 2$ groups, with group $\ell = 1$ playing the role of the target. The data consist of $n$ i.i.d. copies of $(X, Y, Z)$, partitioned into the target group $\mathcal{I}_1 := \{i : z_i = 1\}$ of size $n_1 := |\mathcal{I}_1|$ and the source group $\mathcal{I}_2 := \{i : z_i = 2\}$ of size $n - n_1$. Following (2), the target-group conditional mean function decomposes as

$$g_{0,1}(x) := \mathbb{E}[Y \mid X = x, Z = 1] = \bar{f}(x) + G_1(x),$$

where

$$\bar{f}(x) := \mathbb{E}[Y \mid X = x] \quad \text{and} \quad G_1(x) := g_{0,1}(x) - \bar{f}(x)$$

denote the overall mean and the target-specific offset, respectively, as introduced in Section 2. Throughout this subsection, we work under the independent two-stage estimation setting of Theorem 3.7: the Stage-1 estimator $\hat{f}$ is constructed using a sample independent of the one used in Stage 2.

According to (Wang et al., 2016), let $L_2([0,1]^d)$ denote the space of square-integrable functions on $[0,1]^d$ equipped with the inner product $\langle f, g \rangle = \int_{[0,1]^d} f(x)g(x)\, dx$. Let $\{\varphi_j\}_{j \in \mathbb{Z}}$ be an orthonormal basis of $L_2([0,1])$ with $\sup_j \|\varphi_j\|_\infty \leq C_\varphi$ for some constant $C_\varphi > 0$. For example, we can consider the cosine basis in the following:

$$\varphi_j(x) = \begin{cases} 1, & j = 0, \\ \sqrt{2}\, \cos(j\pi x), & j \geq 1, \end{cases}$$

where the index $j$ orders the basis functions by frequency and $\sup_j \|\varphi_j\|_\infty \leq C_\varphi$ with $C_\varphi = \sqrt{2}$. The tensor product basis is defined as $\{\varphi_\alpha\}_{\alpha \in \mathbb{Z}^d}$, where

$$\varphi_\alpha(x) = \prod_{w=1}^{d} \varphi_{\alpha_w}(x_w), \qquad \alpha = (\alpha_1, \ldots, \alpha_d) \in \mathbb{Z}^d,$$

forms an orthonormal basis of $L_2([0,1]^d)$, so every $f \in L_2([0,1]^d)$ admits the expansion $f = \sum_\alpha a_\alpha(f)\, \varphi_\alpha$ with $a_\alpha(f) := \langle \varphi_\alpha, f \rangle$.

**Definition D.7** (Sobolev Ellipsoid). Fix smoothness $s > 0$, radius $A > 0$, and uniform bound $f_{\max} > 0$, where $A$ and $f_{\max}$ are taken to be common across all smoothness levels $s$ for simplicity. The (isotropic) Sobolev ellipsoid on $[0,1]^d$ is

$$\mathcal{W}^s := \left\{ f \in L_2([0,1]^d) : \sum_{\alpha \in \mathbb{Z}^d} a_\alpha(f)^2\, \kappa_s^2(\alpha) \leq A^2, \; \|f\|_\infty \leq f_{\max} \right\},$$

where

$$\kappa_s^2(\alpha) := \sum_{w=1}^{d} |\alpha_w|^{2s}.$$

The quantity $\kappa_s^2(\alpha)$ assigns larger weights to higher-frequency basis functions. Hence the constraint $\sum_{\alpha \in \mathbb{Z}^d} a_\alpha(f)^2 \kappa_s^2(\alpha) \leq A^2$ forces the coefficients $a_\alpha(f)$ corresponding to high-frequency basis functions to decay sufficiently fast. In this sense, larger values of $s$ correspond to smoother functions.

For each smoothness level $s > 0$ and threshold $t > 0$, define the set

$$M_s(t) := \left\{ \alpha \in \mathbb{Z}^d : \kappa_s^2(\alpha) \leq t \right\},$$

which collects all multi-indices whose Sobolev weight does not exceed $t$. We then write $J := |M_s(t)|$ for the cardinality of $M_s(t)$, which stands for the number of basis functions retained at threshold $t$.

**Function Classes.** We assume that both the overall mean function and the target offset belong to Sobolev ellipsoids:

$$\bar{f} \in \mathcal{W}^\tau, \qquad G_1 \in \mathcal{W}^\nu.$$

The smoothness levels $\tau, \nu > 0$ are allowed to differ. In particular, $\nu > \tau$ corresponds to the regime in which the offset $G_1$ is smoother than the overall mean $\bar{f}$, which is the setting where transfer learning is expected to be most beneficial.

Now consider the Stage-1 estimator. Define the Stage-1 threshold $t_0 > 0$ and let $J_0 := |M_\tau(t_0)|$ denote the number of basis functions retained. For each $\alpha \in M_\tau(t_0)$, let $\theta_\alpha \in \mathbb{R}$ be the scalar coefficient associated with the basis function $\varphi_\alpha$; collecting these $J_0$ coefficients into the vector $\theta = (\theta_\alpha)_{\alpha \in M_\tau(t_0)} \in \mathbb{R}^{J_0}$, we estimate the Stage-1 target function $\bar{f}$ using $\hat{f} \in \mathcal{F}_{t_0}$, where

$$\mathcal{F}_{t_0} := \Big\{ \sum_{\alpha \in M_\tau(t_0)} \theta_\alpha \varphi_\alpha : \ \|\theta\|_2 \le R_0 \Big\},$$

with a fixed $R_0 \in \mathbb{R}_+$, and

$$\hat{f} := \arg \min_{f \in \mathcal{F}_{t_0}} \sum_{i=1}^{n} \big(y_i - f(x_i)\big)^2.$$

Here $\mathcal{F}_{t_0}$ consists of all linear combinations of the $J_0$ lowest-frequency tensor-product basis functions selected under the Sobolev weight corresponding to the smoothness $\tau$. Analogously, for the Stage-2 estimator, let $t_1 > 0$ and $J_1 := |M_\nu(t_1)|$. For each $\alpha \in M_\tau(t_1)$, let $\gamma_\alpha \in \mathbb{R}$ be the scalar coefficient associated with the basis function $\varphi_\alpha$; collecting these $J_1$ coefficients into the vector $\gamma = (\gamma_\alpha)_{\alpha \in M_\tau(t_1)} \in \mathbb{R}^{J_1}$, we estimate the Stage-2 target function $G_1$ using $\hat{f}_1 \in \mathcal{G}_{t_1}$, where

$$\mathcal{G}_{t_1} := \Big\{ \sum_{\alpha \in M_\nu(t_1)} \gamma_\alpha \varphi_\alpha : \ \|\gamma\|_2 \le R_1 \Big\},$$

with a fixed $R_1 \in \mathbb{R}_+$, and

$$\hat{f}_1 := \arg \min_{f \in \mathcal{G}_{t_1}} \sum_{i:\, z_i = 1} \big(y_i - \hat{f}_{\mathcal{A}_n}(x_i) - f(x_i)\big)^2,$$

where we focus on group 1 for simplicity. The final estimator would be

$$\hat{g}_1(x) := \hat{f}_{\mathcal{A}_n}(x) + \hat{f}_{1,\mathcal{B}_n}(x).$$

Now we are ready to state our results.

**Corollary D.8.** *Suppose that $\mathbb{P}(Z = \ell \mid X = x) = \pi_\ell \in (0,1)$ for all $x \in [0,1]$ and $\ell \in \{1, 2\}$, that $\bar{f} \in \mathcal{W}^\tau$ and $G_1 \in \mathcal{W}^\nu$, and that $2\nu \le d$. Assume further that the two stages are estimated on independent samples as in Theorem 3.7, with truncation levels $\mathcal{A}_n = \mathcal{B}_n = f_{\max}$. Suppose that $\mathcal{U}_n$ grows at most in $O(\mathrm{poly}(\log n))$. With*

$$J_0 \asymp \Big( \frac{n}{\log n} \Big)^{d/(2\tau + d)}, \qquad J_1 \asymp \Big( \frac{n_1}{\log n} \Big)^{d/(2\nu + d)},$$

*and $n_1 \asymp n\pi_1$, the transfer learning estimator $\hat{g}_1$ defined in (5) satisfies*

$$\|g_{0,1} - \hat{g}_1\|_{\mathcal{L}_2}^2 = O_\mathbb{P}\Big( n^{-\frac{2\tau}{2\tau + d}} + n_1^{-\frac{2\nu}{2\nu + d}} \Big),$$

*up to logarithmic factors.*

*Remark* D.9. The two terms in the rate correspond, respectively, to the Stage-1 error of estimating $\bar{f}$ from the full sample of size $n$, and the Stage-2 error of estimating the offset $G_1$ from the target-group sample of size $n_1$. To compare with the simplified rate of (Wang et al., 2016), parameterize $n = n_1^\lambda$ for some $\lambda \ge 1$ — in their two-task setting, $\lambda$ captures the relative abundance of source data, with $\lambda = 1$ corresponding to balanced source and target sample sizes and $\lambda \to \infty$ corresponding to a much larger source pool. Substituting yields

$$\|g_{0,1} - \hat{g}_1\|_{\mathcal{L}_2}^2 = O_\mathbb{P}\Big( n_1^{-\frac{2\lambda\tau}{2\tau + d}} + n_1^{-\frac{2\nu}{2\nu + d}} \Big)$$

up to logarithmic factors, which matches the simplified rate stated immediately after Theorem 4.3 of (Wang et al., 2016). Consistent with Remark 3.5, the pretraining estimator achieves a faster rate than the naive estimator using only target-group data — whose rate is $(\log n_1/n_1)^{2\sigma/(2\sigma+d)}$ when $g_{0,1} \in \mathcal{W}^\sigma$ — whenever $\bar{f}$ and $G_1$ are smoother than $g_{0,1}$ itself (i.e., $\tau, \nu > \sigma$, with $\lambda$ large enough).

## D.3. Proof of Corollary D.8

We first establish the approximation rate for $d$-dimensional Sobolev ellipsoids by sieve classes. At the beginning, we state a standard result for approximating a generic function $f$ in a one-dimensional Sobolev ellipsoid. The purpose of the lemma is to bound the projection error incurred by keeping only the first $J$ low-frequency basis functions.

**Lemma D.10** (One-Dimensional Approximation by Sobolev Sieves (Lemma 8.4 in (Wasserman, 2006))). *Consider $f \in \mathcal{W}^s$. Let $d = 1$, so that the multi-index $\alpha$ reduces to a scalar index $\alpha \in \mathbb{Z}$, and $\kappa_s^2(\alpha) = |\alpha|^{2s}$. Let $M_s(t) \subset \mathbb{Z}$ denote the indices of the $J$ one-dimensional basis functions having $J := |M_\tau(t)|$, and define the projection*

$$f_J := \sum_{\alpha \in M_s(t)} a_\alpha(f) \varphi_\alpha.$$

*If $\sum_{\alpha \in \mathbb{Z}} |\alpha|^{2s} a_\alpha(f)^2 \leq C$, then*

$$\|f - f_J\|_{\mathcal{L}_2}^2 = \sum_{\alpha \notin M_s(t)} a_\alpha(f)^2 \leq C J^{-2s}.$$

Now we extend this result to $d$-dimension.

**Proposition D.11.** *Let $f : [0,1]^d \to \mathbb{R}$ be a function in $\mathcal{W}^s$, and let $M_s(t)$ be the set of indices defined earlier, which contains the $J$ multi-indices with the Sobolev weight $\kappa_s^2(\alpha)$ truncated by $t$. For all $J$ sufficiently large, it holds that*

$$\inf_{\tilde{f} \in \mathcal{F}_t} \|f - \tilde{f}\|_{\mathcal{L}_2}^2 = \sum_{\alpha \notin M_s(t)} a_\alpha(f)^2 \leq C \cdot J^{-2s/d},$$

*with $\alpha \in \mathbb{Z}^d$ for some constant $C$ not depending on $J$.*

*Proof.* For $\alpha \notin M_s(t)$, $\kappa_s^2(\alpha) > t$, so

$$\sum_{\alpha \notin M_s(t)} a_\alpha(f)^2 \leq t^{-1} \sum_{\alpha \in \mathbb{Z}^d} a_\alpha(f)^2 \kappa_s^2(\alpha) \leq \frac{A^2}{t}. \tag{63}$$

where $M_s(t) = \{\alpha : \sum_{w=1}^{d} |\alpha_w|^{2s} \leq t\}$, which forces $|\alpha_w| \leq t^{1/(2s)}$ for each dimension $w$. Hence $J \leq \left(2\lfloor t^{1/(2s)} \rfloor + 1\right)^d \leq \left(3t^{1/(2s)}\right)^d$ for $t^{1/(2s)} \geq 1$, giving

$$t \geq \frac{J^{2s/d}}{3^{2s}}. \tag{64}$$

The case $t^{1/(2s)} < 1$ is degenerate: since $\kappa_s^2(\alpha) \geq 1$ for every nonzero integer multi-index $\alpha$, it forces $M_s(t) = \{(0, \ldots, 0)\}$ and $J = 1$, which is excluded in the regime of interest where $J \to \infty$. Combining (63) and (64) yields the claim. $\square$

*Remark* D.12. The proof of Proposition D.11 establishes the upper bound $J \lesssim t^{d/(2s)}$. A matching lower bound $J \gtrsim t^{d/(2s)}$ follows analogously by counting the integer points in a box contained in $M_s(t)$. Let $\Lambda := \lfloor (t/d)^{1/(2s)} \rfloor$. If $\alpha \in \mathbb{Z}^d$ satisfies $|\alpha_w| \leq \Lambda$ for all $w = 1, \ldots, d$, then

$$\sum_{w=1}^{d} |\alpha_w|^{2s} \leq d\Lambda^{2s} \leq d \cdot \frac{t}{d} = t.$$

Hence $\{-\Lambda, \ldots, \Lambda\}^d \subseteq M_s(t)$, and therefore

$$J = |M_s(t)| \geq (2\Lambda + 1)^d = \left(2 \left\lfloor (t/d)^{1/(2s)} \right\rfloor + 1\right)^d.$$

For $(t/d)^{1/(2s)} \geq 1$, this implies $J \gtrsim t^{d/(2s)}$. Combining the two bounds gives $J \asymp t^{d/(2s)}$. For simplicity we track the dimension $J$ rather than the truncation $t$.

**Approximation Error.** By Proposition D.11,

$$\phi_n := \inf_{f \in \mathcal{F}_{t_0}} \|\bar{f} - f\|_{\mathcal{L}_2}^2 \lesssim J_0^{-2\tau/d}, \qquad \phi_{1,n} := \inf_{g \in \mathcal{G}_{t_1}} \|G_1 - g\|_{\mathcal{L}_2}^2 \lesssim J_1^{-2\nu/d}.$$

**Empirical Covering Entropy.** Since $\mathcal{F}_{t_0}$ and $\mathcal{G}_{t_1}$ are finite-dimensional linear spans of bounded basis functions, each function is uniquely parameterized by its coefficient vector. Moreover, for any sample $\{x_i\}_{i=1}^n$ and any coefficient vectors $\theta, \theta'$,

$$\|f_\theta - f_{\theta'}\|_n^2 = \frac{1}{n}\sum_{i=1}^n \Big(\sum_{\alpha \in M_\tau(t_0)} (\theta_\alpha - \theta'_\alpha)\varphi_\alpha(x_i)\Big)^2 \le \frac{1}{n}\sum_{i=1}^n \Big(\sum_{\alpha \in M_\tau(t_0)} (\theta_\alpha - \theta'_\alpha)^2\Big)\Big(\sum_{\alpha \in M_\tau(t_0)} \varphi_\alpha(x_i)^2\Big) \le J_0\, C_\varphi^{2d}\, \|\theta - \theta'\|_2^2,$$

where the first inequality follows from the Cauchy–Schwarz inequality and the second uses the uniform bound $\sup_j \|\varphi_j\|_\infty \le C_\varphi$. Equivalently, $\|f_\theta - f_{\theta'}\|_m \le C_\varphi^{2d}\sqrt{J_0}\,\|\theta - \theta'\|_2$. The same bound holds for $\mathcal{G}_{t_1}$ with $J_1$ in place of $J_0$. Consequently, any $(\delta/(C_\varphi^{2d}\sqrt{J_0}))$-cover of the Euclidean ball $B_2^{J_0}(R_0) := \{\theta \in \mathbb{R}^{J_0} : \|\theta\|_2 \le R_0\}$ in $\mathbb{R}^{J_0}$ induces a $\delta$-cover of $\mathcal{F}_{t_0}$ under $\|\cdot\|_n$:

$$N(\delta, \mathcal{F}_{t_0}, \|\cdot\|_n) \le N\Big(\delta/(C_\varphi^{2d}\sqrt{J_0}),\ B_2^{J_0}(R_0),\ \|\cdot\|_2\Big).$$

Combined with the standard covering bound $N(\eta, B_2^J(R), \|\cdot\|_2) \le (3R/\eta)^J$ (Corollary 4.2.11 of (Vershynin, 2026)), we obtain

$$\log N(\delta, \mathcal{F}_{t_0}, \|\cdot\|_m) \lesssim J_0 \log(C\sqrt{J_0}/\delta), \quad \log N(\delta, \mathcal{G}_{t_1}, \|\cdot\|_m) \lesssim J_1 \log(C\sqrt{J_1}/\delta). \tag{65}$$

**Verification of Conditions (14) and (15) for Stage 1.** By the entropy bound established above, we may take the majorant

$$\eta_n(\delta) := c_0\, J_0 \log(C\sqrt{J_0}/\delta),$$

under our assumption $\mathcal{A}_n = O(1)$, which satisfies $\log N(\delta, \mathcal{F}_{t_0, \mathcal{A}_n}, \|\cdot\|_n) \le \eta_n(\delta)$ as required by Theorem B.1.

*Step 1: Verification of (14).* Under increasing $\sqrt{J_0}$ with $n$, $\log(C\sqrt{J_0}) > 0$, so for any $k \ge 0$ and $k' \ge 1$,

$$\eta_n(2^{-k-k'-1}) = c_0 J_0\big[(k+k'+1)\log 2 + \log(C\sqrt{J_0})\big] \ge c_0 J_0 (k+k'+1)\log 2.$$

Hence for any $C_1 > 0$,

$$\sum_{k=0}^\infty \sum_{k'=1}^\infty \exp\big(-C_1\, \eta_n(2^{-k-k'-1})\big) \le \sum_{k=0}^\infty \sum_{k'=1}^\infty \rho^{k+k'+1} = \rho \cdot \frac{1}{1-\rho} \cdot \frac{\rho}{1-\rho} = \frac{\rho^2}{(1-\rho)^2} \to 0$$

as $J_0 \to \infty$, where $\rho := \exp(-C_1 c_0 J_0 \log 2) \in (0, 1)$. The second summation in (14) is bounded analogously:

$$\sum_{k=0}^\infty \exp\big(-C_2\, \eta_n(2^{-k-1})\big) \le \sum_{k=0}^\infty \rho'^{k+1} = \frac{\rho'}{1-\rho'} \to 0,$$

where $\rho' := \exp(-C_2 c_0 J_0 \log 2) \in (0, 1)$. Together with the noise tail assumption $\mathbb{P}(\|\epsilon\|_\infty > \mathcal{U}_n) \to 0$, condition (14) is verified.

*Step 2: Verification of (15).* For $k \ge 1$ and $k' \ge 1$, since $\log(C\sqrt{J_0}) > 0$,

$$\frac{\eta_n(2^{-k-k'})}{\eta_n(2^{-k})} = \frac{(k+k')\log 2 + \log(C\sqrt{J_0})}{k \log 2 + \log(C\sqrt{J_0})} \le \frac{k+k'}{k}.$$

Therefore

$$\sup_{k\ge 1} \sum_{k'=1}^\infty \frac{\eta_n(2^{-k-k'})}{2^{2k'}\eta_n(2^{-k})} \le \sup_{k\ge 1} \sum_{k'=1}^\infty \frac{k+k'}{2^{2k'} k} = \sup_{k\ge 1} \left[\sum_{k'=1}^\infty \frac{1}{4^{k'}} + \frac{1}{k}\sum_{k'=1}^\infty \frac{k'}{4^{k'}}\right] = \sup_{k\ge 1}\left[\frac{1}{3} + \frac{4}{9k}\right] = \frac{7}{9} < 1,$$

where we used the geometric-series identities $\sum_{k'\ge 1} 4^{-k'} = 1/3$ and $\sum_{k'\ge 1} k' \cdot 4^{-k'} = 4/9$. The case $k = 0$ is handled identically. This verifies (15).

**Verification of Condition (19) for Stage 2.** Under the independent two-stage estimation setting of Theorem 3.7, condition (20) is replaced by the independence assumption between the data used in Stages 1 and 2, so only condition (19) on the Stage-2 requires verification.

Consider $m \in [n\pi_\ell/2, 2n\pi_\ell]$. By the entropy bound for $\mathcal{G}_{t_1}$ established above,

$$\sqrt{\log N(t/4, \mathcal{G}_{t_1, \mathcal{B}_n}, \|\cdot\|_m)} \lesssim \sqrt{J_1 \log(C\sqrt{J_1}/t)}.$$

Therefore, the Stage-2 complexity integral satisfies

$$\frac{1}{\sqrt{m}} \int_{\delta^2/48}^{\delta} \sqrt{\log N(t/4, \mathcal{G}_{t_1, \mathcal{B}_n}, \|\cdot\|_m)}\, dt \lesssim \frac{\sqrt{J_1}}{\sqrt{m}} \int_{\delta^2/48}^{\delta} \sqrt{\log(C\sqrt{J_1}/t)}\, dt.$$

Since $t \mapsto \sqrt{\log(C\sqrt{J_1}/t)}$ is decreasing on $(0, C\sqrt{J_1})$,

$$\int_{\delta^2/48}^{\delta} \sqrt{\log(C\sqrt{J_1}/t)}\, dt \lesssim \delta \sqrt{\log(C\sqrt{J_1}/\delta^2)}.$$

Hence

$$\frac{1}{\sqrt{m}} \int_{\delta^2/48}^{\delta} \sqrt{\log N(t/4, \mathcal{G}_{t_1, \mathcal{B}_n}, \|\cdot\|_m)}\, dt + \frac{\delta}{\sqrt{m}} \sqrt{c_1 \log(48/\delta^2)} \lesssim \frac{\delta}{\sqrt{m}} \left( \sqrt{J_1 \log(C\sqrt{J_1}/\delta^2)} + \sqrt{\log(1/\delta^2)} \right).$$

Thus condition (19) holds whenever $\delta^2 \gtrsim J_1 \log(C\sqrt{J_1}/\delta^2)/(n\pi_\ell)$. Under $n_1 \asymp n\pi_1$, the choice

$$\delta^2 \asymp \frac{J_1 \log(J_1 \cdot n)}{n_1}$$

suffices.

**Combining the Two Stages.** By Theorem B.1, applied to the Stage-1 sieve class $\mathcal{F}_{t_0}$, the Stage-1 error satisfies

$$r_n \lesssim J_0^{-2\tau/d} + \frac{J_0 \log(J_0 n)}{n}$$

up to polylogarithmic factors. Applying Theorem 3.7 with $\phi_{1,n} \lesssim J_1^{-2\nu/d}$ and effective target sample size $n_1 \asymp n\pi_1$,

$$\|g_{0,1} - \hat{g}_1\|_{\mathcal{L}_2}^2 = O_{\mathbb{P}} \left( J_0^{-2\tau/d} + \frac{J_0 \log(J_0 n)}{n} + J_1^{-2\nu/d} + \frac{J_1 \log(J_1 n)}{n_1} + \sqrt{\frac{\log n}{n_1}} \right)$$

up to polylogarithmic factors, where the last term is absorbed by the Stage-2 rate when $2\nu \leq d$. Balancing bias and variance in each stage yields the choices of $J_0$ and $J_1$ stated in Corollary D.8, and substituting back gives the stated rate.

# E. Illustrative Examples to Interpret the Theoretical Results

We provide three illustrative examples to complement the theoretical results in the main text. The first example shows that the two-stage transfer learning procedure can yield estimation targets that are considerably simpler than direct estimation of the group-specific functions. The second example instantiates the convergence rate $\phi_n + \phi_{\ell,n}$ under a concrete compositional structure with 10-dimensional input. The third example compares convergence rates across four estimators under different regimes of the group proportion $\pi_\ell$, illustrating the advantage of NN with TL over classical methods.

## E.1. Two-Stage Targets Can Be Simpler Than Direct Estimation

We provide a concrete example illustrating that, under the hierarchical compositional structure, both the shared and group-specific functions can individually be simple, yet their sum is complex, thus yielding simpler estimation targets in the

two-stage transfer learning procedure. Consider $L = 2$ groups with proportions $\pi_1 + \pi_2 = 1$ and group-specific regression functions

$$g_{0,1}(x_1, \ldots, x_5) = f_0(x_1, x_2) + f_{0,1}(x_3, x_4, x_5), \quad g_{0,2}(x_1, \ldots, x_5) = f_0(x_1, x_2) + f_{0,2}(x_3, x_4, x_5).$$

Each $g_{0,\ell}$ depends on all 5 variables, so direct estimation incurs rates governed by dimension 5. Now suppose the group deviation functions satisfy

$$\pi_1 f_{0,1}(x_3, x_4, x_5) + \pi_2 f_{0,2}(x_3, x_4, x_5) \approx c$$

for some constant $c$. Then the first-stage target

$$\bar{f} = f_0(x_1, x_2) + \pi_1 f_{0,1} + \pi_2 f_{0,2} \approx f_0(x_1, x_2) + c$$

effectively depends only on 2 variables, yielding faster first-stage convergence rates. The second-stage offset for group 2,

$$G_2 = \pi_2(f_{0,2} - f_{0,1}),$$

depends on 3 variables and becomes nearly zero as $\pi_2 \to 0$, making second-stage estimation easy.

### E.2. Concrete Instantiation of Convergence Rates

We provide a concrete example to illustrate the final convergence rate $\phi_n + \phi_{\ell,n}$ under specific settings. Consider $L = 2$ groups with proportions $\pi_1 + \pi_2 = 1$ and 10-dimensional input $x \in \mathbb{R}^{10}$. Both groups share the same compositional structure as illustrated in Figures 2 and 3, where the shared components $h_1^{(1)}, h_2^{(1)}$ have input dimension $K = 2, 3, 3$ and smoothness $p = 1.5, 1.5, 2.5$, respectively. The two groups differ only in the group-specific component $h_{3,\text{group}_\ell}^{(1)}$ (in red color), with $K = 2, p = 1.5$ for both groups.

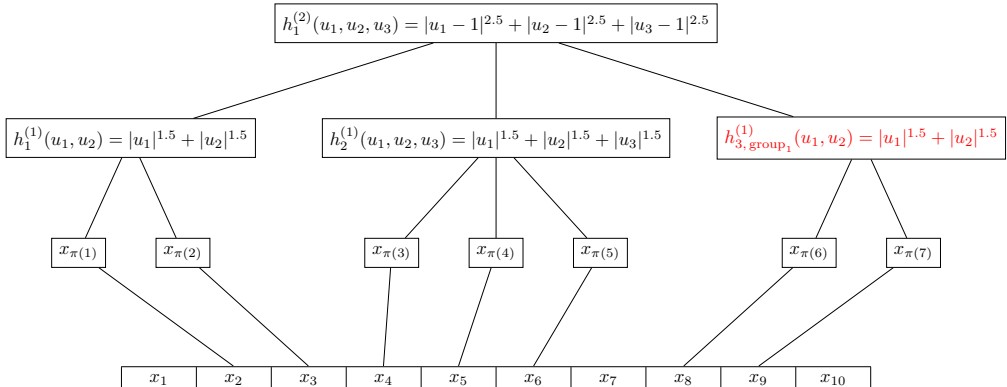

*Figure 2.* Illustration of the estimated group mean function $g_{0,1}(x)$ for Group 1, where $x \in \mathbb{R}^{10}$.

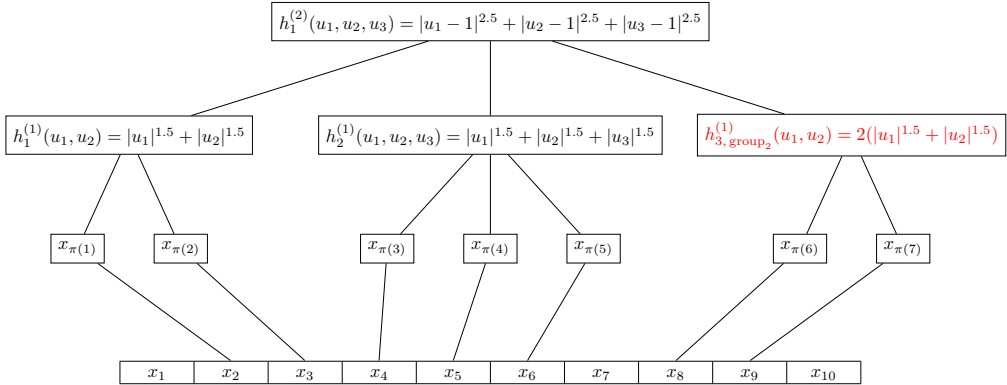

*Figure 3.* Illustration of the estimated group mean function $g_{0,2}(x)$ for Group 2, where $x \in \mathbb{R}^{10}$.

Estimating $\bar{f} = \pi_1 g_{0,1} + \pi_2 g_{0,2}$ in the first stage preserves the same compositional structure as $g_{0,1}$ and $g_{0,2}$, giving an upper bound

$$\phi_n = \max_{(p,K) \in \{(1.5, 2), (1.5, 3), (2.5, 3)\}} n^{-2p/(2p+K)} = n^{-1/2}.$$

In the second stage, the offset $G_1 = \pi_2(g_{0,1} - g_{0,2})$ cancels all shared components, reducing to a 2-layer model with a single branch, giving an upper bound

$$\phi_{1,n} = \max_{(p,K) \in \{(1.5, 2), (2.5, 1)\}} (n\pi_1)^{-2p/(2p+K)} = (n\pi_1)^{-3/5}.$$

The overall rate $n^{-1/2} + (n\pi_1)^{-3/5}$ depends only on the intrinsic dimensions of the compositional structure, not on the ambient dimension $d = 10$. In contrast, standard estimators that treat all 10 dimensions yield the slower rate $n^{-3/13} + (n\pi_1)^{-3/13}$, demonstrating the advantage of our method in overcoming the curse of dimensionality.

### E.3. Benefit of Transfer Learning with DNNs over Classical Estimators

We illustrate the benefit of transfer learning (TL) with DNNs in comparison with classical estimators, such as transfer learning with kernel smoothing (Du et al., 2017), through a simple example applying Corollary 3.8. Let $X \in [0, 1]^{10}$ and $\ell \in \{1, 2, 3\}$ denote the group index. For simplicity, suppose that $\bar{\pi}_\ell = \underline{\pi}_\ell = \pi_\ell$ for all $\ell$. We observe

$$Y_i = g_{0,\ell}(X_i) + \varepsilon_{\ell,i},$$

and consider the additive decomposition

$$g_{0,\ell}(x) = f_0(x) + f_{0,\ell}(x), \quad \ell = 1, 2, 3,$$

with shared component $f_0(x) = |x_1 + \cdots + x_{10} - 1/2|$ and group-specific components

$$f_{0,1}(x) = |x_1+x_2+x_3+x_4-1/2|^{3/2}, \quad f_{0,2}(x) = 2|x_1+x_2+x_3+x_4-1/2|^{3/2}, \quad f_{0,3}(x) = 3|x_1+x_2+x_3+x_4-1/2|^{3/2}.$$

Each group may further have its own noise distribution; for example, $\varepsilon_{\ell,i} \sim \mathcal{N}(0, \ell \cdot \sqrt{X_i^{(\ell)}})$, where $X_i^{(\ell)}$ denotes the $\ell$-th coordinate of $X_i$. Since each $g_{0,\ell}$ and the distribution of $\varepsilon_\ell$ can differ across groups. We now use the example above to compare convergence rates under different $\pi_\ell$ with directly comparable classical estimators under the similar offset TL framework, and illustrate the novelty of our results.

**(a) Without Transfer Learning.** In this example, the smoothness of $g_{0,\ell}$ is $p = 1$ with intrinsic dimensionalities $K = 10$. Training each group separately on $n\pi_\ell$ samples, classical estimators must use all $d = 10$ dimensions, yielding rate

$$(n\pi_\ell)^{-\frac{2p}{2p+d}} = (n\pi_\ell)^{-1/6}.$$

Direct NN estimation without transfer learning exploits intrinsic dimensionality $K = 10$, also giving

$$(n\pi_\ell)^{-\frac{2p}{2p+K}} = (n\pi_\ell)^{-1/6}.$$

This is a deliberately chosen example where direct NN estimation without TL cannot improve upon the classical rate, specifically constructed to motivate the benefit of TL with NN in the subsequent comparison. In most settings, NN estimation does outperform classical methods by exploiting intrinsic dimensionality; in particular, when $d > 10$ but $K = 10$, the NN achieves a strictly faster rate by overcoming the curse of dimensionality. (We note that the same function may admit different hierarchical compositions with different intrinsic dimensionalities, since our upper bounds apply at the function class level rather than to a specific function instance. The composition with intrinsic dimension $K = 10$ chosen here is just for illustrative purposes.)

**(b) With Transfer Learning.** The first-stage target is

$$\bar{f}(x) = |x_1 + \cdots + x_{10} - 1/2| + (\pi_1 + 2\pi_2 + 2\pi_3) \cdot |x_1 + x_2 + x_3 + x_4 - 1/2|^{3/2},$$

which has smoothness $p = 1$ and intrinsic dimension $K = 10$. The second-stage offset (taking group 1 as an example) is

$$G_1(x) = (1 - \pi_1 - 2\pi_2 - 3\pi_3) \cdot |x_1 + x_2 + x_3 + x_4 - 1/2|^{3/2},$$

which has smoothness $p = 3/2$ and intrinsic dimension $K = 4$, hence simpler than $g_{0,\ell}$ and $\bar{f}$. Under this setting, classical estimators, which require all 10 dimensions for both $\bar{f}$ and $G_1$, yield rate

$$n^{-1/6} + (n\pi_1)^{-3/13}.$$

Our Theorem 3.7, by exploiting the intrinsic dimensionality of $G_1$, yields the faster rate

$$n^{-1/6} + (n\pi_1)^{-3/7}.$$

The four regimes of convergence rates for all four estimators as $\pi_1$ varies are summarized in the table below. Specifically:

1. When $\pi_1 \asymp 1$, all methods achieve $n^{-1/6}$ and TL provides no additional gain.

2. When $n^{-5/18} \ll \pi_1 \ll 1$, both classical and NN methods with TL achieve the faster rate $n^{-1/6}$, while classical and NN without TL are slower at $(n\pi_1)^{-1/6}$.

3. When $n^{-11/18} \lesssim \pi_1 \lesssim n^{-5/18}$, NN with TL still achieves $n^{-1/6}$ while the classical TL rate degrades to $(n\pi_1)^{-3/13}$; for instance, at $\pi_1 \asymp n^{-11/18}$, our rate is $n^{-1/6}$ versus $n^{-7/78}$ for classical TL.

4. When $n^{-1} \ll \pi_1 \ll n^{-11/18}$, NN with TL at $(n\pi_1)^{-3/7}$ remains strictly faster than classical TL at $(n\pi_1)^{-3/13}$. The condition $n^{-1} \ll \pi_1$ follows from the group probability requirement in Theorem B.2.

This example clearly illustrates how a small $\pi_1$ can drastically worsen the classical rate while NN with TL remains robust.

| | Classical w/o TL | NN w/o TL | Classical w/ TL | NN w/ TL |
|---|---|---|---|---|
| $\pi_1 \asymp 1$ | $n^{-\frac{1}{6}}$ | $n^{-\frac{1}{6}}$ | $n^{-\frac{1}{6}}$ | $n^{-\frac{1}{6}}$ |
| $n^{-\frac{5}{18}} \ll \pi_1 \ll 1$ | $(n\pi_1)^{-\frac{1}{6}}$ | $(n\pi_1)^{-\frac{1}{6}}$ | $n^{-\frac{1}{6}}$ | $n^{-\frac{1}{6}}$ |
| $n^{-\frac{11}{18}} \lesssim \pi_1 \lesssim n^{-\frac{5}{18}}$ | $(n\pi_1)^{-\frac{1}{6}}$ | $(n\pi_1)^{-\frac{1}{6}}$ | $(n\pi_1)^{-\frac{3}{13}}$ | $n^{-\frac{1}{6}}$ |
| $n^{-1} \ll \pi_1 \ll n^{-\frac{11}{18}}$ | $(n\pi_1)^{-\frac{1}{6}}$ | $(n\pi_1)^{-\frac{1}{6}}$ | $(n\pi_1)^{-\frac{3}{13}}$ | $(n\pi_1)^{-\frac{3}{7}}$ |

*Table 5.* Convergence rates under different regimes of $\pi_1$.

# F. Details of Numerical Experiments

## F.1. Complete Numerical Results

In Section 4 of the main text, we selectively present four scenarios from our simulation experiments. Scenarios 1 and 2 correspond to low-dimensional settings, while Scenarios 3 and 4 are high-dimensional. In this supplementary section, we report results for four additional low-dimensional scenarios (Extra Scenarios 1–4) to provide a more comprehensive evaluation of method performance in low-dimensional settings.

**Extra Scenario 1.** This scenario features a shared additive model with simple group-specific shifts. Let the input vector be $q = (q_1, \ldots, q_{10}) \overset{\text{i.i.d.}}{\sim} \text{Unif}([0,1]^{10})$, holding for Extra Scenario 1–4. With a group-varying scale $w_z = \frac{L-z+1}{L(L+1)/2}$ for group $z$, the shifts are applied to the last five dimensions as follows:

$$\tilde{q}_j^{(z)} = \begin{cases} q_j & \text{if } j = 5 + (z \bmod 5) \text{ or } j = 6 + (z \bmod 5) \\ q_j + w_z \cdot \sin(z \cdot q_j) & \text{if } j \in \{6,7,8,9,10\} \setminus \{5 + (z \bmod 5), 6 + (z \bmod 5)\}. \end{cases}$$

Define $y = g_z(q) + \epsilon$, where $g_z(q) = f_0(q) + f_{0,z}(q) = \sum_{j=1}^{5} \theta_j(q_j^{(z)}) + \sum_{j=6}^{10} \theta_j(\tilde{q}_j^{(z)})$ as the group-specific conditional mean function. The component functions $\{\theta_j : \mathbb{R} \to \mathbb{R}\}_{j=1}^{10}$ are defined as follows:

$$\theta_1(q_1) = \sin(\tanh(q_1^2)), \qquad \theta_6(q_6) = \cos(\sqrt{1 + q_6^2}),$$
$$\theta_2(q_2) = \log(1 + |q_2|), \qquad \theta_7(q_7) = -\log(1 + \tanh(q_7^2)),$$
$$\theta_3(q_3) = \exp(-\sqrt{1 + q_3^2}), \qquad \theta_8(q_8) = -\sin(\log(1 + q_8^2)),$$
$$\theta_4(q_4) = \cos(\log(1 + q_4^2)), \qquad \theta_9(q_9) = \cos(\tanh(q_9^2)),$$
$$\theta_5(q_5) = -\sin(\tanh(|q_5|)), \qquad \theta_{10}(q_{10}) = -\exp(-\sqrt{1 + q_{10}^2}).$$

**Extra Scenario 2.** This scenario combines covariate shifts with mild concept shifts. The shared function remains identical to that in Extra Scenario 1, given by $f_0(q) = \sum_{j=1}^{5} \theta_j(q_j)$. The group-specific effects introduce covariate shifts through the group scale $w_z$, defined as

$$\tilde{q}_j^{(z)} = \begin{cases} q_j & \text{if } j = 5 + z \\ q_j + w_z \cdot \sin(z \cdot q_j) & \text{if } j \in \{6, 7, 8, 9, 10\} \setminus \{5 + z\}. \end{cases}$$

The deviation function $f_{0,z}$ further incorporates concept shift by applying a group-specific scaling $f_{0,z}(q) = (1 + w_z) \cdot \sum_{j=6}^{10} \theta_j(\tilde{q}_j^{(z)})$, where $\theta_6, \ldots, \theta_{10}$ are defined as in Extra Scenario 1.

**Extra Scenario 3.** This scenario introduces a two-layer hierarchical structure where inputs are first mapped to intermediate representations before final transformation. With the scaling factor $w_z$, let

$$\tilde{q}_j^{(z)} = \begin{cases} q_j & \text{if } j = 5 + (z \bmod 5) \text{ or } j = 6 + (z \bmod 5) \\ q_j + w_z \cdot \sin(z \cdot q_j) & \text{if } j \in \{6, 7, 8, 9, 10\} \setminus \{5 + (z \bmod 5), 6 + (z \bmod 5)\} \end{cases}$$

The intermediate transformations are:

$$\begin{aligned} h_1(q) &= \sin(q_1) \cdot q_2^2 + \exp(q_3) - q_4 \cdot q_5, \\ h_2(q) &= \cos(q_2) + q_3 \cdot \tanh(q_4) + q_5^3, \\ h_3(q; z) &= \log\left(1 + \tilde{q}_6 \cdot \tilde{q}_7 - \sin(\tilde{q}_8)\right) - \tanh(\tilde{q}_9 \cdot \tilde{q}_{10}), \\ h_4(q; z) &= \exp(-|\tilde{q}_8|) + \tilde{q}_9 \cdot (1 + \tilde{q}_{10}^2)^{-1}. \end{aligned}$$

The shared and group-specific functions are constructed as:

$$f_0(q) = \sqrt{1 + h_1(q)^2} + \sqrt{1 + h_2(q)^2} \quad \text{and} \quad f_{0,z}(q) = |h_3(q; z) - h_4(q; z)|,$$

and the final response is generated according to

$$y = f_0(q) + f_{0,z}(q) + \epsilon.$$

**Extra Scenario 4.** This scenario extends Extra Scenario 3 beyond the additive model by introducing more complex group effects through mixed dimensional shifts. Specifically, for even indices $j \in \{2, 4, \ldots, 10\}$, we define

$$\tilde{q}_j^{(z)} = q_j + w_z \cdot \sin(z\, q_j).$$

Group effects are then incorporated by modifying the even-indexed dimensions in the following components:

$$\begin{aligned} h_1(q; z) &= \sin(q_1) \cdot \tilde{q}_2^2 + \exp(q_3) - \tilde{q}_4 \cdot q_5, \\ h_2(q; z) &= \cos(\tilde{q}_2) + q_3 \cdot \tanh(\tilde{q}_4) + q_5^3, \\ h_3(q; z) &= \log\left(1 + \tilde{q}_6 \cdot q_7 - \sin(\tilde{q}_8)\right) - \tanh(q_9 \cdot \tilde{q}_{10}), \\ h_4(q; z) &= \exp(-|\tilde{q}_8|) + q_9 \cdot (1 + \tilde{q}_{10}^2)^{-1}. \end{aligned}$$

The final conditional mean for group $z$ is defined as

$$f_z(q) = \sqrt{|h_1(q; z) \cdot h_3(q; z)| + (h_2(q; z) + h_4(q; z))^2},$$

and the response is generated according to $y = f_z(q) + \epsilon$.

In the following, we present the complete set of experimental results, including all signal-to-noise ratio (SNR) settings, as well as group-level results for SNR $= 5$ in the low-dimensional settings. The results are organized from low-dimensional settings (Scenarios 1 and 2, and Extra Scenarios 1–4) to high-dimensional settings (Scenarios 3 and 4). In most cases, the proposed two-stage transfer learning method achieves the best performance as the sample size increases.

*Table 6.* Average MSE for Scenario 1 with different SNR levels over 50 independent trials. The best performance in terms of lower MSE within each SNR group is bolded.

| SNR | Model / Sample size | 1000 | 5000 | 10000 | 30000 | 50000 |
|---|---|---|---|---|---|---|
| | Pooled (NN) | 0.0352 (0.0356) | 0.0214 (0.0016) | 0.0197 (0.0011) | 0.0209 (0.0056) | 0.0184 (0.0016) |
| | 2-Stage (NN) | **0.0175 (0.0064)** | **0.0067 (0.0012)** | **0.0045 (0.0007)** | **0.0025 (0.0003)** | **0.0020 (0.0002)** |
| | Separate (NN) | 0.8039 (0.0735) | 0.0673 (0.0130) | 0.0079 (0.0031) | 0.0041 (0.0005) | 0.0034 (0.0005) |
| | Top-FT (NN) | 0.0315 (0.0249) | 0.0188 (0.0022) | 0.0154 (0.0017) | 0.0093 (0.0020) | 0.0074 (0.0015) |
| 10 | Pool-w-L (NN) | 0.0177 (0.0037) | 0.0069 (0.0011) | 0.0055 (0.0016) | 0.0069 (0.0061) | 0.0036 (0.0016) |
| | Pooled (RF) | 0.0957 (0.0109) | 0.0721 (0.0037) | 0.0690 (0.0028) | 0.0675 (0.0019) | 0.0691 (0.0012) |
| | 2-Stage (RF) | 0.0675 (0.0085) | 0.0410 (0.0026) | 0.0338 (0.0016) | 0.0255 (0.0011) | 0.0242 (0.0008) |
| | Separate (RF) | 0.1300 (0.0198) | 0.0785 (0.0046) | 0.0661 (0.0026) | 0.0548 (0.0015) | 0.0518 (0.0011) |
| | Pool-w-L (RF) | 0.0957 (0.0109) | 0.0707 (0.0035) | 0.0674 (0.0028) | 0.0658 (0.0019) | 0.0675 (0.0012) |
| | Pooled (NN) | 0.0364 (0.0355) | 0.0228 (0.0017) | 0.0213 (0.0022) | 0.0234 (0.0068) | 0.0193 (0.0029) |
| | 2-Stage (NN) | 0.0222 (0.0078) | **0.0093 (0.0016)** | **0.0064 (0.0009)** | **0.0034 (0.0005)** | **0.0027 (0.0004)** |
| | Separate (NN) | 0.8183 (0.0788) | 0.0664 (0.0116) | 0.0097 (0.0035) | 0.0060 (0.0007) | 0.0049 (0.0008) |
| | Top-FT (NN) | 0.0331 (0.0249) | 0.0212 (0.0018) | 0.0177 (0.0018) | 0.0114 (0.0019) | 0.0092 (0.0017) |
| 5 | Pool-w-L (NN) | **0.0219 (0.0044)** | 0.0103 (0.0021) | 0.0074 (0.0017) | 0.0103 (0.0097) | 0.0048 (0.0023) |
| | Pooled (RF) | 0.0952 (0.0105) | 0.0708 (0.0039) | 0.0673 (0.0028) | 0.0655 (0.0018) | 0.0669 (0.0012) |
| | 2-Stage (RF) | 0.0680 (0.0085) | 0.0410 (0.0028) | 0.0337 (0.0016) | 0.0252 (0.0011) | 0.0236 (0.0008) |
| | Separate (RF) | 0.1318 (0.0195) | 0.0775 (0.0047) | 0.0650 (0.0028) | 0.0532 (0.0015) | 0.0500 (0.0011) |
| | Pool-w-L (RF) | 0.0945 (0.0106) | 0.0692 (0.0038) | 0.0656 (0.0028) | 0.0637 (0.0018) | 0.0652 (0.0012) |
| | Pooled (NN) | 0.0395 (0.0350) | 0.0248 (0.0018) | 0.0243 (0.0027) | 0.0271 (0.0094) | 0.0212 (0.0046) |
| | 2-Stage (NN) | 0.0346 (0.0100) | **0.0151 (0.0032)** | **0.0110 (0.0019)** | **0.0059 (0.0010)** | **0.0045 (0.0008)** |
| | Separate (NN) | 0.8097 (0.0875) | 0.0685 (0.0112) | 0.0115 (0.0011) | 0.0087 (0.0008) | 0.0078 (0.0015) |
| | Top-FT (NN) | 0.0370 (0.0244) | 0.0239 (0.0017) | 0.0212 (0.0014) | 0.0155 (0.0018) | 0.0129 (0.0019) |
| 2 | Pool-w-L (NN) | **0.0306 (0.0081)** | 0.0151 (0.0024) | 0.0120 (0.0026) | 0.0146 (0.0105) | 0.0071 (0.0030) |
| | Pooled (RF) | 0.0970 (0.0106) | 0.0703 (0.0041) | 0.0658 (0.0032) | 0.0625 (0.0019) | 0.0631 (0.0012) |
| | 2-Stage (RF) | 0.0738 (0.0089) | 0.0446 (0.0028) | 0.0366 (0.0021) | 0.0267 (0.0013) | 0.0243 (0.0010) |
| | Separate (RF) | 0.1401 (0.0216) | 0.0793 (0.0050) | 0.0655 (0.0033) | 0.0520 (0.0015) | 0.0480 (0.0011) |
| | Pool-w-L (RF) | 0.0960 (0.0105) | 0.0683 (0.0038) | 0.0636 (0.0031) | 0.0603 (0.0018) | 0.0611 (0.0012) |

*Table 7.* Average per-group MSE for Scenario 1 over 50 independent trials at an SNR of 5 using neural networks. The best performance in terms of lower MSE is bolded.

| $n$ | Group | Pooled | 2-Stage | Separate | Top-FT | Pool-w-L |
|---|---|---|---|---|---|---|
| | 1 | 0.0367 (0.0296) | 0.0435 (0.0341) | 0.8045 (0.3100) | 0.0426 (0.0378) | **0.0353 (0.0154)** |
| | 2 | 0.0441 (0.0444) | 0.0302 (0.0302) | 0.8241 (0.2425) | 0.0417 (0.0380) | **0.0283 (0.0100)** |
| 1000 | 3 | 0.0359 (0.0325) | 0.0222 (0.0117) | 0.8402 (0.2157) | 0.0342 (0.0303) | **0.0206 (0.0077)** |
| | 4 | 0.0373 (0.0426) | **0.0205 (0.0151)** | 0.8012 (0.1283) | 0.0328 (0.0272) | 0.0208 (0.0075) |
| | 5 | 0.0326 (0.0351) | **0.0157 (0.0051)** | 0.8198 (0.1723) | 0.0272 (0.0198) | 0.0180 (0.0059) |
| | 1 | 0.0302 (0.0058) | **0.0164 (0.0078)** | 0.8092 (0.1464) | 0.0294 (0.0064) | 0.0171 (0.0059) |
| | 2 | 0.0266 (0.0039) | **0.0107 (0.0031)** | 0.0209 (0.0421) | 0.0259 (0.0043) | 0.0118 (0.0033) |
| 5000 | 3 | 0.0248 (0.0036) | **0.0088 (0.0029)** | 0.0151 (0.0212) | 0.0232 (0.0043) | 0.0105 (0.0029) |
| | 4 | 0.0219 (0.0027) | **0.0087 (0.0024)** | 0.0104 (0.0016) | 0.0205 (0.0030) | 0.0094 (0.0023) |
| | 5 | 0.0191 (0.0026) | **0.0082 (0.0020)** | 0.0094 (0.0016) | 0.0170 (0.0027) | 0.0090 (0.0027) |
| | 1 | 0.0285 (0.0039) | **0.0090 (0.0028)** | 0.0224 (0.0510) | 0.0261 (0.0041) | 0.0103 (0.0032) |
| | 2 | 0.0254 (0.0036) | **0.0069 (0.0012)** | 0.0104 (0.0018) | 0.0214 (0.0038) | 0.0083 (0.0022) |
| 10000 | 3 | 0.0231 (0.0028) | **0.0067 (0.0019)** | 0.0089 (0.0015) | 0.0184 (0.0036) | 0.0076 (0.0023) |
| | 4 | 0.0206 (0.0031) | **0.0061 (0.0016)** | 0.0083 (0.0012) | 0.0167 (0.0030) | 0.0069 (0.0019) |
| | 5 | 0.0176 (0.0026) | **0.0057 (0.0010)** | 0.0085 (0.0014) | 0.0150 (0.0027) | 0.0066 (0.0023) |
| | 1 | 0.0308 (0.0070) | **0.0045 (0.0013)** | 0.0091 (0.0014) | 0.0206 (0.0053) | 0.0122 (0.0094) |
| | 2 | 0.0280 (0.0069) | **0.0040 (0.0012)** | 0.0075 (0.0012) | 0.0154 (0.0037) | 0.0109 (0.0107) |
| 30000 | 3 | 0.0251 (0.0071) | **0.0036 (0.0008)** | 0.0062 (0.0016) | 0.0124 (0.0033) | 0.0107 (0.0104) |
| | 4 | 0.0224 (0.0073) | **0.0031 (0.0007)** | 0.0053 (0.0013) | 0.0105 (0.0031) | 0.0099 (0.0110) |
| | 5 | 0.0197 (0.0068) | **0.0032 (0.0007)** | 0.0053 (0.0014) | 0.0079 (0.0023) | 0.0097 (0.0088) |
| | 1 | 0.0263 (0.0039) | **0.0040 (0.0012)** | 0.0089 (0.0015) | 0.0187 (0.0044) | 0.0060 (0.0026) |
| | 2 | 0.0235 (0.0036) | **0.0030 (0.0006)** | 0.0061 (0.0014) | 0.0124 (0.0037) | 0.0055 (0.0032) |
| 50000 | 3 | 0.0209 (0.0033) | **0.0030 (0.0008)** | 0.0053 (0.0020) | 0.0096 (0.0029) | 0.0048 (0.0029) |
| | 4 | 0.0189 (0.0029) | **0.0026 (0.0006)** | 0.0042 (0.0017) | 0.0084 (0.0025) | 0.0045 (0.0023) |
| | 5 | 0.0156 (0.0032) | **0.0024 (0.0005)** | 0.0039 (0.0013) | 0.0063 (0.0020) | 0.0044 (0.0025) |

*Table 8.* Average MSE for Scenario 2 with different SNR levels over 50 independent trials. The best performance in terms of lower MSE within each SNR group is bolded.

| SNR | Model / Sample size | 1000 | 5000 | 10000 | 30000 | 50000 |
|---|---|---|---|---|---|---|
| | Pooled (NN) | 0.0592 (0.0098) | 0.0358 (0.0049) | 0.0331 (0.0039) | 0.0355 (0.0132) | 0.0263 (0.0026) |
| | 2-Stage (NN) | **0.0537 (0.0097)** | **0.0220 (0.0020)** | **0.0169 (0.0011)** | **0.0109 (0.0010)** | **0.0083 (0.0005)** |
| | Separate (NN) | 0.6172 (0.0661) | 0.0759 (0.0129) | 0.0273 (0.0053) | 0.0160 (0.0014) | 0.0140 (0.0015) |
| | Top-FT (NN) | 0.0579 (0.0103) | 0.0303 (0.0027) | 0.0252 (0.0027) | 0.0166 (0.0018) | 0.0135 (0.0011) |
| 10 | Pool-w-L (NN) | 0.0817 (0.0191) | 0.0271 (0.0026) | 0.0218 (0.0039) | 0.0190 (0.0068) | 0.0112 (0.0020) |
| | Pooled (RF) | 0.1841 (0.0239) | 0.1249 (0.0103) | 0.1161 (0.0066) | 0.1112 (0.0031) | 0.1134 (0.0033) |
| | 2-Stage (RF) | 0.1339 (0.0198) | 0.0770 (0.0077) | 0.0642 (0.0047) | 0.0504 (0.0020) | 0.0475 (0.0019) |
| | Separate (RF) | 0.2683 (0.0426) | 0.1566 (0.0124) | 0.1315 (0.0067) | 0.1045 (0.0026) | 0.0978 (0.0024) |
| | Pool-w-L (RF) | 0.1851 (0.0240) | 0.1241 (0.0105) | 0.1152 (0.0066) | 0.1106 (0.0031) | 0.1129 (0.0033) |
| | Pooled (NN) | 0.0779 (0.0146) | 0.0397 (0.0051) | 0.0351 (0.0033) | 0.0367 (0.0105) | 0.0277 (0.0021) |
| | 2-Stage (NN) | **0.0774 (0.0162)** | **0.0275 (0.0025)** | **0.0209 (0.0015)** | **0.0144 (0.0011)** | **0.0109 (0.0008)** |
| | Separate (NN) | 0.6209 (0.0756) | 0.0884 (0.0138) | 0.0375 (0.0076) | 0.0200 (0.0011) | 0.0175 (0.0019) |
| | Top-FT (NN) | 0.0782 (0.0155) | 0.0352 (0.0026) | 0.0292 (0.0026) | 0.0198 (0.0021) | 0.0164 (0.0015) |
| 5 | Pool-w-L (NN) | 0.1094 (0.0195) | 0.0352 (0.0071) | 0.0273 (0.0054) | 0.0253 (0.0121) | 0.0151 (0.0021) |
| | Pooled (RF) | 0.1828 (0.0245) | 0.1237 (0.0100) | 0.1143 (0.0062) | 0.1087 (0.0032) | 0.1103 (0.0032) |
| | 2-Stage (RF) | 0.1366 (0.0205) | 0.0796 (0.0078) | 0.0658 (0.0045) | 0.0512 (0.0021) | 0.0476 (0.0019) |
| | Separate (RF) | 0.2722 (0.0436) | 0.1570 (0.0129) | 0.1312 (0.0067) | 0.1031 (0.0027) | 0.0957 (0.0024) |
| | Pool-w-L (RF) | 0.1835 (0.0245) | 0.1226 (0.0100) | 0.1133 (0.0061) | 0.1078 (0.0032) | 0.1097 (0.0032) |
| | Pooled (NN) | **0.1080 (0.0168)** | 0.0513 (0.0075) | 0.0420 (0.0056) | 0.0460 (0.0172) | 0.0342 (0.0076) |
| | 2-Stage (NN) | 0.1187 (0.0214) | **0.0433 (0.0059)** | **0.0298 (0.0030)** | **0.0202 (0.0018)** | **0.0165 (0.0014)** |
| | Separate (NN) | 0.6185 (0.0803) | 0.1192 (0.0148) | 0.0615 (0.0108) | 0.0307 (0.0038) | 0.0249 (0.0028) |
| | Top-FT (NN) | 0.1101 (0.0169) | 0.0463 (0.0045) | 0.0369 (0.0025) | 0.0254 (0.0021) | 0.0217 (0.0017) |
| 2 | Pool-w-L (NN) | 0.1423 (0.0272) | 0.0565 (0.0100) | 0.0395 (0.0082) | 0.0291 (0.0111) | 0.0209 (0.0040) |
| | Pooled (RF) | 0.1878 (0.0260) | 0.1244 (0.0107) | 0.1131 (0.0061) | 0.1046 (0.0030) | 0.1047 (0.0029) |
| | 2-Stage (RF) | 0.1508 (0.0225) | 0.0892 (0.0085) | 0.0729 (0.0046) | 0.0548 (0.0021) | 0.0496 (0.0019) |
| | Separate (RF) | 0.2912 (0.0448) | 0.1625 (0.0138) | 0.1346 (0.0071) | 0.1033 (0.0031) | 0.0940 (0.0022) |
| | Pool-w-L (RF) | 0.1883 (0.0261) | 0.1228 (0.0104) | 0.1116 (0.0061) | 0.1034 (0.0030) | 0.1037 (0.0029) |

*Table 9.* Average per-group MSE for Scenario 2 over 50 independent trials at an SNR of 5 using neural networks. The best performance in terms of lower MSE is bolded.

| $n$ | Group | Pooled | 2-Stage | Separate | Top-FT | Pool-w-L |
|---|---|---|---|---|---|---|
| | 1 | **0.0768 (0.0401)** | 0.0852 (0.0420) | 0.6384 (0.2559) | 0.0820 (0.0420) | 0.1501 (0.0883) |
| | 2 | **0.0804 (0.0265)** | 0.0843 (0.0322) | 0.6267 (0.1901) | 0.0810 (0.0262) | 0.1269 (0.0490) |
| 1000 | 3 | **0.0861 (0.0384)** | 0.0862 (0.0416) | 0.6510 (0.1942) | 0.0871 (0.0407) | 0.1133 (0.0402) |
| | 4 | 0.0811 (0.0211) | **0.0770 (0.0212)** | 0.6245 (0.1700) | 0.0807 (0.0224) | 0.1038 (0.0327) |
| | 5 | 0.0698 (0.0201) | **0.0680 (0.0194)** | 0.5940 (0.1198) | 0.0691 (0.0197) | 0.0957 (0.0258) |
| | 1 | 0.0467 (0.0128) | **0.0398 (0.0123)** | 0.6086 (0.1238) | 0.0435 (0.0103) | 0.0507 (0.0131) |
| | 2 | 0.0450 (0.0087) | **0.0302 (0.0061)** | 0.0865 (0.0717) | 0.0408 (0.0069) | 0.0418 (0.0102) |
| 5000 | 3 | 0.0421 (0.0083) | **0.0281 (0.0054)** | 0.0523 (0.0174) | 0.0379 (0.0075) | 0.0368 (0.0089) |
| | 4 | 0.0393 (0.0070) | **0.0261 (0.0051)** | 0.0461 (0.0100) | 0.0344 (0.0058) | 0.0336 (0.0103) |
| | 5 | 0.0350 (0.0054) | **0.0246 (0.0041)** | 0.0389 (0.0077) | 0.0304 (0.0036) | 0.0298 (0.0072) |
| | 1 | 0.0424 (0.0089) | **0.0275 (0.0068)** | 0.1050 (0.1076) | 0.0373 (0.0079) | 0.0382 (0.0101) |
| | 2 | 0.0406 (0.0064) | **0.0235 (0.0049)** | 0.0456 (0.0092) | 0.0349 (0.0064) | 0.0311 (0.0068) |
| 10000 | 3 | 0.0374 (0.0061) | **0.0212 (0.0031)** | 0.0346 (0.0054) | 0.0308 (0.0060) | 0.0273 (0.0057) |
| | 4 | 0.0340 (0.0038) | **0.0199 (0.0031)** | 0.0298 (0.0059) | 0.0277 (0.0044) | 0.0251 (0.0066) |
| | 5 | 0.0309 (0.0048) | **0.0192 (0.0027)** | 0.0285 (0.0062) | 0.0255 (0.0038) | 0.0253 (0.0063) |
| | 1 | 0.0441 (0.0133) | **0.0182 (0.0032)** | 0.0353 (0.0068) | 0.0301 (0.0068) | 0.0322 (0.0170) |
| | 2 | 0.0416 (0.0119) | **0.0163 (0.0027)** | 0.0231 (0.0035) | 0.0219 (0.0040) | 0.0280 (0.0132) |
| 30000 | 3 | 0.0387 (0.0107) | **0.0145 (0.0022)** | 0.0198 (0.0031) | 0.0208 (0.0035) | 0.0247 (0.0114) |
| | 4 | 0.0352 (0.0108) | **0.0139 (0.0019)** | 0.0182 (0.0034) | 0.0185 (0.0033) | 0.0236 (0.0141) |
| | 5 | 0.0331 (0.0105) | **0.0131 (0.0016)** | 0.0172 (0.0024) | 0.0174 (0.0023) | 0.0245 (0.0128) |
| | 1 | 0.0359 (0.0047) | **0.0147 (0.0028)** | 0.0284 (0.0057) | 0.0256 (0.0048) | 0.0218 (0.0043) |
| | 2 | 0.0328 (0.0044) | **0.0126 (0.0016)** | 0.0206 (0.0045) | 0.0193 (0.0037) | 0.0171 (0.0036) |
| 50000 | 3 | 0.0293 (0.0030) | **0.0110 (0.0012)** | 0.0182 (0.0049) | 0.0170 (0.0024) | 0.0152 (0.0030) |
| | 4 | 0.0265 (0.0035) | **0.0104 (0.0012)** | 0.0156 (0.0027) | 0.0153 (0.0020) | 0.0142 (0.0032) |
| | 5 | 0.0242 (0.0033) | **0.0097 (0.0011)** | 0.0152 (0.0034) | 0.0139 (0.0019) | 0.0136 (0.0029) |

*Table 10.* Average MSE for Extra Scenario 1 with different SNR levels over 50 independent trials. The best performance in terms of lower MSE within each SNR group is bolded.

| SNR | Model / Sample size | 1000 | 5000 | 10000 | 30000 | 50000 |
|---|---|---|---|---|---|---|
| | Pooled (NN) | 0.0146 (0.0021) | 0.0106 (0.0009) | 0.0096 (0.0009) | 0.0100 (0.0024) | 0.0085 (0.0007) |
| | 2-Stage (NN) | **0.0109 (0.0016)** | **0.0051 (0.0007)** | **0.0035 (0.0003)** | **0.0021 (0.0002)** | **0.0016 (0.0001)** |
| | Separate (NN) | 0.4149 (0.0643) | 0.0367 (0.0080) | 0.0059 (0.0005) | 0.0032 (0.0003) | 0.0026 (0.0003) |
| | Top-FT (NN) | 0.0114 (0.0020) | 0.0060 (0.0006) | 0.0044 (0.0003) | 0.0030 (0.0002) | 0.0026 (0.0002) |
| 10 | Pool-w-L (NN) | 0.0115 (0.0019) | 0.0059 (0.0008) | 0.0042 (0.0008) | 0.0043 (0.0026) | 0.0019 (0.0006) |
| | Pooled (RF) | 0.0405 (0.0056) | 0.0278 (0.0017) | 0.0255 (0.0013) | 0.0247 (0.0006) | 0.0254 (0.0005) |
| | 2-Stage (RF) | 0.0279 (0.0039) | 0.0157 (0.0010) | 0.0124 (0.0007) | 0.0093 (0.0004) | 0.0086 (0.0003) |
| | Separate (RF) | 0.0640 (0.0092) | 0.0326 (0.0019) | 0.0261 (0.0012) | 0.0201 (0.0006) | 0.0188 (0.0004) |
| | Pool-w-L (RF) | 0.0402 (0.0055) | 0.0271 (0.0017) | 0.0247 (0.0012) | 0.0240 (0.0006) | 0.0247 (0.0005) |
| | Pooled (NN) | 0.0170 (0.0027) | 0.0121 (0.0011) | 0.0106 (0.0011) | 0.0119 (0.0039) | 0.0089 (0.0011) |
| | 2-Stage (NN) | **0.0140 (0.0028)** | **0.0068 (0.0008)** | **0.0047 (0.0005)** | **0.0028 (0.0003)** | **0.0022 (0.0002)** |
| | Separate (NN) | 0.4180 (0.0605) | 0.0365 (0.0099) | 0.0079 (0.0020) | 0.0045 (0.0006) | 0.0037 (0.0005) |
| | Top-FT (NN) | **0.0140 (0.0026)** | 0.0074 (0.0007) | 0.0054 (0.0005) | 0.0035 (0.0003) | 0.0030 (0.0002) |
| 5 | Pool-w-L (NN) | 0.0148 (0.0028) | 0.0078 (0.0011) | 0.0059 (0.0012) | 0.0050 (0.0023) | 0.0026 (0.0007) |
| | Pooled (RF) | 0.0412 (0.0060) | 0.0277 (0.0017) | 0.0252 (0.0013) | 0.0241 (0.0006) | 0.0246 (0.0005) |
| | 2-Stage (RF) | 0.0294 (0.0043) | 0.0164 (0.0011) | 0.0130 (0.0008) | 0.0097 (0.0004) | 0.0088 (0.0004) |
| | Separate (RF) | 0.0653 (0.0096) | 0.0330 (0.0021) | 0.0263 (0.0013) | 0.0200 (0.0006) | 0.0185 (0.0004) |
| | Pool-w-L (RF) | 0.0408 (0.0059) | 0.0269 (0.0016) | 0.0243 (0.0013) | 0.0233 (0.0006) | 0.0239 (0.0005) |
| | Pooled (NN) | 0.0216 (0.0044) | 0.0143 (0.0017) | 0.0128 (0.0013) | 0.0122 (0.0030) | 0.0098 (0.0010) |
| | 2-Stage (NN) | 0.0220 (0.0060) | 0.0101 (0.0017) | **0.0074 (0.0011)** | **0.0045 (0.0006)** | **0.0037 (0.0004)** |
| | Separate (NN) | 0.4198 (0.0632) | 0.0411 (0.0119) | 0.0113 (0.0019) | 0.0070 (0.0010) | 0.0057 (0.0007) |
| | Top-FT (NN) | **0.0197 (0.0042)** | **0.0097 (0.0009)** | **0.0074 (0.0007)** | 0.0047 (0.0004) | 0.0040 (0.0003) |
| 2 | Pool-w-L (NN) | 0.0207 (0.0036) | 0.0108 (0.0017) | 0.0084 (0.0015) | 0.0069 (0.0028) | 0.0040 (0.0009) |
| | Pooled (RF) | 0.0444 (0.0071) | 0.0290 (0.0018) | 0.0259 (0.0013) | 0.0235 (0.0007) | 0.0236 (0.0005) |
| | 2-Stage (RF) | 0.0350 (0.0055) | 0.0196 (0.0014) | 0.0156 (0.0009) | 0.0112 (0.0005) | 0.0099 (0.0004) |
| | Separate (RF) | 0.0696 (0.0104) | 0.0353 (0.0020) | 0.0280 (0.0014) | 0.0210 (0.0007) | 0.0189 (0.0005) |
| | Pool-w-L (RF) | 0.0440 (0.0068) | 0.0279 (0.0017) | 0.0248 (0.0013) | 0.0226 (0.0007) | 0.0226 (0.0005) |

*Table 11.* Average per-group MSE for Extra Scenario 1 over 50 independent trials at an SNR of 5 using neural networks. The best performance in terms of lower MSE is bolded.

| $n$ | Group | Pooled | 2-Stage | Separate | Top-FT | Pool-w-L |
|---|---|---|---|---|---|---|
| | 1 | 0.0254 (0.0121) | **0.0224 (0.0112)** | 0.4225 (0.2332) | 0.0225 (0.0106) | 0.0253 (0.0109) |
| | 2 | 0.0271 (0.0116) | **0.0199 (0.0085)** | 0.5008 (0.1793) | 0.0227 (0.0096) | 0.0208 (0.0088) |
| 1000 | 3 | 0.0152 (0.0051) | 0.0136 (0.0050) | 0.4046 (0.1851) | **0.0128 (0.0044)** | 0.0151 (0.0056) |
| | 4 | 0.0110 (0.0035) | 0.0118 (0.0036) | 0.3859 (0.0916) | **0.0108 (0.0036)** | 0.0123 (0.0037) |
| | 5 | 0.0170 (0.0035) | **0.0117 (0.0030)** | 0.4167 (0.0992) | 0.0119 (0.0025) | 0.0120 (0.0035) |
| | 1 | 0.0213 (0.0064) | **0.0112 (0.0034)** | 0.4110 (0.1464) | 0.0157 (0.0042) | 0.0131 (0.0032) |
| | 2 | 0.0180 (0.0046) | **0.0085 (0.0024)** | 0.0135 (0.0027) | 0.0114 (0.0026) | 0.0100 (0.0022) |
| 5000 | 3 | 0.0101 (0.0026) | 0.0075 (0.0017) | 0.0107 (0.0018) | **0.0073 (0.0011)** | 0.0088 (0.0019) |
| | 4 | 0.0069 (0.0010) | 0.0064 (0.0012) | 0.0085 (0.0011) | **0.0060 (0.0008)** | 0.0070 (0.0014) |
| | 5 | 0.0131 (0.0049) | **0.0053 (0.0014)** | 0.0074 (0.0010) | 0.0053 (0.0010) | 0.0060 (0.0011) |
| | 1 | 0.0207 (0.0053) | **0.0067 (0.0019)** | 0.0178 (0.0281) | 0.0121 (0.0029) | 0.0094 (0.0028) |
| | 2 | 0.0166 (0.0040) | **0.0056 (0.0011)** | 0.0093 (0.0014) | 0.0088 (0.0020) | 0.0074 (0.0019) |
| 10000 | 3 | 0.0088 (0.0027) | **0.0052 (0.0011)** | 0.0083 (0.0013) | 0.0053 (0.0007) | 0.0065 (0.0015) |
| | 4 | 0.0057 (0.0009) | 0.0045 (0.0007) | 0.0066 (0.0011) | **0.0044 (0.0006)** | 0.0052 (0.0012) |
| | 5 | 0.0112 (0.0049) | 0.0038 (0.0009) | 0.0060 (0.0012) | **0.0036 (0.0005)** | 0.0047 (0.0016) |
| | 1 | 0.0221 (0.0093) | **0.0039 (0.0007)** | 0.0085 (0.0014) | 0.0076 (0.0017) | 0.0073 (0.0032) |
| | 2 | 0.0181 (0.0079) | **0.0038 (0.0006)** | 0.0058 (0.0009) | 0.0055 (0.0009) | 0.0061 (0.0028) |
| 30000 | 3 | 0.0101 (0.0059) | **0.0034 (0.0007)** | 0.0050 (0.0010) | 0.0038 (0.0004) | 0.0052 (0.0024) |
| | 4 | 0.0071 (0.0036) | **0.0028 (0.0005)** | 0.0038 (0.0010) | 0.0030 (0.0003) | 0.0047 (0.0025) |
| | 5 | 0.0124 (0.0115) | **0.0020 (0.0005)** | 0.0034 (0.0013) | 0.0021 (0.0003) | 0.0043 (0.0029) |
| | 1 | 0.0184 (0.0050) | **0.0035 (0.0006)** | 0.0078 (0.0012) | 0.0064 (0.0012) | 0.0040 (0.0010) |
| | 2 | 0.0146 (0.0046) | **0.0030 (0.0005)** | 0.0048 (0.0010) | 0.0050 (0.0009) | 0.0033 (0.0011) |
| 50000 | 3 | 0.0067 (0.0031) | **0.0025 (0.0004)** | 0.0039 (0.0009) | 0.0033 (0.0003) | 0.0029 (0.0010) |
| | 4 | 0.0042 (0.0014) | **0.0021 (0.0003)** | 0.0033 (0.0011) | 0.0027 (0.0003) | 0.0023 (0.0008) |
| | 5 | 0.0098 (0.0040) | **0.0016 (0.0003)** | 0.0025 (0.0008) | 0.0017 (0.0002) | 0.0021 (0.0008) |

*Table 12.* Average MSE for Extra Scenario 2 with different SNR levels over 50 independent trials. The best performance in terms of lower MSE within each SNR group is bolded.

| SNR | Model / Sample size | 1000 | 5000 | 10000 | 30000 | 50000 |
|---|---|---|---|---|---|---|
| | Pooled (NN) | 0.0223 (0.0040) | 0.0163 (0.0014) | 0.0153 (0.0016) | 0.0147 (0.0021) | 0.0133 (0.0011) |
| | 2-Stage (NN) | 0.0163 (0.0036) | **0.0065 (0.0008)** | **0.0045 (0.0004)** | **0.0028 (0.0003)** | **0.0020 (0.0002)** |
| | Separate (NN) | 0.4959 (0.0843) | 0.0482 (0.0115) | 0.0079 (0.0007) | 0.0043 (0.0004) | 0.0037 (0.0004) |
| | Top-FT (NN) | 0.0181 (0.0038) | 0.0082 (0.0011) | 0.0065 (0.0008) | 0.0039 (0.0004) | 0.0031 (0.0005) |
| 10 | Pool-w-L (NN) | **0.0157 (0.0026)** | 0.0076 (0.0010) | 0.0054 (0.0009) | 0.0059 (0.0046) | 0.0024 (0.0006) |
| | Pooled (RF) | 0.0518 (0.0078) | 0.0355 (0.0023) | 0.0331 (0.0015) | 0.0318 (0.0010) | 0.0325 (0.0007) |
| | 2-Stage (RF) | 0.0358 (0.0055) | 0.0199 (0.0014) | 0.0159 (0.0008) | 0.0117 (0.0006) | 0.0108 (0.0004) |
| | Separate (RF) | 0.0773 (0.0103) | 0.0398 (0.0023) | 0.0316 (0.0014) | 0.0240 (0.0007) | 0.0222 (0.0005) |
| | Pool-w-L (RF) | 0.0510 (0.0076) | 0.0338 (0.0022) | 0.0308 (0.0014) | 0.0291 (0.0009) | 0.0295 (0.0006) |
| | Pooled (NN) | 0.0248 (0.0049) | 0.0180 (0.0018) | 0.0159 (0.0013) | 0.0157 (0.0027) | 0.0141 (0.0012) |
| | 2-Stage (NN) | **0.0200 (0.0044)** | **0.0088 (0.0010)** | **0.0060 (0.0006)** | **0.0037 (0.0003)** | **0.0029 (0.0003)** |
| | Separate (NN) | 0.4885 (0.0793) | 0.0504 (0.0136) | 0.0099 (0.0007) | 0.0059 (0.0006) | 0.0049 (0.0006) |
| | Top-FT (NN) | 0.0212 (0.0046) | 0.0103 (0.0013) | 0.0077 (0.0008) | 0.0048 (0.0005) | 0.0040 (0.0004) |
| 5 | Pool-w-L (NN) | **0.0200 (0.0038)** | 0.0098 (0.0014) | 0.0078 (0.0019) | 0.0065 (0.0029) | 0.0039 (0.0013) |
| | Pooled (RF) | 0.0524 (0.0076) | 0.0357 (0.0021) | 0.0329 (0.0016) | 0.0312 (0.0009) | 0.0317 (0.0007) |
| | 2-Stage (RF) | 0.0376 (0.0058) | 0.0209 (0.0014) | 0.0167 (0.0009) | 0.0122 (0.0006) | 0.0110 (0.0005) |
| | Separate (RF) | 0.0793 (0.0115) | 0.0404 (0.0024) | 0.0320 (0.0014) | 0.0239 (0.0007) | 0.0219 (0.0005) |
| | Pool-w-L (RF) | 0.0513 (0.0076) | 0.0338 (0.0020) | 0.0306 (0.0015) | 0.0284 (0.0009) | 0.0287 (0.0007) |
| | Pooled (NN) | 0.0306 (0.0067) | 0.0208 (0.0023) | 0.0188 (0.0020) | 0.0191 (0.0054) | 0.0152 (0.0017) |
| | 2-Stage (NN) | 0.0296 (0.0067) | **0.0129 (0.0016)** | **0.0096 (0.0013)** | **0.0055 (0.0006)** | **0.0045 (0.0005)** |
| | Separate (NN) | 0.5007 (0.0691) | 0.0556 (0.0130) | 0.0147 (0.0031) | 0.0091 (0.0008) | 0.0076 (0.0009) |
| | Top-FT (NN) | 0.0285 (0.0065) | 0.0138 (0.0015) | 0.0106 (0.0013) | 0.0065 (0.0005) | 0.0055 (0.0007) |
| 2 | Pool-w-L (NN) | **0.0281 (0.0058)** | 0.0140 (0.0017) | 0.0111 (0.0016) | 0.0095 (0.0042) | 0.0060 (0.0024) |
| | Pooled (RF) | 0.0563 (0.0088) | 0.0374 (0.0022) | 0.0338 (0.0017) | 0.0307 (0.0009) | 0.0306 (0.0008) |
| | 2-Stage (RF) | 0.0439 (0.0075) | 0.0249 (0.0017) | 0.0199 (0.0012) | 0.0141 (0.0006) | 0.0123 (0.0005) |
| | Separate (RF) | 0.0851 (0.0128) | 0.0434 (0.0029) | 0.0344 (0.0017) | 0.0252 (0.0009) | 0.0226 (0.0006) |
| | Pool-w-L (RF) | 0.0554 (0.0088) | 0.0353 (0.0022) | 0.0314 (0.0016) | 0.0279 (0.0009) | 0.0277 (0.0007) |

*Table 13.* Average per-group MSE for Extra Scenario 2 over 50 independent trials at an SNR of 5 using neural networks. The best performance in terms of lower MSE is bolded.

| $n$ | Group | Pooled | 2-Stage | Separate | Top-FT | Pool-w-L |
|---|---|---|---|---|---|---|
| | 1 | 0.0463 (0.0228) | **0.0426 (0.0209)** | 0.5862 (0.3274) | 0.0436 (0.0225) | 0.0459 (0.0173) |
| | 2 | 0.0631 (0.0245) | **0.0372 (0.0186)** | 0.6185 (0.2614) | 0.0505 (0.0221) | 0.0373 (0.0180) |
| 1000 | 3 | 0.0168 (0.0058) | 0.0183 (0.0089) | 0.4944 (0.1956) | **0.0162 (0.0059)** | 0.0175 (0.0059) |
| | 4 | 0.0141 (0.0044) | 0.0144 (0.0044) | 0.4476 (0.1183) | **0.0134 (0.0037)** | 0.0154 (0.0050) |
| | 5 | 0.0176 (0.0040) | 0.0137 (0.0037) | 0.4433 (0.1004) | 0.0135 (0.0039) | **0.0124 (0.0034)** |
| | 1 | 0.0393 (0.0091) | **0.0198 (0.0088)** | 0.5857 (0.2020) | 0.0259 (0.0095) | 0.0205 (0.0049) |
| | 2 | 0.0465 (0.0112) | **0.0127 (0.0034)** | 0.0201 (0.0038) | 0.0216 (0.0057) | 0.0160 (0.0032) |
| 5000 | 3 | 0.0100 (0.0022) | 0.0086 (0.0022) | 0.0131 (0.0023) | **0.0082 (0.0012)** | 0.0099 (0.0023) |
| | 4 | 0.0097 (0.0022) | 0.0077 (0.0018) | 0.0101 (0.0017) | **0.0069 (0.0010)** | 0.0081 (0.0016) |
| | 5 | 0.0136 (0.0037) | **0.0059 (0.0011)** | 0.0084 (0.0013) | 0.0066 (0.0012) | 0.0064 (0.0013) |
| | 1 | 0.0396 (0.0075) | **0.0111 (0.0028)** | 0.0217 (0.0049) | 0.0203 (0.0054) | 0.0146 (0.0050) |
| | 2 | 0.0428 (0.0068) | **0.0085 (0.0021)** | 0.0146 (0.0023) | 0.0159 (0.0029) | 0.0121 (0.0031) |
| 10000 | 3 | 0.0084 (0.0020) | **0.0063 (0.0012)** | 0.0102 (0.0015) | 0.0065 (0.0010) | 0.0082 (0.0021) |
| | 4 | 0.0077 (0.0020) | 0.0053 (0.0009) | 0.0081 (0.0012) | **0.0051 (0.0006)** | 0.0067 (0.0023) |
| | 5 | 0.0114 (0.0032) | **0.0043 (0.0008)** | 0.0069 (0.0011) | 0.0047 (0.0008) | 0.0054 (0.0019) |
| | 1 | 0.0380 (0.0077) | **0.0067 (0.0010)** | 0.0127 (0.0022) | 0.0134 (0.0031) | 0.0105 (0.0041) |
| | 2 | 0.0418 (0.0122) | **0.0055 (0.0008)** | 0.0084 (0.0012) | 0.0094 (0.0019) | 0.0084 (0.0038) |
| 30000 | 3 | 0.0082 (0.0040) | **0.0038 (0.0005)** | 0.0069 (0.0017) | 0.0044 (0.0006) | 0.0065 (0.0031) |
| | 4 | 0.0079 (0.0038) | 0.0035 (0.0006) | 0.0049 (0.0009) | **0.0034 (0.0004)** | 0.0056 (0.0028) |
| | 5 | 0.0115 (0.0052) | **0.0025 (0.0004)** | 0.0037 (0.0007) | 0.0026 (0.0004) | 0.0056 (0.0035) |
| | 1 | 0.0366 (0.0051) | **0.0053 (0.0008)** | 0.0118 (0.0029) | 0.0108 (0.0022) | 0.0067 (0.0024) |
| | 2 | 0.0398 (0.0075) | **0.0041 (0.0008)** | 0.0069 (0.0019) | 0.0083 (0.0020) | 0.0054 (0.0021) |
| 50000 | 3 | 0.0062 (0.0021) | **0.0031 (0.0006)** | 0.0052 (0.0011) | 0.0035 (0.0005) | 0.0043 (0.0014) |
| | 4 | 0.0064 (0.0020) | **0.0028 (0.0006)** | 0.0040 (0.0010) | 0.0029 (0.0003) | 0.0035 (0.0013) |
| | 5 | 0.0101 (0.0029) | **0.0019 (0.0003)** | 0.0034 (0.0014) | 0.0021 (0.0004) | 0.0027 (0.0013) |

*Table 14.* Average MSE for Extra Scenario 3 with different SNR levels over 50 independent trials. The best performance in terms of lower MSE within each SNR group is bolded.

| SNR | Model / Sample size | 1000 | 5000 | 10000 | 30000 | 50000 |
|---|---|---|---|---|---|---|
| | Pooled (NN) | 0.0536 (0.0095) | 0.0375 (0.0031) | 0.0345 (0.0026) | 0.0374 (0.0114) | 0.0298 (0.0025) |
| | 2-Stage (NN) | **0.0393 (0.0090)** | **0.0164 (0.0020)** | **0.0112 (0.0011)** | **0.0068 (0.0007)** | **0.0050 (0.0004)** |
| | Separate (NN) | 1.4760 (0.1806) | 0.1328 (0.0264) | 0.0200 (0.0018) | 0.0112 (0.0012) | 0.0091 (0.0010) |
| | Top-FT (NN) | **0.0393 (0.0082)** | 0.0191 (0.0018) | 0.0150 (0.0013) | 0.0106 (0.0010) | 0.0086 (0.0008) |
| 10 | Pool-w-L (NN) | 0.0480 (0.0100) | 0.0181 (0.0021) | 0.0142 (0.0025) | 0.0143 (0.0062) | 0.0064 (0.0014) |
| | Pooled (RF) | 0.0875 (0.0139) | 0.0624 (0.0044) | 0.0578 (0.0035) | 0.0543 (0.0016) | 0.0535 (0.0013) |
| | 2-Stage (RF) | 0.0631 (0.0107) | 0.0346 (0.0029) | 0.0276 (0.0019) | 0.0195 (0.0007) | 0.0172 (0.0006) |
| | Separate (RF) | 0.1239 (0.0140) | 0.0652 (0.0049) | 0.0503 (0.0026) | 0.0366 (0.0009) | 0.0331 (0.0008) |
| | Pool-w-L (RF) | 0.0843 (0.0138) | 0.0567 (0.0042) | 0.0510 (0.0029) | 0.0460 (0.0013) | 0.0450 (0.0011) |
| | Pooled (NN) | 0.0610 (0.0100) | 0.0407 (0.0032) | 0.0376 (0.0042) | 0.0406 (0.0103) | 0.0312 (0.0038) |
| | 2-Stage (NN) | 0.0500 (0.0117) | **0.0210 (0.0027)** | **0.0144 (0.0015)** | **0.0088 (0.0006)** | **0.0068 (0.0006)** |
| | Separate (NN) | 1.4668 (0.1573) | 0.1385 (0.0261) | 0.0262 (0.0019) | 0.0148 (0.0014) | 0.0119 (0.0012) |
| | Top-FT (NN) | **0.0472 (0.0091)** | 0.0223 (0.0019) | 0.0175 (0.0016) | 0.0121 (0.0011) | 0.0099 (0.0008) |
| 5 | Pool-w-L (NN) | 0.0546 (0.0099) | 0.0240 (0.0037) | 0.0181 (0.0042) | 0.0174 (0.0066) | 0.0098 (0.0036) |
| | Pooled (RF) | 0.0910 (0.0142) | 0.0641 (0.0044) | 0.0588 (0.0037) | 0.0542 (0.0016) | 0.0530 (0.0014) |
| | 2-Stage (RF) | 0.0688 (0.0118) | 0.0376 (0.0030) | 0.0301 (0.0020) | 0.0209 (0.0008) | 0.0182 (0.0007) |
| | Separate (RF) | 0.1296 (0.0150) | 0.0684 (0.0051) | 0.0530 (0.0028) | 0.0380 (0.0010) | 0.0339 (0.0009) |
| | Pool-w-L (RF) | 0.0873 (0.0138) | 0.0584 (0.0043) | 0.0519 (0.0031) | 0.0458 (0.0014) | 0.0444 (0.0012) |
| | Pooled (NN) | 0.0720 (0.0121) | 0.0483 (0.0051) | 0.0422 (0.0035) | 0.0449 (0.0160) | 0.0344 (0.0043) |
| | 2-Stage (NN) | 0.0649 (0.0166) | 0.0302 (0.0037) | **0.0214 (0.0023)** | **0.0131 (0.0012)** | **0.0105 (0.0011)** |
| | Separate (NN) | 1.4359 (0.1835) | 0.1539 (0.0312) | 0.0370 (0.0031) | 0.0222 (0.0025) | 0.0185 (0.0024) |
| | Top-FT (NN) | **0.0596 (0.0119)** | **0.0294 (0.0035)** | 0.0220 (0.0018) | 0.0155 (0.0014) | 0.0128 (0.0010) |
| 2 | Pool-w-L (NN) | 0.0654 (0.0114) | 0.0354 (0.0054) | 0.0255 (0.0043) | 0.0260 (0.0153) | 0.0145 (0.0043) |
| | Pooled (RF) | 0.1018 (0.0160) | 0.0701 (0.0051) | 0.0624 (0.0040) | 0.0556 (0.0019) | 0.0532 (0.0015) |
| | 2-Stage (RF) | 0.0853 (0.0142) | 0.0473 (0.0041) | 0.0372 (0.0023) | 0.0254 (0.0010) | 0.0215 (0.0009) |
| | Separate (RF) | 0.1460 (0.0177) | 0.0796 (0.0057) | 0.0615 (0.0031) | 0.0433 (0.0011) | 0.0377 (0.0011) |
| | Pool-w-L (RF) | 0.0982 (0.0151) | 0.0642 (0.0049) | 0.0555 (0.0035) | 0.0471 (0.0017) | 0.0445 (0.0013) |

*Table 15.* Average per-group MSE for Extra Scenario 3 over 50 independent trials at an SNR of 5 using neural networks. The best performance in terms of lower MSE is bolded.

| $n$ | Group | Pooled | 2-Stage | Separate | Top-FT | Pool-w-L |
|---|---|---|---|---|---|---|
| | 1 | 0.2043 (0.0972) | **0.0975 (0.0573)** | 1.8615 (0.9071) | 0.1011 (0.0584) | 0.1085 (0.0573) |
| | 2 | 0.0747 (0.0239) | 0.0681 (0.0414) | 1.6906 (0.5349) | **0.0609 (0.0212)** | 0.0675 (0.0227) |
| 1000 | 3 | 0.0646 (0.0187) | 0.0468 (0.0169) | 1.3591 (0.4425) | **0.0452 (0.0140)** | 0.0503 (0.0186) |
| | 4 | 0.0401 (0.0114) | 0.0395 (0.0105) | 1.3825 (0.3397) | **0.0373 (0.0107)** | 0.0473 (0.0115) |
| | 5 | 0.0392 (0.0078) | 0.0426 (0.0124) | 1.4208 (0.3058) | **0.0390 (0.0084)** | 0.0459 (0.0113) |
| | 1 | 0.2000 (0.0467) | 0.0556 (0.0215) | 1.6041 (0.3878) | 0.0699 (0.0161) | **0.0528 (0.0175)** |
| | 2 | 0.0496 (0.0129) | **0.0249 (0.0051)** | 0.0507 (0.0119) | 0.0272 (0.0050) | 0.0295 (0.0061) |
| 5000 | 3 | 0.0421 (0.0116) | 0.0199 (0.0054) | 0.0368 (0.0076) | **0.0191 (0.0037)** | 0.0235 (0.0046) |
| | 4 | 0.0203 (0.0049) | 0.0168 (0.0032) | 0.0272 (0.0043) | **0.0157 (0.0020)** | 0.0201 (0.0045) |
| | 5 | 0.0203 (0.0029) | **0.0165 (0.0033)** | 0.0260 (0.0047) | 0.0179 (0.0027) | 0.0193 (0.0038) |
| | 1 | 0.1944 (0.0449) | **0.0283 (0.0075)** | 0.0599 (0.0170) | 0.0569 (0.0129) | 0.0375 (0.0106) |
| | 2 | 0.0456 (0.0153) | **0.0181 (0.0039)** | 0.0378 (0.0082) | 0.0210 (0.0033) | 0.0237 (0.0075) |
| 10000 | 3 | 0.0386 (0.0161) | **0.0140 (0.0027)** | 0.0247 (0.0043) | 0.0143 (0.0023) | 0.0170 (0.0046) |
| | 4 | 0.0178 (0.0073) | **0.0119 (0.0016)** | 0.0204 (0.0041) | 0.0120 (0.0013) | 0.0152 (0.0037) |
| | 5 | 0.0179 (0.0041) | **0.0124 (0.0020)** | 0.0204 (0.0030) | 0.0145 (0.0018) | 0.0150 (0.0038) |
| | 1 | 0.2025 (0.0765) | **0.0161 (0.0030)** | 0.0342 (0.0048) | 0.0455 (0.0090) | 0.0304 (0.0089) |
| | 2 | 0.0503 (0.0297) | **0.0108 (0.0017)** | 0.0205 (0.0036) | 0.0152 (0.0021) | 0.0207 (0.0086) |
| 30000 | 3 | 0.0395 (0.0333) | **0.0088 (0.0015)** | 0.0140 (0.0026) | 0.0094 (0.0014) | 0.0162 (0.0075) |
| | 4 | 0.0210 (0.0169) | **0.0078 (0.0012)** | 0.0119 (0.0026) | 0.0083 (0.0010) | 0.0154 (0.0072) |
| | 5 | 0.0207 (0.0101) | **0.0073 (0.0008)** | 0.0116 (0.0025) | 0.0088 (0.0011) | 0.0157 (0.0067) |
| | 1 | 0.1874 (0.0393) | **0.0128 (0.0020)** | 0.0284 (0.0049) | 0.0394 (0.0076) | 0.0189 (0.0049) |
| | 2 | 0.0393 (0.0140) | **0.0082 (0.0014)** | 0.0156 (0.0025) | 0.0123 (0.0016) | 0.0123 (0.0052) |
| 50000 | 3 | 0.0316 (0.0165) | **0.0068 (0.0011)** | 0.0123 (0.0030) | 0.0076 (0.0010) | 0.0094 (0.0042) |
| | 4 | 0.0123 (0.0079) | **0.0063 (0.0010)** | 0.0096 (0.0020) | 0.0069 (0.0008) | 0.0090 (0.0040) |
| | 5 | 0.0116 (0.0039) | **0.0053 (0.0007)** | 0.0088 (0.0023) | 0.0069 (0.0007) | 0.0079 (0.0035) |

*Table 16.* Average MSE for Extra Scenario 4 with different SNR levels over 50 independent trials. The best performance in terms of lower MSE within each SNR group is bolded.

| SNR | Model / Sample size | 1000 | 5000 | 10000 | 30000 | 50000 |
|---|---|---|---|---|---|---|
| | Pooled (NN) | 0.0159 (0.0026) | 0.0089 (0.0009) | 0.0076 (0.0010) | 0.0075 (0.0017) | 0.0060 (0.0008) |
| | 2-Stage (NN) | **0.0144 (0.0029)** | **0.0058 (0.0007)** | **0.0042 (0.0005)** | **0.0025 (0.0002)** | **0.0019 (0.0001)** |
| | Separate (NN) | 0.3787 (0.0438) | 0.0354 (0.0068) | 0.0077 (0.0006) | 0.0043 (0.0004) | 0.0037 (0.0004) |
| | Top-FT (NN) | 0.0145 (0.0025) | 0.0066 (0.0007) | 0.0052 (0.0006) | 0.0035 (0.0004) | 0.0028 (0.0003) |
| 10 | Pool-w-L (NN) | 0.0192 (0.0049) | 0.0068 (0.0008) | 0.0054 (0.0015) | 0.0052 (0.0032) | 0.0027 (0.0009) |
| | Pooled (RF) | 0.0323 (0.0039) | 0.0217 (0.0013) | 0.0193 (0.0009) | 0.0179 (0.0006) | 0.0179 (0.0005) |
| | 2-Stage (RF) | 0.0253 (0.0034) | 0.0135 (0.0009) | 0.0105 (0.0006) | 0.0079 (0.0003) | 0.0073 (0.0003) |
| | Separate (RF) | 0.0501 (0.0064) | 0.0285 (0.0019) | 0.0229 (0.0011) | 0.0174 (0.0005) | 0.0159 (0.0004) |
| | Pool-w-L (RF) | 0.0322 (0.0041) | 0.0214 (0.0013) | 0.0190 (0.0009) | 0.0176 (0.0006) | 0.0176 (0.0005) |
| | Pooled (NN) | 0.0202 (0.0047) | 0.0101 (0.0011) | 0.0087 (0.0010) | 0.0089 (0.0029) | 0.0066 (0.0009) |
| | 2-Stage (NN) | 0.0194 (0.0049) | **0.0076 (0.0008)** | **0.0056 (0.0006)** | **0.0034 (0.0003)** | **0.0026 (0.0002)** |
| | Separate (NN) | 0.3739 (0.0474) | 0.0381 (0.0071) | 0.0101 (0.0009) | 0.0057 (0.0006) | 0.0048 (0.0006) |
| | Top-FT (NN) | **0.0191 (0.0048)** | 0.0080 (0.0008) | 0.0063 (0.0007) | 0.0041 (0.0004) | 0.0034 (0.0003) |
| 5 | Pool-w-L (NN) | 0.0244 (0.0047) | 0.0092 (0.0014) | 0.0067 (0.0009) | 0.0061 (0.0029) | 0.0032 (0.0006) |
| | Pooled (RF) | 0.0332 (0.0042) | 0.0219 (0.0012) | 0.0195 (0.0009) | 0.0177 (0.0006) | 0.0176 (0.0005) |
| | 2-Stage (RF) | 0.0270 (0.0037) | 0.0145 (0.0010) | 0.0114 (0.0007) | 0.0083 (0.0003) | 0.0075 (0.0003) |
| | Separate (RF) | 0.0514 (0.0066) | 0.0292 (0.0021) | 0.0235 (0.0011) | 0.0176 (0.0005) | 0.0159 (0.0004) |
| | Pool-w-L (RF) | 0.0331 (0.0043) | 0.0216 (0.0012) | 0.0191 (0.0009) | 0.0173 (0.0006) | 0.0173 (0.0005) |
| | Pooled (NN) | 0.0266 (0.0051) | 0.0132 (0.0016) | 0.0110 (0.0016) | 0.0103 (0.0033) | 0.0079 (0.0016) |
| | 2-Stage (NN) | 0.0280 (0.0053) | 0.0114 (0.0015) | **0.0081 (0.0010)** | **0.0050 (0.0004)** | **0.0040 (0.0003)** |
| | Separate (NN) | 0.3740 (0.0515) | 0.0448 (0.0090) | 0.0153 (0.0016) | 0.0087 (0.0009) | 0.0070 (0.0007) |
| | Top-FT (NN) | **0.0259 (0.0052)** | **0.0112 (0.0012)** | 0.0083 (0.0009) | 0.0055 (0.0005) | 0.0046 (0.0004) |
| 2 | Pool-w-L (NN) | 0.0302 (0.0045) | 0.0140 (0.0025) | 0.0099 (0.0016) | 0.0093 (0.0056) | 0.0052 (0.0014) |
| | Pooled (RF) | 0.0362 (0.0053) | 0.0234 (0.0014) | 0.0206 (0.0010) | 0.0178 (0.0006) | 0.0173 (0.0005) |
| | 2-Stage (RF) | 0.0321 (0.0049) | 0.0178 (0.0012) | 0.0142 (0.0008) | 0.0098 (0.0003) | 0.0086 (0.0003) |
| | Separate (RF) | 0.0571 (0.0074) | 0.0324 (0.0022) | 0.0256 (0.0012) | 0.0189 (0.0006) | 0.0169 (0.0004) |
| | Pool-w-L (RF) | 0.0359 (0.0055) | 0.0230 (0.0014) | 0.0201 (0.0011) | 0.0174 (0.0006) | 0.0169 (0.0005) |

*Table 17.* Average per-group MSE for Extra Scenario 4 over 50 independent trials at an SNR of 5 using neural networks. The best performance in terms of lower MSE is bolded.

| $n$ | Group | Pooled | 2-Stage | Separate | Top-FT | Pool-w-L |
|---|---|---|---|---|---|---|
| | 1 | 0.0294 (0.0171) | 0.0317 (0.0186) | 0.4555 (0.2110) | **0.0294 (0.0169)** | 0.0454 (0.0236) |
| | 2 | 0.0337 (0.0126) | **0.0298 (0.0134)** | 0.4091 (0.1315) | 0.0306 (0.0126) | 0.0342 (0.0103) |
| 1000 | 3 | 0.0190 (0.0071) | 0.0190 (0.0074) | 0.3641 (0.1010) | **0.0183 (0.0065)** | 0.0255 (0.0080) |
| | 4 | 0.0151 (0.0046) | 0.0154 (0.0049) | 0.3518 (0.0806) | **0.0149 (0.0046)** | 0.0205 (0.0065) |
| | 5 | 0.0173 (0.0048) | 0.0161 (0.0043) | 0.3653 (0.0886) | **0.0158 (0.0044)** | 0.0183 (0.0051) |
| | 1 | 0.0160 (0.0045) | 0.0148 (0.0046) | 0.3776 (0.1010) | **0.0147 (0.0046)** | 0.0178 (0.0062) |
| | 2 | 0.0193 (0.0046) | **0.0120 (0.0024)** | 0.0251 (0.0064) | 0.0150 (0.0032) | 0.0142 (0.0030) |
| 5000 | 3 | 0.0083 (0.0019) | 0.0074 (0.0016) | 0.0152 (0.0033) | **0.0071 (0.0014)** | 0.0096 (0.0019) |
| | 4 | 0.0065 (0.0014) | 0.0060 (0.0010) | 0.0116 (0.0027) | **0.0058 (0.0009)** | 0.0074 (0.0013) |
| | 5 | 0.0090 (0.0022) | **0.0058 (0.0011)** | 0.0094 (0.0019) | 0.0062 (0.0010) | 0.0066 (0.0015) |
| | 1 | 0.0146 (0.0039) | **0.0102 (0.0028)** | 0.0240 (0.0064) | 0.0124 (0.0032) | 0.0122 (0.0028) |
| | 2 | 0.0178 (0.0045) | **0.0090 (0.0016)** | 0.0163 (0.0033) | 0.0121 (0.0025) | 0.0106 (0.0024) |
| 10000 | 3 | 0.0070 (0.0021) | **0.0053 (0.0009)** | 0.0100 (0.0013) | 0.0053 (0.0009) | 0.0067 (0.0013) |
| | 4 | 0.0053 (0.0010) | 0.0046 (0.0008) | 0.0076 (0.0011) | **0.0043 (0.0005)** | 0.0055 (0.0012) |
| | 5 | 0.0076 (0.0022) | **0.0044 (0.0010)** | 0.0068 (0.0012) | 0.0050 (0.0007) | 0.0050 (0.0010) |
| | 1 | 0.0141 (0.0042) | **0.0068 (0.0011)** | 0.0129 (0.0025) | 0.0093 (0.0016) | 0.0095 (0.0036) |
| | 2 | 0.0164 (0.0061) | **0.0053 (0.0008)** | 0.0088 (0.0011) | 0.0082 (0.0014) | 0.0091 (0.0043) |
| 30000 | 3 | 0.0066 (0.0039) | **0.0033 (0.0004)** | 0.0061 (0.0009) | 0.0035 (0.0005) | 0.0060 (0.0026) |
| | 4 | 0.0058 (0.0033) | **0.0027 (0.0003)** | 0.0043 (0.0011) | 0.0028 (0.0003) | 0.0052 (0.0030) |
| | 5 | 0.0086 (0.0054) | **0.0025 (0.0003)** | 0.0038 (0.0009) | 0.0030 (0.0005) | 0.0049 (0.0028) |
| | 1 | 0.0122 (0.0025) | **0.0050 (0.0008)** | 0.0110 (0.0021) | 0.0084 (0.0014) | 0.0062 (0.0015) |
| | 2 | 0.0149 (0.0034) | **0.0040 (0.0006)** | 0.0071 (0.0009) | 0.0070 (0.0013) | 0.0051 (0.0009) |
| 50000 | 3 | 0.0049 (0.0020) | **0.0026 (0.0003)** | 0.0050 (0.0012) | 0.0027 (0.0003) | 0.0033 (0.0007) |
| | 4 | 0.0034 (0.0011) | **0.0021 (0.0004)** | 0.0036 (0.0009) | 0.0022 (0.0003) | 0.0026 (0.0007) |
| | 5 | 0.0057 (0.0025) | **0.0020 (0.0003)** | 0.0036 (0.0013) | 0.0024 (0.0004) | 0.0024 (0.0006) |

*Table 18.* Average MSE for Scenario 3 (high-dimensional) with different SNR levels over 50 independent trials. The best performance in terms of lower MSE within each SNR group is bolded.

| Std | Model / Sample size | 5000 | 10000 | 30000 | 50000 |
|-----|---------------------|------|-------|-------|-------|
| 0.1 | Pooled (NN) | 3.5772 (0.4119) | 2.8049 (0.3413) | 2.4122 (0.3941) | 1.8712 (0.2405) |
| | 2-Stage (NN) | **3.4045 (0.3687)** | **2.5317 (0.2908)** | **1.7826 (0.2254)** | **1.4377 (0.1909)** |
| | Separate (NN) | 18.4700 (1.3653) | 16.7562 (1.0588) | 7.3540 (0.6464) | 5.1880 (0.3722) |
| | Top-FT (NN) | 3.4864 (0.3970) | 2.6559 (0.3170) | 2.0301 (0.2903) | 1.6499 (0.2149) |
| | Pool-w-L (NN) | 3.7586 (0.4649) | 2.6799 (0.3126) | 2.0207 (0.3539) | 1.5020 (0.2305) |
| | ptLasso | 19.2066 (1.1992) | 18.1132 (1.1836) | 17.7140 (0.8321) | 17.5210 (0.6882) |
| 1 | Pooled (NN) | 4.0795 (0.4407) | 3.2150 (0.3589) | 2.6265 (0.3937) | 2.1231 (0.2551) |
| | 2-Stage (NN) | **3.9457 (0.4051)** | **2.9710 (0.3132)** | **2.1172 (0.2096)** | **1.7750 (0.1908)** |
| | Separate (NN) | 18.5199 (1.4433) | 16.8376 (1.1317) | 8.0144 (0.6275) | 5.7036 (0.4192) |
| | Top-FT (NN) | 3.9666 (0.4319) | 3.0476 (0.3328) | 2.2643 (0.2707) | 1.8705 (0.2096) |
| | Pool-w-L (NN) | 4.3708 (0.4846) | 3.1855 (0.3481) | 2.3404 (0.2989) | 1.8470 (0.2286) |
| | ptLasso | 20.1427 (1.3865) | 19.1391 (1.2099) | 18.7576 (0.8002) | 18.5122 (0.7150) |

*Table 19.* Average MSE for Scenario 4 (high-dimensional) with different SNR levels over 50 independent trials. The best performance in terms of lower MSE within each SNR group is bolded.

| Std | Model / Sample size | 5000 | 10000 | 30000 | 50000 |
|-----|---------------------|------|-------|-------|-------|
| 0.1 | Pooled (NN) | 8.5086 (0.8878) | 5.8025 (0.5703) | 3.8477 (0.6878) | 2.9454 (0.3160) |
| | 2-Stage (NN) | **8.4782 (0.8924)** | **5.6875 (0.5791)** | **3.2175 (0.3005)** | **2.5256 (0.2120)** |
| | Separate (NN) | 22.0682 (1.5491) | 21.4115 (1.0980) | 14.8534 (0.6357) | 13.0925 (0.7355) |
| | Top-FT (NN) | 8.5038 (0.8945) | 5.7637 (0.5768) | 3.5023 (0.3075) | 2.8225 (0.2286) |
| | Pool-w-L (NN) | 8.5776 (0.8611) | 5.8783 (0.5824) | 3.8614 (0.7056) | 2.9559 (0.2845) |
| | ptLasso | 21.7040 (1.6185) | 19.7228 (0.9760) | 19.2816 (0.8314) | 19.2234 (0.7995) |
| 1 | Pooled (NN) | 9.1348 (0.9145) | 6.4555 (0.6181) | 4.3988 (0.5348) | 3.5638 (0.2752) |
| | 2-Stage (NN) | **9.0841 (0.9062)** | **6.3287 (0.5981)** | **3.8919 (0.3088)** | **3.1943 (0.2400)** |
| | Separate (NN) | 22.3329 (1.5473) | 21.5863 (0.9121) | 15.1587 (0.5992) | 13.5028 (0.7384) |
| | Top-FT (NN) | 9.1156 (0.9069) | 6.4022 (0.5908) | 4.1114 (0.3084) | 3.4415 (0.2636) |
| | Pool-w-L (NN) | 9.2112 (0.9200) | 6.4724 (0.6281) | 4.5905 (0.6670) | 3.5074 (0.3140) |
| | ptLasso | 21.7648 (1.6522) | 19.7493 (0.9892) | 19.3060 (0.8403) | 19.2003 (0.8071) |

## F.2. Detailed Configurations for Numerical Simulation

This section presents the detailed configurations used in our numerical simulations. Regarding the neural network setups, the Pooled, Pool-w-L, and Top-FT strategies each employ a single neural network instance, whereas the 2-Stage strategy involves two networks: one trained on the full dataset and another trained separately for each group. Within each scenario, all NN instances share the same network architecture.

For Scenarios 1,2 and Extra Scenarios 1–4, which correspond to the low-dimensional settings, each NN is implemented as a four-layer MLP with 64 hidden units per layer, using ReLU activations after the first three layers. For the high-dimensional settings, we adopt a five-layer MLP with 128 hidden units per layer for Scenario 3, and an eight-layer MLP with 256 hidden units per layer for Scenario 4. All neural networks are trained using the Adam optimizer with a learning rate of $10^{-3}$ and a batch size of 256. For each setting, data are randomly partitioned using a group-wise stratified split into training (70%), validation (15%), and testing (15%) subsets. Early stopping is applied with a patience of 10 epochs, based on validation performance.

For the competing methods, random forests are implemented with 100 tree estimators and a maximum tree depth of 10, while other hyperparameters follow the default settings from the `sklearn.ensemble` package. For ptLasso, we perform 10-fold cross-validation over $\alpha \in \{0, 0.25, 0.5, 0.75, 1\}$, keeping other parameters at their default values.

## F.3. Experiments with Highly Unbalanced Groups

To further investigate the robustness of our proposed framework under highly unbalanced groups, we examine the six low-dimensional scenarios previously introduced with unbalanced group assignment proportions of [0.02, 0.25, 0.25, 0.24, 0.24], where the first group represents only 2% of the total sample. Here we focus on the DNN realization and present

our experimental results at SNR=5 across total sample sizes $n \in \{5000, 10000, 30000, 50000\}$. Table 20 summarizes the MSE performance of the unbalanced group (Group 1) over 50 independent Monte Carlo trials across the six simulation scenarios. The results demonstrate that as the sample size increases, the proposed two-stage transfer learning method yields increasingly pronounced performance improvements for the extremely low-proportion group, achieving the best performance in most cases.

*Table 20.* Average MSE for the unbalanced group (Group 1) across six low-dimensional scenarios at $\mathrm{SNR} = 5$ over 50 independent trials. The best performance for each sample size is highlighted in bold.

| | Model / Sample size | 5000 | 10000 | 30000 | 50000 |
|---|---|---|---|---|---|
| | Pooled (NN) | 0.0339 (0.0164) | 0.0333 (0.0113) | 0.0297 (0.0047) | 0.0308 (0.0065) |
| | 2-Stage (NN) | 0.0327 (0.0179) | **0.0184 (0.0061)** | **0.0088 (0.0018)** | **0.0060 (0.0009)** |
| Scenario 1 | Separate (NN) | 0.6937 (0.4308) | 0.7003 (0.3028) | 0.0236 (0.0047) | 0.0176 (0.0020) |
| | Top-FT (NN) | **0.0236 (0.0099)** | 0.0232 (0.0099) | 0.0132 (0.0039) | 0.0131 (0.0041) |
| | Pool-w-L (NN) | 0.0334 (0.0149) | **0.0184 (0.0050)** | 0.0155 (0.0076) | 0.0108 (0.0038) |
| | Pooled (NN) | **0.0488 (0.0211)** | 0.0456 (0.0144) | 0.0437 (0.0124) | 0.0375 (0.0086) |
| | 2-Stage (NN) | 0.0498 (0.0210) | 0.0415 (0.0199) | **0.0250 (0.0074)** | **0.0185 (0.0050)** |
| Scenario 2 | Separate (NN) | 0.6019 (0.2203) | 0.5966 (0.1470) | 0.1108 (0.0858) | 0.0571 (0.0147) |
| | Top-FT (NN) | 0.0514 (0.0257) | **0.0410 (0.0121)** | 0.0358 (0.0073) | 0.0312 (0.0067) |
| | Pool-w-L (NN) | 0.0918 (0.0427) | 0.0594 (0.0255) | 0.0413 (0.0223) | 0.0286 (0.0069) |
| | Pooled (NN) | 0.0206 (0.0103) | 0.0186 (0.0061) | 0.0169 (0.0061) | 0.0163 (0.0049) |
| | 2-Stage (NN) | 0.0160 (0.0078) | **0.0112 (0.0038)** | **0.0053 (0.0020)** | **0.0042 (0.0011)** |
| Extra Scenario 1 | Separate (NN) | 0.4917 (0.2299) | 0.4940 (0.1773) | 0.0145 (0.0032) | 0.0113 (0.0016) |
| | Top-FT (NN) | **0.0156 (0.0069)** | 0.0146 (0.0051) | 0.0099 (0.0021) | 0.0085 (0.0020) |
| | Pool-w-L (NN) | 0.0200 (0.0096) | 0.0128 (0.0047) | 0.0095 (0.0050) | 0.0060 (0.0018) |
| | Pooled (NN) | 0.0344 (0.0184) | 0.0337 (0.0119) | 0.0297 (0.0079) | 0.0291 (0.0043) |
| | 2-Stage (NN) | 0.0355 (0.0156) | **0.0167 (0.0064)** | **0.0089 (0.0015)** | **0.0064 (0.0016)** |
| Extra Scenario 2 | Separate (NN) | 0.6453 (0.3275) | 0.6625 (0.3514) | 0.0228 (0.0029) | 0.0177 (0.0032) |
| | Top-FT (NN) | **0.0255 (0.0147)** | 0.0187 (0.0086) | 0.0133 (0.0028) | 0.0120 (0.0029) |
| | Pool-w-L (NN) | 0.0321 (0.0112) | 0.0182 (0.0035) | 0.0116 (0.0044) | 0.0098 (0.0038) |
| | Pooled (NN) | 0.1963 (0.0784) | 0.1868 (0.0610) | 0.2073 (0.0686) | 0.2012 (0.0375) |
| | 2-Stage (NN) | **0.0696 (0.0393)** | **0.0545 (0.0231)** | **0.0231 (0.0061)** | **0.0182 (0.0049)** |
| Extra Scenario 3 | Separate (NN) | 1.9146 (0.7185) | 1.8240 (0.4825) | 0.0644 (0.0147) | 0.0505 (0.0119) |
| | Top-FT (NN) | 0.0747 (0.0376) | 0.0703 (0.0223) | 0.0534 (0.0134) | 0.0494 (0.0102) |
| | Pool-w-L (NN) | 0.0710 (0.0376) | 0.0562 (0.0214) | 0.0407 (0.0142) | 0.0267 (0.0074) |
| | Pooled (NN) | **0.0149 (0.0070)** | 0.0143 (0.0062) | 0.0121 (0.0036) | 0.0100 (0.0017) |
| | 2-Stage (NN) | 0.0174 (0.0098) | 0.0124 (0.0060) | **0.0081 (0.0020)** | **0.0065 (0.0013)** |
| Extra Scenario 4 | Separate (NN) | 0.4438 (0.2062) | 0.4219 (0.1147) | 0.0263 (0.0059) | 0.0197 (0.0063) |
| | Top-FT (NN) | 0.0155 (0.0076) | **0.0123 (0.0050)** | 0.0084 (0.0017) | 0.0078 (0.0015) |
| | Pool-w-L (NN) | 0.0283 (0.0128) | 0.0180 (0.0068) | 0.0122 (0.0047) | 0.0083 (0.0023) |

# G. Details of Real Data Experiments

## G.1. Beijing PM2.5 Dataset

In this section, we evaluate our framework on an air quality prediction task using an open-source dataset provided by (Chen, 2015) and following the study of (Liang et al., 2015). This dataset records hourly meteorological data from Beijing, China over several years, with particulate matter with a diameter of 2.5 micrometers (PM2.5), a central index for air pollution levels, as the predicted response variable. Specifically, the dataset spans January 1, 2010 to December 31, 2014, containing 43824 hourly records after removing missing values. Features include six numerical meteorological covariates (dew point, temperature, pressure, cumulated wind speed, and cumulated hours of snow and rain) and one categorical variable for dominant wind direction with four categories: northeast (NE), northwest (NW), south (S), and calm (CV). Due to Beijing's geography with mountains to the north and west and industrial zones to the south and east, wind direction significantly influences air quality by determining whether clean air flows in or pollutants become trapped (Liang et al., 2015). This makes wind direction a natural group label, as the PM2.5-meteorological relationship varies substantially across wind directions. We conduct transfer learning based on these groups, with 4997, 14150, 9387, and 15290 samples for NE, NW, CV, and S winds, respectively.

We specify a similar autoregressive model to that stated in (Liang et al., 2015) equation (4.3) for predicting PM2.5 as

follows:

$$y_t = f_{z_t}(x_t, y_{t-1}) + \epsilon_t,$$

where $y_t$ is the PM2.5 concentration at hour $t$, and $y_{t-1}$ is the lag-1 PM2.5 value to capture temporal dependencies. The covariate vector $x_t$ includes the six numerical meteorological variables, along with four additional variables from sine and cosine transformations of the hour-of-day and month-of-year to help capture diurnal and seasonal patterns. The group label $z_t$ corresponds to one of the four dominant wind directions.

We follow a similar setting to the low-dimensional numerical experiments, where we compare DNN and RF as estimators across various learning strategies. To prevent potential data leakage risks in time series prediction tasks, we use data from 2014 as the hold-out test set. We repeat the entire procedure for 20 independent trials using different random seeds and report the average test MSE across trials. For the neural network implementations, each model instance is a five-layer MLP with 64 neurons per layer, using ReLU activation functions after the first four layers. All networks are trained with the Adam optimizer at a learning rate of $10^{-3}$, a batch size of 256, and early stopping with a patience of 10 epochs based on validation performance. For the random forest models, we use 100 decision trees with a maximum depth of 10, while other hyperparameters remain at their default settings. All input features are standardized to have zero mean and unit variance.

*Table 21.* Overall and per-group average MSE for PM2.5 models over 20 independent trials. The best performance in terms of lower MSE within each group is bolded.

| Model | Overall | NE | NW | S | CV |
|---|---|---|---|---|---|
| Pooled (NN) | 0.0555 (0.0023) | 0.0509 (0.0024) | 0.0566 (0.0014) | 0.0617 (0.0030) | 0.0469 (0.0046) |
| 2-Stage (NN) | **0.0519 (0.0007)** | 0.0486 (0.0016) | **0.0548 (0.0024)** | **0.0573 (0.0009)** | **0.0418 (0.0007)** |
| Separate (NN) | 0.0541 (0.0009) | 0.0555 (0.0039) | 0.0561 (0.0027) | 0.0591 (0.0010) | 0.0434 (0.0011) |
| Top-FT (NN) | 0.0529 (0.0011) | 0.0481 (0.0017) | 0.0561 (0.0036) | 0.0581 (0.0010) | 0.0435 (0.0019) |
| Pool-w-L (NN) | 0.0538 (0.0023) | 0.0521 (0.0020) | 0.0557 (0.0031) | 0.0585 (0.0022) | 0.0452 (0.0039) |
| Pooled (RF) | 0.0556 (0.0002) | 0.0514 (0.0004) | 0.0581 (0.0005) | 0.0626 (0.0004) | 0.0437 (0.0004) |
| 2-Stage (RF) | 0.0534 (0.0003) | **0.0456 (0.0004)** | 0.0555 (0.0005) | 0.0606 (0.0006) | 0.0438 (0.0005) |
| Separate (RF) | 0.0580 (0.0002) | 0.0502 (0.0005) | 0.0602 (0.0005) | 0.0643 (0.0004) | 0.0494 (0.0010) |
| Pool-w-L (RF) | 0.0542 (0.0002) | 0.0503 (0.0004) | 0.0558 (0.0005) | 0.0610 (0.0003) | 0.0435 (0.0005) |

From Table 21, DNN trained with two-stage transfer learning achieves the lowest overall prediction error and wins in most subgroup metrics, highlighting the effectiveness of our proposed training strategy in low-dimensional settings. Notably, the two-stage training strategy also performs best when applied to RF, demonstrating the potential applicability of our framework and theorems to other nonparametric estimators.

### G.2. Detailed Configurations for UTKFace Experiment Using Pretrained Model

As required by the pretrained face feature extraction model, all images were resized to $160 \times 160 \times 3$ resolution using OpenCV (Bradski, 2000), with pixel values normalized to the range $[0, 1]$ prior to feature extraction. For the neural networks used for age regression based on the extracted features, in addition to the MLP architecture described in the main text, each MLP instance was trained using the Adam optimizer with an initial learning rate of $10^{-4}$. The model was trained under a cosine annealing learning rate schedule (Loshchilov & Hutter, 2016), where the rate smoothly decreased to $10^{-6}$ over the course of training. Early stopping was applied with a patience of 5 epochs based on the validation loss.

### G.3. Complementary Experiments on the UTKFace Dataset

**Joint Grouping by Ethnicity and Gender.** In addition to ethnicity, gender is another potential factor that may influence facial characteristics and consequently affect age estimation. To further investigate this, we conducted complementary experiments where both gender and ethnicity were considered simultaneously, resulting in 10 subgroups. The following table presents the results, with all other configurations kept consistent with those in the experiments that considered only ethnicity. As shown in the table, the proposed two-stage transfer learning method improves models' performance on each subgroup in comparison with the pooled training, achieving the lowest average MSE both overall and across most subgroups.

*Table 22.* Overall and per-group average MSE for age regression models over 20 independent trials, where both ethnicity and gender are used as grouping factors. Standard deviation is computed across seeds.

| Group | Pooled | 2-Stage | Separate | Top-FT | Pool-w-L |
|---|---|---|---|---|---|
| Overall | 59.1 (2.6) | **56.4 (2.1)** | 63.4 (3.2) | 58.8 (2.5) | 57.0 (1.8) |
| White-M | 63.8 (4.8) | **61.2 (4.2)** | 65.0 (6.6) | 63.6 (4.7) | 62.0 (4.3) |
| White-F | 77.2 (5.0) | **71.7 (3.3)** | 78.5 (8.2) | 76.7 (4.8) | 73.7 (4.3) |
| Black-M | 58.0 (5.8) | **56.4 (6.0)** | 62.3 (7.0) | 58.0 (5.8) | 56.5 (6.0) |
| Black-F | 57.9 (6.8) | 57.1 (6.6) | 68.0 (5.5) | 57.7 (6.8) | **56.3 (5.5)** |
| Asian-M | 49.1 (12.6) | **47.4 (12.7)** | 60.0 (12.1) | 48.8 (12.5) | 48.3 (12.2) |
| Asian-F | 34.7 (8.4) | **32.0 (8.2)** | 39.4 (9.8) | 34.4 (8.2) | 33.7 (8.4) |
| Indian-M | 52.5 (3.7) | 51.0 (4.1) | 58.0 (6.3) | 52.2 (3.9) | **50.2 (4.0)** |
| Indian-F | 43.1 (6.4) | 41.0 (5.3) | 50.0 (9.9) | 43.1 (6.2) | **40.8 (5.8)** |

**Direct Modeling with MLP.** As discussed in the main text, we can also directly employ MLPs to perform age estimation from raw image inputs. As a toy experiment, all images are resized to a resolution of $32 \times 32 \times 3$ using OpenCV, with pixel values normalized to the range $[0, 1]$. Each image is then flattened into a 3,072-dimensional feature vector, which serves as the input $x_i$.

We compare five learning strategies using MLPs as nonparametric estimator, following the same data-splitting setting as before. For each neural network instance, we employ an MLP with six layers and ReLU activation functions. Specifically, the input vector has 3,072 dimensions for all strategies except Pooled-with-Label, where we concatenate the one-hot encoded ethnicity feature with the image vector, resulting in a 3,077-dimensional input. This input passes through layers with 2048, 1024, 512, 256, 128, and 64 units, respectively, and the network concludes with a scalar output for age prediction. We adopt the AdamW optimizer ($\text{lr} = 10^{-3}$, weight decay $= 10^{-4}$) and train the model under a cosine annealing learning rate schedule. Early stopping is applied with a patience of 10 epochs based on validation loss. In Table 23, we report the average MSE over 20 random trials, where the two-stage transfer learning achieves the lowest overall MSE and wins in the most subgroups.

*Table 23.* Overall and per-group average MSE for age regression models over 20 independent trials.

| Model | Overall | White | Black | Asian | Indian |
|---|---|---|---|---|---|
| Pooled (NN) | 140.8 (13.7) | 165.9 (13.0) | 135.2 (14.3) | 112.6 (16.4) | 107.8 (12.0) |
| 2-Stage (NN) | **136.7 (12.1)** | **161.2 (11.3)** | **129.9 (12.3)** | **110.9 (15.7)** | **104.6 (9.8)** |
| Pool-w-L (NN) | 139.3 (12.5) | 164.4 (10.6) | 134.0 (12.2) | 111.8 (18.8) | 105.4 (10.3) |
| Separate (NN) | 161.9 (14.8) | 180.0 (15.2) | 157.3 (16.1) | 147.5 (17.0) | 133.9 (14.3) |
| Top-FT (NN) | 140.3 (13.5) | 165.2 (13.0) | 135.1 (13.5) | 112.8 (16.8) | 107.0 (11.6) |

