# OpenReview forum: "Transfer Learning in Nonparametric Regression with Deep ReLU Networks"
_ICML.cc/2026/Conference — ICML 2026 regular_

### Official Review · Reviewer_55mL · 2026-03-01

**Soundness:** 3
**Presentation:** 4
**Significance:** 3
**Originality:** 3
**Overall Recommendation:** 4
**Confidence:** 4

**Summary:**

The authors study transfer learning for deep ReLU networks in a heterogeneous group setting under an additive decomposition of shared and group-specific functions. They follow a two-stage transfer learning approach, first pretraining on pooled data and then learning group-specific corrections. The paper provides theoretical convergence guarantees and derives rates for deep ReLU networks under hierarchical composition assumptions. Extensive experiments on simulated data and two real applications (Beijing PM2.5 and UTKFace) show that the proposed two-stage approach generally outperforms pooled, separate, fine-tuning baselines, as well as Random Forest and ptLasso, in terms of MSE.

**Compliance With Llm Reviewing Policy:**

Affirmed.

**Final Justification:**

The authors have clarified things that were previously not clear, and I expect them to add those clarifications to the paper. I  will keep my positive score.

**Key Questions For Authors:**

Please also refer to the weaknesses discussed above. I have two additional questions:

1. In Scenarios 1 and 2, the performance gap between 2-Stage NN and 2-Stage RF is quite large (with relative improvements around 77%–90%). While this strongly supports the proposed method, Random Forest is generally competitive on tabular data. Could authors provide some intuition for why RF performs much worse than the neural approach in these settings?

1. In the simulations, the predictors were all generated independently. Have authors considered setups with correlated covariates (like, multivariate normal with covariance)? Since correlation is common in practical datasets, it would be helpful to understand whether the proposed two-stage method maintains its advantage under correlated designs.

**Limitations:**

The paper does not discuss its limitations. A brief discussion clarifying the scope of the additive assumption and the conditions under which the method may not perform well would improve the completeness of the presentation.

**Strengths And Weaknesses:**

Strengths:
1. The paper is well-written and easy to follow.
1. It provides solid convergence guarantees for general nonparametric estimators and specializes to deep ReLU networks under structured assumptions.
1. It demonstrates consistent improvements over pooled, separate, fine-tuning baselines, as well as RF and ptLasso.

Weaknesses:
1. The methodological novelty seems somewhat incremental. The two-stage procedure is very close to classical additive transfer formulations. While the theoretical analysis for deep ReLU networks is technically nontrivial, it is not entirely clear how much improvement the resulting convergence rates offer over existing kernel-based or other deep neural network transfer approaches, or under what regimes the gain is substantial.

1. Although the related work discusses several established nonparametric transfer learning methods (including kernel-based approaches and other deep nonparametric transfer frameworks), these methods are not included in the experimental comparisons. Without benchmarking against representative prior transfer methods, it is difficult to precisely delineate the empirical contribution of the proposed approach.

1. The paper claims non-asymptotic bounds, but the main theorems are only presented in convergence in probability. As written, the results read more like asymptotic convergence rates.

1. The final rate (e.g., simplified into $\phi_n + \phi_{l,n}$ ) is still difficult to interpret. It would be helpful to provide a concrete instance that simplify the rate under specific settings. For example, specializing the rate using the simulation setup or the example on page 12 and presenting the resulting simplified form would make the theoretical contribution more transparent.

1. Most of the main experiments focus on relatively large sample sizes (e.g., $n\geq 5000$, also $n=1000$ in appendix), where neural networks are generally expected to perform well. While this is reasonable, it would be informative to include more comprehensive evaluations at smaller sample sizes (like 100, 200, 500?) to assess how stable and effective the two-stage neural approach remains in more data-limited settings.

1. The paper does not address the issue of negative transfer. Since negative transfer is a well-known concern in transfer learning, it would strengthen the contribution to clarify under what violations of the additive assumption the proposed method might underperform separate training. It would also be helpful to discuss whether any assumptions or theoretical guarantees in the framework help prevent negative transfer.

---

> ### Author Rebuttal · Authors · 2026-03-29
>
> We thank the reviewer for the positive feedback and respond to each point below. Due to space limits, we refer to other reviewers' responses and use abbreviated citations as in the main text.
>
> **Weakness**
>
> Q1. We refer to the response to Weakness Q1 (Reviewer YeVm) for a thorough clarification of our contributions to prior work.
>
> Q2. Our baselines include various training strategies on NN and RF, and also compare with ptLasso. Tables 1–2 present experiment result on 2 additional competitors (https://anonymous.4open.science/r/anonymousfiles-6EDA/1.pdf):
>
> 1. Multi-task kernel ridge regression (Wang et al., 2016) on Scenarios 1–2 (low-dimensional; computationally prohibitive for $n>10000$), with central+offset penalty, RBF kernel, and grid-searched hyperparameters.
>
> 2. (Jiao et al., 2024) on Scenarios 1–3 (Scenario 3 as a high-dimensional setting), with NN architecture matched to ours. Their method resembles last-layer FT but requires a more complex training procedure and a shared-representation assumption.
>
> We find our method outperforms both across all settings.
>
> Q3. We rephrased "non-asymptotic bounds" as "risk bounds" in the revised version.
>
> Q4. For the final rate $\phi_n+\phi_{\ell,n}$, $\phi_n$ is the rate for the first-stage target $\bar f=f_0(x) + \sum_{\ell'=1}^{L} \mathbb{P}(Z=\ell' \mid X=x)f_{0,\ell'}(x)$, and $\phi_{\ell,n}$ for the second-stage offset $G_\ell=f_{0,\ell}(x) - \sum_{\ell'=1}^{L} \mathbb{P}(Z=\ell' \mid X=x)f_{0,\ell'}(x)$. Both $\bar f$ and $G_\ell$ are assumed to satisfy hierarchical composition as Definition A.2.
>
> Now consider a 2-group example ($\pi_1+\pi_2=1$) with 10-d input. Each group's regression function follows structure in Figures 1–2 (https://anonymous.4open.science/r/anonymousfiles-6EDA/1.pdf). Both groups share $g_1^{(1)},g_2^{(1)},g_1^{(2)}$ with input dimension $K=2,3,3$ and smoothness $p=1.5,1.5,2.5$, respectively. The 2 groups differ only in $g_{3,group}^{(1)}$, with $K=2,p=1.5$ for both. Plugging into our rate, estimating $\bar{f}=\pi_1g_1+\pi_2g_2$ keeps the same composition structure, giving an upper bound
>
> $\phi_n=\max_{(p,K)\in\lbrace(1.5,2),(1.5,3),(1.5,2),(2.5,3)\rbrace}n^{-2p/(2p+K)}= n^{-1/2}.$
>
> The offset $G_1=\pi_2(g_1-g_2)$ cancels shared components, reducing to a 2-layer model with a single branch, giving an upper bound
>
> $\phi_{1,n}=\max_{(p,K)\in \lbrace(1.5,2),(2.5,1)\rbrace}
> (n\pi_1)^{-2p/(2p+K)}=(n\pi_1)^{-3/5}$.
>
> The overall rate $n^{-1/2}+(n\pi_1)^{-3/5}$ depends only on intrinsic dimensions, not the ambient dimension 10, whereas standard estimators treating all 10 dimensions yield an upper bound with slower rate $n^{-3/13}+(n\pi_1)^{-3/13}$.
>
> Q5. Our simulations include $n=1000$, which after train/val/test splitting yields $\approx700$ training samples; with unbalanced 5-group assignment, the smallest group has $\approx47$ samples. We acknowledge no clear gain in this regime, which is expected: DNNs are known to underperform in extremely small-sample settings, hindering second-stage group adaptation. Though we consider this acceptable as modern datasets for DNNs are typically large in practice, we now discuss this limitation in a dedicated limitations section.
>
> Q6. In our setting, negative transfer is theoretically possible when the offset is highly complex or first-stage aggregation yields a complicated $\bar{f}$; empirically, however, many datasets exhibit strong cross-group similarity (as in our real data experiments), since our transfer operates at the group level rather than the dataset level. We include this in limitations and leave formal detection of negative transfer as future work.
>
> **Key Questions**
>
> Q1. Across all experiments within the RF class, RF trained with our strategy consistently outperforms competitors, and RF is able to perform competitively with NN on PM2.5 dataset. The large gap in simulation between 2-Stage RF and NN stems from estimator intrinsic properties, with DNNs known to have strong capacity for modeling complex and high-dimensional functions.
>
> Q2. In high-dimensional Scenarios 3–4, covariates are generated via a low-dimensional latent variable mapped through a neural network, inducing complex dependence structures; our method remains effective in these settings. We additionally ran Scenarios 1–2 with correlated covariates: we sample 10-d $Z\sim\mathcal{N}(0,\Sigma)$ with $\Sigma=(1-\rho)I+\rho\mathbf{1}\mathbf{1}^\top$, $\rho=0.2$, and transform via $X=\Phi(Z)$ to obtain correlated covariates. Table 4–5 (https://anonymous.4open.science/r/anonymousfiles-6EDA/1.pdf) show our method maintains strong performance. On UTKFace, images are inherently highly correlated, and our method achieves best overall performance in various settings (Section 4.2 Appendix E.3).
>
> **Limitations**: We will add a section of limitations covering additive assumption, small-sample performance of NNs, negative TL, and our connection to the practically popular last-layer FT, with developing theory for the last 2 points as future work.

---

> > ### Author Rebuttal · Reviewer_55mL · 2026-04-02
> >
> > Thanks for your responses. I do not have further questions and will keep my score.

---

> > > ### Author Response · Authors · 2026-04-05
> > >
> > > Thank you again for your helpful comments!

---

### Official Review · Reviewer_65D9 · 2026-03-04

**Soundness:** 3
**Presentation:** 4
**Significance:** 4
**Originality:** 3
**Overall Recommendation:** 5
**Confidence:** 4

**Summary:**

The paper is about nonparametric regression in a transfer learning setting for data with heterogenous groups. A general model and estimation framework is presented and specialized to deep ReLu neural networks. The main theoretical results derive onvergence rates for the resulting estimators. Experiments show that the method performs better than alternative estimation strategies.

**Compliance With Llm Reviewing Policy:**

Affirmed.

**Final Justification:**

While the paper is not ground-breaking or overly flashy/innovative, the proposed methodology and theoretical bounds are meaningful, and the paper was executed with appropriate care. Smaller technical issues I raised were addressed, so I am fine with the acceptance of the paper.

**Key Questions For Authors:**

1. The abstract and introduction say you establish non-asymptotic bounds. The bounds and some regularity conditions presented are asymptotic, however. Please rephrase.
2. Are the bounds really of order $o_{\mathbb P}$ and not $O_{\mathbb P}$? By the way, you may want to define the $o_{\mathbb P}$ notation in Section 1.2.
3. Your related work paragraph on DNNs is about overcoming the curse of dimensionality using hierarchical compositions and manifold assumptions. You may want to include two works that combine these two assumptions for ReLU networks [[1]](https://proceedings.mlr.press/v267/schulte25a.html) [[2]](https://arxiv.org/abs/2602.03539).
4. The contributions section refers to Theorems from the appendix as main results. I recommend referring only to the main text results here. Similarly, condition (20) is mentioned in the main text, but the corresponding paragraphs cannot be understood without referring to the appendix. This should be improved.

**Limitations:**

yes

**Strengths And Weaknesses:**

**Strenghts**

* The paper addresses an important problem and proposes a well-thought-out method. While the framework appears in a similar form in previous work, its general formulation and specialization to neural networks have great merit.
* As far as I checked, the (non-trivial) proofs are carried out with care and correctly.
* The paper is well written and easy to follow.

**Weaknesses**

* Some minor issues need clarification or could be improved; see the questions below.

---

> ### Author Rebuttal · Authors · 2026-03-30
>
> We appreciate the reviewer’s positive feedback and respond to each point below.
>
> **Key Questions**
>
> 1. We apologize for the imprecise terminology and have rephrased "non-asymptotic bounds" as "risk bounds" in the revised version.
>
> 2. We thank the reviewer for this question. We have made a typo,  our bounds hold in  $O_{\mathbb{P}}$ sense. For a sequence of random variables $X_n$, $X_n=O_p(a_n)$ means that for any $\varepsilon>0$, there exist $M>0$ and $N>0$ such that $\mathbb{P}(X_n > M a_n) < \varepsilon$ for all $n>N$. We have added these definitions to Section 1.2. However, our bounds are stronger than $O_{\mathbb{P}}$, in fact they hold in the sense that $\lim_{n\to\infty}\mathbb{P}(|X_n/a_n|\ge M)=0$, for a positive constant $M$ whenever we mean $X_n = O_{\mathbb{P}}(a_n)$.
>
> 3. We thank the reviewer for pointing out these recent results. We have added the citations and are interested in extending our TL theory building on these works in future research.
>
> 4. We appreciate this suggestion. Due to the 8-page limit, we had to adopt the current layout. If the paper is accepted, we will further improve the readability, as the page limit will be extended to 9 pages.

---

> > ### Author Rebuttal · Reviewer_65D9 · 2026-04-01
> >
> > All my questions have been resolved. I keep my positive evaluation.

---

> > > ### Author Response · Authors · 2026-04-02
> > >
> > > Thank you again for your helpful comments!

---

### Official Review · Reviewer_YeVm · 2026-03-06

**Soundness:** 2
**Presentation:** 1
**Significance:** 2
**Originality:** 1
**Overall Recommendation:** 2
**Confidence:** 4

**Summary:**

This paper studies transfer learning for nonparametric regression. Specifically,
1. The framework in this paper follows the offset transfer learning framework and assumes the regression function for each source is a summation of a shared function and a group-specific offset.
2. The authors propose a two-stage process for learning: first pooling all data to estimate the shared function and using the source-specific data to estimate the source-specific function.
3. This paper provides a general asymptotic upper bound for this two-stage estimation procedure, followed by concrete rates by applying deep ReLU networks as estimators in both stages.
4. Experiments on four simulation settings and two real datasets (Beijing PM2.5 and UTKFace age estimation) show the two-stage deep net estimator generally outperforms alternatives like pooled training, separate training, fine-tuning, and random forests.

**Compliance With Llm Reviewing Policy:**

Affirmed.

**Key Questions For Authors:**

1. Since the authors claim the general results in Theorem 3.2 are model-agnostic. Can authors provide concrete examples of how it recovers the results in previous transfer learning nonparametric regression with the kernel method or local polynomial estimator? It is well-known that the technique to bound error induced by using pseudo labels in stage 2 is typically model-specific.

2. Can you provide a concrete example of when both the shared and source-specific functions are simple but their addition is complex? In classical nonparametric theory, this is counterintuitive, but it would be interesting to see an example under the hierarchical compositional structure.

**Limitations:**

The authors mentioned some of the limitations in the paper, including why not study the minimax optimality like current transfer learning literature.

**Strengths And Weaknesses:**

### Strengths
1. The paper is generally well-organized and easy to follow.
2. The paper covers a wide range of experiments.


### Weaknesses
1. The two-stage offset transfer learning framework is well-established even just limited to the nonparametric regression field (Wang & Schneider 2015, Du et al. 2017, Lin & Reimherr 2024). The more interesting and promising investigation of transfer learning under deep networks should follow the paradigm in practice. For example, the simplest case is linear probing.

2. For the deep ReLU theory: The usage of hierarchical compositions is from Kohler & Langer (2021) and Padilla et al. (2024a), and deep ReLU networks ``mitigate'' the curse of dimension, which is directly from previous literature. The paper's contribution reduces to analyzing error propagation from Stage 1 to Stage 2, using similar proof techniques as prior work.

The conceptual and technical contributions are incremental. For a paper of such nature, it is more desired to see how combining different techniques results in interesting and new insights. I didn't see those in the current form of the paper.

3. The framework of this paper fits more in multi-task learning than in transfer learning. The setup treats all $L$ sources symmetrically, i.e., no source/target distinction, and the goal is to estimate $g_{0,\ell}$ for all sources simultaneously, which is the multi-task learning setting. Calling this "transfer learning" is somewhat misleading and disconnects the paper from the actual transfer learning problem (leveraging abundant source data for a data-scarce target).

4. Both the abstract and the introduction claim ''non-asymptotic,'' but the main results presented in the main paper are in the asymptotic regime. While theorems in Appendix B do present the somewhat ``non-asymptotic'' version, I think such writing is misleading.

---

> ### Author Rebuttal · Authors · 2026-03-29
>
> We thank the reviewer for the feedback. Due to space limits, we refer to other reviewers' responses at certain points and abbreviate citations as in the main text.
>
> **Weakness**
>
> Q1. We agree the two-stage transfer learning (TL) procedure relates to classical additive formulations in TL. Our core contributions are:
>
> (1) General theory for TL. We develop a general framework applicable to a broad class of base learners. We also establish a theorem for independent training with faster rates, which has not been established previously. A direct application of our theory recovers known rates for kernel ridge regression (Wang et al., 2016) up to logarithmic factors, and also yields new results for trend filtering (RJ Tibshirani 2014). We will provide a formal proof in the second-round rebuttal and in the final version.
>
> (2) New TL theory for neural networks (NNs) and connections to linear probing (Top-FT). To the best of our knowledge, we are the first to provide explicit TL upper bounds for NNs that can beat the curse of dimensionality, with regression function belongs to a broad hierarchical composition class. Existing works on TL with NNs typically assume a shared representation and are algorithmically similar to Top-FT; however, their nonparametric theoretical properties remain largely unexplored with strong assumptions: (Tripuraneni et al., 2020; Xu & Tewari, 2021) assume approximation to a neural network class without approximation error, while (Jiao et al., 2024) adopts a nonparametric perspective but cannot beat the curse of dimensionality. Our strategy is algorithmically similar but uses separate NN instances in the second stage, enabling explicit convergence rates under mild assumptions thus offers a theoretical advance toward understanding Top-FT. Lastly, we include Top-FT as a baseline in all experiments and leave its full characterization to future work.
>
> (3) New insights for TL. Beyond viewing TL as data augmentation for targets with scarce data, our theory also identifies a transfer mechanism: pooling all groups can lead to a simpler $\bar{f}$ via weighted averaging (Remark 3.5). Moreover, our framework accommodates multi-task settings where the number of groups grows with $n$, e.g. the number of tasks can scale as $\sqrt{n}$, with each task having $\sqrt{n}$ samples, providing a more general theory.
>
> Q2. We agree our analysis builds on similar proof techniques from the deep ReLU literature. However, existing works do not study the TL setting considered here, where the goal is to characterize when and how transfer can yield faster rates than training on a single group alone. A unified analysis of error propagation across two stages is conducted in this transfer setting. In particular, we establish results for both using the same data in both stages (Theorem 3.4) and data splitting (Corollary 3.8). Notably, the data-splitting result yields improved rates in regimes where $\underline{\pi}_\ell\to0$, which has not been characterized in prior work. Also, our general theory can be provenly applied to other base learners such as KRR and trend filtering.
>
> Q3. We agree our strategy closely relates to multi-task learning (MTL). However, our theoretical analysis is more motivated by TL: we consider an overall data of size $n$ together with a specific target of size $n\pi_\ell$, demonstrating improved target performance by transferring shared structure. Our results cover both vanishing $\pi_\ell$ (data-scarce target) and constant $\pi_\ell$, interpolating between classical TL and MTL. We also note the distinction for MTL and TL is not sharply defined in the literature; works such as (Tripuraneni et al., 2020; Cai & Pu, 2024; Jiao et al., 2024) use TL terminology while studying multiple tasks of comparable sizes. We revised Section 1.1 to more clearly separate TL and MTL, putting especially all papers in the last paragraph of the original TL subsection into a dedicated MTL subsection with more cited works.
>
> Q4. We rephrased "non-asymptotic bounds" as "risk bounds" in the revised version.
>
> **Key Questions**
>
> Q1. Yes; we refer the reviewer to Weakness Q1, point (1) in this response for details.
>
> Q2. Consider $L=2$ groups with proportions $\pi_1+\pi_2=1$ and $g_1(x_1,\ldots,x_5) = f_0(x_1,x_2)+f_{01}(x_3,x_4,x_5), g_2(x_1,\ldots,x_5) = f_0(x_1,x_2)+f_{02}(x_3,x_4,x_5).$ Each $g_\ell$ depends on all 5 variables, so direct estimation incurs rates governed by dimension 5. Now suppose $\pi_1 f_{01}(x_3,x_4,x_5)+\pi_2 f_{02}(x_3,x_4,x_5) \approx c$ (constant). Then $\bar{f} = f_0(x_1,x_2) + \pi_1 f_{01} + \pi_2 f_{02} \approx f_0(x_1,x_2) + c$ effectively depends only on 2 variables, yielding faster first-stage rates. The offset $G_2 = \pi_2(f_{02}-f_{01})$ depends on 3 variables and becomes nearly zero when $\pi_2 \to 0$, making second-stage estimation easy. This shows the two-stage TL can yield much simpler targets than direct estimation of $g_{0,\ell}$ under a composited model. We will include this in the paper.

---

> > ### Author Rebuttal · Reviewer_YeVm · 2026-04-03
> >
> > The authors' response partially addressed my concerns. However, I remain somewhat unsatisfied with the response regarding Q1, as any work involving combinations (thus being incremental) could potentially claim novelty on such grounds. Especially as the selling point of the paper, such as "beat the curse of dimensionality", is directly inherited from the assumptions and techniques in previous work.
> >
> > Therefore, I maintain my original score.

---

> > > ### Author Response · Authors · 2026-04-04
> > >
> > > We thank the reviewer for the continued engagement. We respectfully disagree with the characterization that our contribution is incremental, and address the concern below.
> > >
> > > **Bringing DNN into nonparametric TL theory is non-trivial.**
> > > We agree that DNN's ability to beat the curse of dimensionality is an established property (Kohler & Langer, 2021; Padilla et al., 2024a). However, this property concerns single-task nonparametric regression. Our contribution is to bring DNN into the
> > > nonparametric transfer learning (TL) framework and show that this property is preserved, and moreover, that positive transfer yields faster rates than single-group DNN estimation.
> > >
> > > Crucially, existing nonparametric TL theories with concrete convergence rates (Wang et al., 2016; Du et al., 2017; Lin & Reimherr, 2024; Cai & Pu, 2024) are developed for specific classical estimators, namely orthogonal series, kernel smoothing, kernel ridge regression, and local polynomials, and **cannot be extended to DNNs.** Their proof techniques rely on estimator-specific properties that have no DNN counterpart. The DNN-based attempt, Jiao et al. (2024), requires data scarcity for positive transfer, yields an upper bound slower than standard Hölder minimax rate, imposes strong assumptions on network weights and response, and has worse empirical performance (Weakness Q2 of Reviewer 55mL). Thus, a substantial gap exists between existing nonparametric TL theory and the estimators most widely adopted in practice, namely DNN, and closing this gap is far from straightforward.
> > >
> > > **The reviewer's own assessment in the 1st-round rebuttal also supports this point.**
> > >
> > > In the 1st-round review, the reviewer stated
> > >
> > > >**The more interesting and promising investigation of TL under DNN should follow the paradigm in practice, e.g. linear probing.**
> > >
> > > We agree, and as discussed in our round-1 response (Weakness Q1, point (2)), our framework is algorithmically
> > > connected to linear probing, and our theory provides a step toward understanding it.
> > >
> > > Furthermore, in response to the reviewer's Key Question Q2, we provided an example where positive transfer arises from the hierarchical compositional structure in a way that is impossible for classical estimators, which the reviewer initially considered **"interesting and counterintuitive."** The reviewer did not raise objections to this example. Thus, it is inconsistent to acknowledge the interest of DNN-based TL theory and the compositional property in TL, while characterizing the contribution of establishing such theory as incremental.
> > >
> > > **Summary of contributions beyond the DNN instantiation.**
> > >
> > > We also briefly recapitulate our contributions to highlight that, beyond bringing DNNs into nonparametric TL theory, this work makes several additional contributions that the reviewer may have underweighted:
> > >
> > > 1. **General theoretical framework.** Theorem 3.2 provides upper bound for TL applicable to a broad class of estimators under mild complexity conditions. As a direct consequence, this general theory recovers existing kernel-based
> > > rates under Sobolev sieves (Wang et al., 2016), and yields new results for previously unstudied estimators such as trend
> > > filtering (RJ Tibshirani 2014). To our knowledge, no existing TL theory offers comparable generality in accommodating different base learners within a single framework.
> > >
> > > 2. **New transfer-specific insights.** In addition to leveraging pooled data for a data-scarce target and learning a simpler offset function, Remark 3.5 reveals that pooling all groups can naturally lead to a simpler first-stage estimation target $\bar{f}$ through weighted averaging of group deviations, which further strengthens the benefit of TL. This mechanism is especially relevant when the number of groups $L$ grows, e.g. at rate $\sqrt{n}$, a regime our theory explicitly supports. Such analysis is absent from most prior work, which mainly focuses on the source-target data scarcity regime.
> > >
> > > 3. **Two complementary theoretical results.** Theorems 3.4 and 3.7 account for offset learning using all data in both stages versus with data splitting, respectively. The former stays closer to practical implementations, while the latter provides a tighter upper bound by removing the $\pi_l^{-1}$ factor, yielding a substantially weaker condition for positive transfer. This comparison has not appeared in prior work.
> > >
> > > 4. **Extensive empirical validation.** We validated across 8 simulation scenarios with various SNR settings and 2 real datasets with different task configurations, evaluating both NN and random forests as base learners. In the round-1 rebuttal to Reviewer 55mL, we also provided additional comparisons with other competitors across further settings.
> > >
> > > We hope this clarifies that our contribution is not merely combining existing ingredients, but providing a unified TL analysis, supported by both theoretical and empirical evidence, for when two-stage transfer can be beneficial, particularly with DNN estimators.

---

### Official Review · Reviewer_q2iV · 2026-03-09

**Soundness:** 2
**Presentation:** 3
**Significance:** 2
**Originality:** 2
**Overall Recommendation:** 2
**Confidence:** 4

**Summary:**

This work presents a transfer learning framework designed for nonparametric regression across distinct, heterogeneous sub-datasets. The approach hinges on the assumption that data structures across groups are additive, comprising both shared and subset-specific information. In the first stage, the method pools all subsets to estimate the common structure via deep learning; in the second stage, it focuses on capturing the idiosyncratic features of each subset. This two-stage estimation leads to the final regression results. Besides the methodological contribution, the paper provides a theoretical analysis of convergence and evaluates the model's performance through comprehensive simulations and real-data applications.

**Compliance With Llm Reviewing Policy:**

Affirmed.

**Key Questions For Authors:**

1. Could the authors clarify the specific problem setup? The term 'heterogeneous transfer' is mentioned but not explicitly defined. I would appreciate it if the authors could elaborate on the following:
（1). What constitutes 'heterogeneity' in your framework?
（2). What are the key characteristics of the datasets that distinguish this from standard transfer learning？
2. Are the assumptions concerning the shared latent structure and structural additivity overly strong for practical applications? How do existing approaches in transfer learning relax or address these specific assumptions? Furthermore, what is the technical differentiation of this work compared to those prior methods in terms of novelty?

**Limitations:**

1. The paper lacks a clear description of the transfer mechanism. It remains ambiguous how the source and target domains are characterized and how the information flow between them is managed. If the framework’s efficacy hinges on extracting shared structural information, then providing guarantees or diagnostics for the quality of this information should be a foundational component of the work. Currently, the analysis in this area lacks sufficient depth.
2. The paper asserts that the proposed method achieves a fast rate and overcomes the curse of dimensionality, yet these points are not adequately discussed. There is no clear theoretical justification explaining why the rate is 'fast' in comparison to existing nonparametric estimators. Furthermore, the claim regarding the curse of dimensionality is problematic: without demonstrating an explicit dimensionality reduction process or a specific compositional structure in the functional class, it is unclear how the estimator avoids the typical exponential dependence on the input dimension. This lack of explanation constitutes a significant gap in the theoretical narrative.
3. Regarding Remark 3.9, the comparison between the two-stage method and the independent data approach hinges on the difference between $n$ and $n\underline{\pi}{l}$. However, if the sample proportion $\underline{\pi}{l}$ for each group remains asymptotically constant as $n \to \infty$, the convergence rates in Theorem 3.4 and Corollary 3.8 are of the same order. This raises a critical question: what is the fundamental theoretical advantage of the proposed method? If the improvement is merely a constant factor rather than an improvement in the convergence rate (order of $n$), the claimed distinction between these two approaches lacks significant statistical weight.

**Strengths And Weaknesses:**

Strengths:
1. Utilizing deep neural networks as a function class for nonparametric regression is a sound and well-justified approach.
2. The paper provides a comprehensive theoretical analysis, specifically investigating the impact of using independent versus non-independent data within the proposed two-stage framework.
3. This work designed a comprehensive experimental workflow incorporating both simulated and real data, and conducted tests on independent data and non-flat data.

Weaknesses:
1 The experiments evaluate the proposed method on both synthetic and real-world datasets. While the results on synthetic data are promising, the performance gains on real-world datasets appear less pronounced. Does the effectiveness of the method hinge on the specific data-generating process (DGP) assumed in the simulations?
2. The results show a substantial difference between the RF-based and NN-based approaches. To better understand the source of the improvement, could the authors clarify whether the gains are primarily driven by the network architecture itself? Specifically, would the proposed two-stage methodology yield similar benefits if applied to other base learners, such as Random Forests?
3. Regarding the experiments on the UTKFace dataset, the authors utilized weights pretrained on VGGFace via FaceNet. To isolate the contribution of the proposed framework and eliminate the influence of pretraining as a confounding factor, I suggest including a baseline using randomly initialized weights. This would clarify whether the performance gains are primarily driven by the methodology or the high-quality feature representations from the pretrained model.
4. The inclusion of imbalanced data scenarios in the experimental design is commendable, as it effectively demonstrates the robustness of the proposed method under non-ideal data distributions.
5. Considering that Wang & Schneider established the transferability of add-on models and Cagnetta explored how hierarchical networks mitigate the curse of dimensionality, the conceptual novelty of this work appears somewhat incremental. The proposed framework seems to be a straightforward combination of these existing ideas. Could the authors clarify the distinct technical contributions that set this work apart from these prior studies?
6.Could the authors clarify the theoretical advantages of the additive model hypothesis compared to composite fine-tuning (FT)? Given that high-dimensional unstructured data (e.g., images, text) typically exhibit complex, non-linear dependencies, is the additivity assumption overly restrictive ? I am concerned about whether such a framework can effectively capture the intricate structures and high-order interactions present in complex, high-dimensional data.

---

> ### Author Rebuttal · Authors · 2026-03-29
>
> We greatly thank the reviewer for the valuable feedback and respond to each point below. We apologize that, due to space limitations, we must refer to our responses to other reviewers for certain points.
>
> Weaknesses
>
> 1. Our framework is estimator-agnostic with no DGP assumptions, validated across 8 simulation scenarios with various SNR settings, yielding consistent gains for both DNNs and random forest (RF). On UTKFace, our method achieves best overall performance across ethnicity groups, robust to different group partitions and direct MLP modeling (Appendices E.3.1–E.3.2). On Beijing PM2.5 (Appendix E.1), when trained with our strategy, DNN ranks best overall and random forest leads among competitors.
>
> 2. Our method yields consistent gains across base learners. Besides NN, in 6 simulation scenarios and Beijing PM2.5, RF trained with our two stage strategy also outperforms pooling, pooling-with-label, and separate learning within the RF class. Performance gaps between RF and NN stem from estimator intrinsic capacities, as DNNs are known to have strong capacity for modeling complex and high-dimensional functions. Theoretically, our framework recovers established rates for kernel ridge regression and yields new results for trend filtering ("Adaptive Piecewise Polynomial Estimation via Trend Filtering"). We are ready to present these in the next rebuttal round due to space limits.
>
> 3. The main text follows the more practical pretrained setting; Appendix E.3.2 reports results with randomly initialized MLP trained from scratch, where our method still achieves best overall performance, confirming gains stem from our framework.
>
> 4. We thank the reviewer for the positive feedback.
>
> 5. We thank the reviewer for the question. We refer to our response to Weakness Q1 (Reviewer YeVm) for a detailed clarification of our contributions relative to prior work, and to Q2 for further discussion.
>
> 6. To our knowledge, theoretical properties of last-layer FT under the same pretraining scheme remain largely unexplored nonparametrically, though we include it as a baseline in all experiments. While our model adopts an additive formulation, both shared and group-specific components are allowed to be separately modeled as hierarchical compositional functions (empirically capture many complex real-world processes [1]), enabling highly complex and high-dimensional representations. We also validate on high-dimensional and non-explicitly-additive settings across simulations and real data, demonstrating its flexibility.
>
> Key Questions
>
> 1. (1) Heterogeneity refers to variation in regression functions and noise distributions across all groups; (2) Our setting includes the data-scarce regime ($n_\ell \ll n$), and also allows $n_\ell \asymp n$ while still achieving faster rates. Practically, the distinction mirrors between transfer learning (TL) and multi-task learning (MTL): beyond improving a single target group, we aim to improve performance across most groups, as supported by real-data results (see our response to Reviewer YeVm Weaknesses Q3 for details).
>
> 2. We thank the reviewer for the comment. The adaptability of additive structure has been discussed in the Weakness section. Regarding novelty, we refer to our response to Reviewer YeVm Weaknesses Q1 for a thorough clarification of our contributions relative to prior work.
>
> Limitations
>
> 1. Transfer occurs via two-stage decomposition: $g_{0,\ell}(x) = \bar{f}(x) + G_\ell(x)$, where $\bar{f}(x) = f_0(x) + \sum_{\ell'} \mathbb{P}(Z=\ell' \mid X=x)f_{0,\ell'}(x)$ aggregates cross-group information using all data, and $G_\ell(x) = f_{0,\ell}(x) - \sum_{\ell'}\mathbb{P}(Z=\ell' \mid X=x)f_{0,\ell'}(x)$ is the group-specific offset learned in Stage 2. In Remark 3.5, we identify two transfer mechanisms: (i) learning simpler functions via averaging and offset learning, and (ii) data augmentation. For (i) We notice that because of cross-group aggregation: $\bar{f}$ can be smoother than group-specific $g_{0,\ell}$ due to averaging over group deviations, and $G_\ell$ can be simpler than $g_{0,\ell}$ as the shared component $f_0$ is removed, and when the groups are similar we can expect  $G_{\ell}(x)$ to be simple. This enables positive transfer even when $n_\ell \asymp n$, without requiring $n_\ell \ll n$.
>
> 2. We have added a concrete example for clarity; see our response to Reviewer 55mL item [1] for details.
>
> 3. It is right that when $\pi_\ell$ remains constant, the rates in Theorem 3.4 and Corollary 3.8 are of the same order, but the distinction becomes important when $\pi_\ell \to 0$. Both results are new: Theorem 3.4 concerns the estimator using the full sample in both stages, while Corollary 3.8 applies to the data-splitting version. The potential benefit of data splitting is noted, though whether this reflects a genuine statistical advantage or a proof artifact remains open.
>
> [1] Sclocchi et al., 2025, PNAS, "A Phase Transition in Diffusion Models Reveals the Hierarchical Nature of Data".

---

> > ### Author Rebuttal · Reviewer_q2iV · 2026-04-02
> >
> > Thanks for the authors' response.  However, the explanation of heterogeneity is still not sufficiently clear; providing a concrete example would help clarify this point. In addition, the discussion of $\pi_{\ell}$ remains limited. As the authors claim this part to be a new result, it would be important to provide comparisons with existing literature to better highlight its novelty and advantages.

---

> > > ### Author Response · Authors · 2026-04-04
> > >
> > > We thank the reviewer for the continued engagement.
> > >
> > > **(1) Discussion of heterogeneity.**
> > >
> > > We agree that "heterogeneity'' may be imprecise and are willing to remove it in the final version. Our original use of the term refers to the fact that each group can have its own regression function and noise distribution, which we clarify below through a concrete example.
> > >
> > > Let $X\in[0,1]^{4}$ and $l\in\{1,2,3\}$ denote the group index with proportion $\pi_l$. We observe
> > >
> > > $Y_{i}=g_{0,l}(X_i)+\varepsilon_{l,i},$
> > >
> > > and consider the additive decomposition
> > >
> > > $g_{0,l}(x)=f_0(x)+f_{0,l}(x), \quad l=1,2,3,$
> > >
> > > with shared component $f_0(x)=\sqrt{x_1+x_2+x_3}$ and group-specific components
> > >
> > > $f_{0,1}(x)=\vert x_4 - 1/2 \vert, \quad f_{0,2}(x)=2\vert x_4 - 1/2 \vert ,\quad f_{0,3}(x)= 3\vert x_4 - 1/2 \vert.$
> > >
> > > Each group may further have its own noise distribution; for example,
> > >
> > > $\varepsilon_{l,i}\sim\mathcal{N}(0,l\cdot\sqrt{X^{(l)}_{i}})$,
> > >
> > > where $X_i^{(l)}$ denotes the $l$-th coordinate of $X_i$. Since each $g_{0,l}$ and the distribution of $\varepsilon_l$ can differ across groups, we refer to this as a heterogeneous setting.
> > >
> > > **(2) Rate comparison and the discussion on the role of $\pi_l$.**
> > >
> > > We now use the example above to compare convergence rates under different $\pi_l$ with directly comparable classical estimators under the similar offset TL framework mentioned in our main text (Wang et al., 2016; Du et al., 2017; Lin \& Reimherr, 2024; Cai \& Pu, 2024), and illustrate the novelty of our results.
> > >
> > > > **a) Without transfer learning.**
> > >
> > > In this example, the smoothness of $g_{0,l}$ is $p=0.5$ with intrinsic dimensionalities $K=4$. Training each group separately on $n\pi_l$ samples, classical estimators must use all $d=4$ dimensions, yielding rate
> > >
> > > $(n\pi_l)^{-\frac{2p}{2p+d}}=(n\pi_l)^{-1/5}.$
> > >
> > > Direct NN estimation without transfer learning exploits intrinsic dimensionality $K=4$, also giving
> > >
> > > $(n\pi_l)^{-\frac{2p}{2p+K}}=(n\pi_l)^{-1/5}.$
> > >
> > > This is a deliberately chosen example where direct NN estimation without TL cannot improve upon the classical rate, specifically constructed to motivate the benefit of TL with NN in the subsequent comparison. We note that in most settings, NN estimation does outperform classical methods by exploiting intrinsic dimensionality; in particular, when $d > 4$ but $K=4$, the NN achieves a strictly faster rate by overcoming the curse of dimensionality.
> > >
> > > (Remark: We note that the same function may admit different hierarchical compositions with different intrinsic dimensionalities, since our upper bounds apply at the function class level rather than to a specific function instance. The composition with intrinsic dimension $K=4$ chosen here is just for illustrative purposes.)
> > >
> > > > **b) With transfer learning.**
> > >
> > > The first-stage target is $$\bar{f}(x)=\sqrt{x_1+x_2+x_3}+(\pi_1+2\pi_2+3\pi_3) \cdot \vert x_4-1/2\vert,$$ which has smoothness $p=0.5$ and intrinsic dimension $K=4$. The second-stage offset (taking group 1 as an example) is $$G_1(x)=(1-{\pi_1-2\pi_2-3\pi_3})\cdot\vert x_4-1/2\vert,$$ which has smoothness $p=1$ and intrinsic dimension $K=1$, hence simpler than $g_{0,l}$ and $\bar{f}$. Following the same rate calculations as in our response to Weakness Q4 of Reviewer 55mL, classical estimators, which require all 4 dimensions for both $\bar{f}$ and $G_1$, yield rate $$n^{-1/5}+(n\pi_1)^{-1/3}.$$ Our Theorem 3.7, by exploiting the intrinsic dimensionality of $G_1$, yields rate $$n^{-1/5}+(n\pi_1)^{-2/3}.$$ The four regimes of convergence rates for all four estimators as $\pi_1$ varies are summarized in the table below.
> > >
> > > | | Classical (no TL) | NN (no TL) | Classical (TL) | NN (TL) |
> > > |---|:---:|:---:|:---:|:---:|
> > > | $\pi_1 \asymp 1$ | $n^{-1/5}$ | $n^{-1/5}$ | $n^{-1/5}$ | $n^{-1/5}$ |
> > > | $n^{-2/5} \ll \pi_1 \ll 1$ | $(n\pi_1)^{-1/5}$ | $(n\pi_1)^{-1/5}$ | $n^{-1/5}$ | $n^{-1/5}$ |
> > > | $n^{-7/10} \lesssim \pi_1 \lesssim n^{-2/5}$ | $(n\pi_1)^{-1/5}$ | $(n\pi_1)^{-1/5}$ | $(n\pi_1)^{-1/3}$ | $n^{-1/5}$ |
> > > | $\pi_1 \ll n^{-7/10}$ | $(n\pi_1)^{-1/5}$ | $(n\pi_1)^{-1/5}$ | $(n\pi_1)^{-1/3}$ | $(n\pi_1)^{-2/3}$ |
> > >
> > > *Table: Convergence rates under different regimes of $\pi_1$.*
> > >
> > >
> > > With this table:
> > >
> > > 1. When $\pi_1 \asymp 1$, all methods achieve $n^{-1/5}$ and TL provides no additional gain.
> > >
> > > 2. When $n^{-2/5}\ll\pi_1 \ll1$, both classical and NN methods with TL achieve the faster rate $n^{-1/5}$, while classical and NN without TL are slower at $(n\pi_1)^{-1/5}$.
> > >
> > > 3. When $n^{-7/10} \lesssim \pi_1 \lesssim n^{-2/5}$, NN with TL still achieves $n^{-1/5}$ while the classical TL rate degrades to $(n\pi_1)^{-1/3}$; for instance, at $\pi_1 \asymp n^{-7/10}$, our rate is $n^{-1/5}$ versus $n^{-1/10}$ for classical TL.
> > >
> > > 4. When $\pi_1 \ll n^{-7/10}$, NN with TL at $(n\pi_1)^{-2/3}$ remains strictly faster than classical TL at $(n\pi_1)^{-1/3}$.
> > >
> > > This example clearly illustrates how a small $\pi_1$ can drastically worsen the classical rate while NN with TL remains robust.

---

### Decision · Program_Chairs · 2026-04-30

**Decision:**

Accept (regular)

**Comment:**

This paper develops a two-stage transfer learning framework for nonparametric regression with heterogeneous grouped data, providing general upper bounds for a broad class of estimators and explicit convergence rates for deep ReLU networks under hierarchical composition models.

The positive reviewers found the proofs non-trivial and correct, acknowledged the framework's generality, and had all concerns resolved.
The negative reviewers argued the contribution is incremental, combining a well-established two-stage offset procedure with known DNN approximation techniques. However, please note that existing nonparametric TL theories with concrete rates are all tied to classical estimators, such as kernel methods, orthogonal series, local polynomials, whose proofs do not extend to DNNs. Closing such a gap is a clear contribution, not a routine exercise.

For the final version, the authors should more carefully distinguish their TL framing from multi-task learning, incorporate the concrete rate examples from the rebuttal into the main text, and expand the discussion of negative transfer.